# Hodge-Aware Convolutional Learning on Simplicial Complexes

**Maosheng Yang**                                        *m.yang-2@tudelft.nl*
*Department of Intelligent Systems*
*Delft University of Technology*

**Geert Leus**                                           *g.j.t.leus@tudelft.nl*
*Department of Microelectronics*
*Delft University of Technology*

**Elvin Isufi**                                          *e.isufi-1@tudelft.nl*
*Department of Intelligent Systems*
*Delft University of Technology*

**Reviewed on OpenReview:** *https://openreview.net/forum?id=Nm5sp09Q25*

## Abstract

Neural networks on simplicial complexes (SCs) can learn representations from data residing on simplices such as nodes, edges, triangles, etc. However, existing works often overlook the Hodge theorem that decomposes simplicial data into three orthogonal characteristic subspaces, such as the identifiable gradient, curl and harmonic components of edge flows. This provides a universal tool to understand the machine learning models on SCs, thus, allowing for better principled and effective learning. In this paper, we study the effect of this data inductive bias on learning on SCs via the principle of convolutions. Particularly, we present a general convolutional architecture that respects the three key principles of uncoupling the lower and upper simplicial adjacencies, accounting for the inter-simplicial couplings, and performing higher-order convolutions. To understand these principles, we first use Dirichlet energy minimizations on SCs to interpret their effects on mitigating simplicial oversmoothing. Then, we show the three principles promote the Hodge-aware learning of this architecture, through the lens of *spectral simplicial theory*, in the sense that the three Hodge subspaces are invariant under its learnable functions and the learning in two nontrivial subspaces is independent and expressive. Third, we investigate the learning ability of this architecture in optic of perturbation theory on simplicial topologies and prove that the convolutional architecture is stable to small perturbations. Finally, we corroborate the three principles by comparing with methods that either violate or do not respect them. Overall, this paper bridges learning on SCs with the Hodge theorem, highlighting its importance for rational and effective learning from simplicial data, and provides theoretical insights to convolutional learning on SCs.

## 1 Introduction

In the line of geometric deep learning (Bronstein et al., 2021), there is a growing interest in learning from data defined on simplicial complexes. The motivation behind this comes from two limitations of standard graph neural networks (GNNs). First, graphs are limited to model pairwise interactions between data entites on nodes, yet polyadic (multi-way) interactions often arise in real-world networks (Battiston et al., 2020; Benson et al., 2021; Torres et al., 2021), such as friendship networks (Newman et al., 2002), collaboration networks (Benson et al., 2018), gene regulatory networks (Masoomy et al., 2021). Second, graphs are often used to support signals on the nodes, and standard graph signal processing and GNN approaches often

revolve around signals and features on nodes. Yet, signals involved with multiple entities are less researched compared to signals on nodes (with one entity). They arise as signal flows on edges, signals on triangles and so on. For example, in physical networks, we may encounter water flows in a water supply network (Money et al., 2022), traffic flows in a road network (Jia et al., 2019), trading flows in financial networks (Lim, 2020) and information flows in brain networks (Anand et al., 2022), as well as in human-generated networks, we have collaboration data, such as triadic collaborations in coauthorship networks (Benson et al., 2018).

Simplicial complexes are a popular higher-order network model and have been shown effective to address both limitations of graph-based models (Bick et al., 2021). They are composed of topological objects, namely, nodes, edges, triangles, etc., which are simplices of different orders. Simplicial complexes naturally describe more topological (higher-order) relationships in networks, thus, having more topological expressive power than graphs. This has been the main motivation of recent neural networks developed on simplicial complexes (Roddenberry & Segarra, 2019; Bunch et al., 2020; Ebli et al., 2020; Roddenberry et al., 2021; Bodnar et al., 2021b; Chen et al., 2022b; Giusti et al., 2022). We also refer readers to the recent surveys (Papamarkou et al., 2024; Besta et al., 2024). In analogy to standard GNNs relying on the adjacency between nodes, the central idea behind these works is to rely on the relationships between simplices to enable learning. Such relations can be twofold: first, two simplices can be lower and upper adjacent to each other, e.g., an edge can be (lower) adjacent to another via a shared node, and can also be (upper) adjacent to another by locating in a common triangle; and second, there exist inter-simplicial couplings (or simplicial incidences) between simplices of different orders, as shown in Fig. 1a. The aforementioned works mainly vary in terms of either message-passing or convolutional flavor, or the type of simplicial relationships relying, either on only simplicial adjacencies or on both adjacencies and incidences.

Furthermore, signals can be defined on simplices to model the data related to multiple entites in networks. This has been the main focus of topological signal processing literature Barbarossa & Sardellitti (2020); Schaub et al. (2021); Yang et al. (2022a). The celebrated *combinatorial Hodge decomposition* arising from discrete calculus (Grady & Polimeni, 2010; Lim, 2020) provides a unique and characteristic decomposition of simplicial signals into three components. This is particularly intuitive for edge flows which allows their decomposition into *gradient flows, curl flows and harmonic flows*, that are, respectively, curl-free, divergence-free or both. These notions from discrete calculus interestingly allow us to capture some physical properties of the simplicial signals, such as the conservation laws (Grady & Polimeni, 2010). More importantly, this decomposition offers a tool to better analyze simplicial signals, as reported in statistical ranking problems, financial exchange markets (Jiang et al., 2011), traffic networks (Jia et al., 2019), brain networks (Anand et al., 2022) and game theory (Candogan et al., 2011). We hypothesize it will further promote better principled and effective learning methods on simplicial complexes.

Given this context, we reckon that the aforementioned works on simplicial neural networks mostly focus on the pure topological aspect of simplicial complexes. It lacks theoretical analyses of their learning capabilities from the Hodge spectral perspective. Also, since SCs are often built from data and are prone to estimation uncertainty, the learning on SCs benefits from a stability analysis to investigate their robustness against perturbations on the simplicial topologies. Thus, in this paper, after reviewing some background on simplicial complexes and simplicial signals in Section 2, we propose a more general and unified framework, namely, *simplicial complex convolutional neural network* (SCCNN), and we focus on the following three theoretical aspects.

**Contributions**

- In Section 3 we introduce SCCNN and emphasize its three principles, namely, uncoupling the lower and upper simplicial adjacencies, accounting for the inter-simplicial couplings, and performing higher-order convolutions. We then use the Dirichlet energy minimization on SCs to understand how uncoupling the lower and upper adjacencies in Hodge Laplacians, as well as the inter-simplicial couplings can mitigate simplicial oversmoothing.

- In Section 4, we characterize the spectral behavior of SCCNN and its expressive power under the help of spectral simplicial theory (Steenbergen, 2013; Barbarossa & Sardellitti, 2020; Yang et al., 2021). We show that an SCCNN performs independent and expressive learning in the three subspaces of the

Hodge decomposition, which are invariant under its learning operators. This Hodge-awareness (or Hodge-aided bias) allows for effective and rational learning on SCs compared to MLPs or simplicial message-passing networks (Bodnar et al., 2021b).

- In Section 5, we obtain a theoretical stability bound on the SCCNN outputs against small perturbations on the simplicial connections. This allows us to see how the three principles and other network factors can affect the stability, as well as the limitations of SCCNNs. This analysis in turn guides the design of convolutional architectures.

In Section 6, we validate our theoretical findings and highlight the effect of the three principles, the need for the Hodge-aware learning, as well as the stability, based on different simplicial tasks including recovering foreign currency exchange (forex) rates, predicting triadic and tetradic collaborations, and ocean current trajectories. Finally, we conclude the paper in Section 7 with a discussion on this work and its relations to existing works.

## 2 Background

We first review simplicial complexes and data supported on simplices, which are natural generalizations of the corresponding notions on graphs. Then, we introduce discrete calculus on simplicial complexes, which is linked to the incidence matrices. Finally, we discuss the Hodge decomposition, which uniquely characterizes the simplicial signals from three subspaces.

### 2.1 Simplicial complex and simplicial signals

Given a set $\mathcal{V} = \{1, \dots, n_0\}$ of vertices, a $k$-simplex $s^k$ is a subset of $\mathcal{V}$ with cardinality $k + 1$. Geometrically, a node is a 0-simplex, an edge connecting two vertices is a 1-simplex and a triangular face (we shorten it as a triangle) is a 2-simplex. A subset, with cardinality $k$, of $s^k$ is a *face*. A *coface* of $s^k$ is a $(k+1)$-simplex that has $s^k$ as a face. Furthermore, one can collect $k$-simplices for $k = 0, \dots, K$ to form a simplicial complex (SC) $\mathcal{S}$ of order $K$ with the *inclusion* restriction that if a simplex is in the SC, so are its subsets. A graph is an SC of order one and by including some triangles, we obtain an SC of order two, as shown in Fig. 1a. We denote the set of all $k$-simplices in $\mathcal{S}$ as $\mathcal{S}^k = \{s_i^k\}_{i=1,\dots,n_k}$ where $n_k = |\mathcal{S}^k|$, i.e., $\mathcal{S} = \cup_{k=0}^K \mathcal{S}^k$.

**Simplicial adjacency** For any two $k$-simplices, we say they are *lower (upper) adjacent* if they share a common face (coface), which natualy defines the notion of simplicial neighborhoods. For example, two nodes are (upper) adjacent in a graph if they are connected by an edge. In Fig. 1a, edges $e_1$ and $e_3$ are lower neighbors as they share node 1; while $e_1$ and $e_2$ are upper neighbors since they are located in the triangle $t_1$.

**Orientation** For computational purposes, we annotate each simplex with an *orientation*, as an ordering of the labels of its vertices (a node has a trivial orientation). Here we consider the increasing ordering as the reference orientation, that is, a triangle $s^2 = \{i, j, k\}$ is oriented as $[i, j, k]$ for $i < j < k$, and an edge $s^1 = \{i, j\}$ is oriented as $[i, j]$ for $i < j$.

**Algebraic representation** We use the incidence matrix $\boldsymbol{B}_k \in \mathbb{R}^{n_{k-1} \times n_k}$ to describe the relationships between $(k - 1)$- and $k$-simplices. Thus, $\boldsymbol{B}_1$ encodes the node-to-edge incidence and $\boldsymbol{B}_2$ the edge-to-triangle incidence. In an oriented SC $\mathcal{S}$, the entries of $\boldsymbol{B}_1$ and $\boldsymbol{B}_2$ are given by

$$[\boldsymbol{B}_1]_{ie} = \begin{cases} -1, & \text{for } e = [i, \cdot] \\ 1, & \text{for } e = [\cdot, i] \\ 0, & \text{otherwise.} \end{cases} \quad [\boldsymbol{B}_2]_{et} = \begin{cases} 1, & \text{for } e = [i, j], t = [i, j, k] \\ -1, & \text{for } e = [i, k], t = [i, j, k] \\ 1, & \text{for } e = [j, k], t = [i, j, k] \\ 0, & \text{otherwise.} \end{cases} \tag{1}$$

We further define the *$k$-Hodge Laplacian*

$$\boldsymbol{L}_k = \boldsymbol{B}_k^\top \boldsymbol{B}_k + \boldsymbol{B}_{k+1} \boldsymbol{B}_{k+1}^\top \tag{2}$$

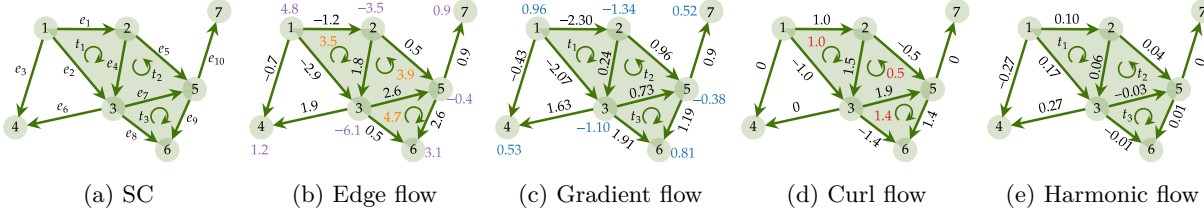

Figure 1: (a) A simplicial 2-complex where green shaded triangles denote 2-simplices and the arrows denote the chosen reference orientations. (b) An edge flow where we denote its divergence (div) and curl in purple and orange, respectively. (c)-(d) The Hodge decomposition of the edge flow in (b). The gradient flow is the gradient of some node signal (in blue) and is curl-free. The curl flow can be obtained from some triangle flow (in red), and is div-free. The harmonic flow has zero div and zero curl, and is circulating around the hole $\{1, 3, 4\}$. Note that in this figure, the flow numbers are rounded up to two decimal places. Thus, at some nodes or triangles with zero-div or zero-curl, the div or curl might not be exactly zero.

with the *lower Laplacian* $\boldsymbol{L}_{k,\mathrm{d}} = \boldsymbol{B}_k^\top \boldsymbol{B}_k$ and the *upper Laplacian* $\boldsymbol{L}_{k,\mathrm{u}} = \boldsymbol{B}_{k+1} \boldsymbol{B}_{k+1}^\top$. We have a set of $\boldsymbol{L}_k, k = 1, \ldots, K-1$ in an SC of order $K$ with $\boldsymbol{L}_0 = \boldsymbol{B}_1 \boldsymbol{B}_1^\top$ the graph Laplacian, and $\boldsymbol{L}_K = \boldsymbol{B}_K^\top \boldsymbol{B}_K$. Topologically, $\boldsymbol{L}_{k,\mathrm{d}}$ and $\boldsymbol{L}_{k,\mathrm{u}}$ encode the lower and upper adjacencies of $k$-simplices, respectively. For example, $\boldsymbol{L}_{1,\mathrm{d}}$ encodes the edge-to-edge adjacencies through nodes while $\boldsymbol{L}_{1,\mathrm{u}}$ encodes the adjacencies through triangles.

## 2.2 Simplicial signals and Hodge decomposition

**Simplicial signals** A *$k$-simplicial signal* (or data) $\boldsymbol{x}_k \in \mathbb{R}^{n_k}$ supported on the simplicial set $\mathcal{S}^k$ is defined by an *alternating* map $f_k : \mathcal{S}^k \to \mathbb{R}^{n_k}$, which assigns a real value to a simplex, with the condition that if the orientation of a simplex is anti-aligned with the reference orientation, then the signal will change the sign (Lim, 2020). For convenience, we call $\boldsymbol{x}_0$ a *node signal*. We also refer to a 1-simplicial signal $\boldsymbol{x}_1$ as an *edge flow* and a 2-simplicial signal $\boldsymbol{x}_2$ as a *triangle flow*. A $d$-dimensional simplicial feature $\boldsymbol{X}_k \in \mathbb{R}^{n_k \times d}$ can be defined for a rich representation learning of simplices. For simplicity, we restrict our analysis to $d = 1$.

**Incidence matrices as derivatives on SCs** Given a simplicial signal $\boldsymbol{x}_k$, we can measure its variability with respect to the faces and cofaces of $k$-simplices by computing $\boldsymbol{B}_k \boldsymbol{x}_k$ and $\boldsymbol{B}_{k+1}^\top \boldsymbol{x}_k$ (Grady & Polimeni, 2010). Specifically, $\boldsymbol{B}_1^\top \boldsymbol{x}_0$ computes the *gradient* of a node signal $\boldsymbol{x}_0$ as the signal difference between the adjacent nodes, i.e., $[\boldsymbol{B}_1^\top \boldsymbol{x}_0]_{[i,j]} = [\boldsymbol{x}_0]_j - [\boldsymbol{x}_0]_i$, which is often used in the GNN literature. For an edge flow $\boldsymbol{x}_1$, $\boldsymbol{B}_1 \boldsymbol{x}_1$ computes its *divergence*, which is the difference between the total in-flow and out-flow at node $j$, i.e., $[\boldsymbol{B}_1 \boldsymbol{x}_1]_j = \sum_{i<j} [\boldsymbol{x}_1]_{[i,j]} - \sum_{j<k} [\boldsymbol{x}_1]_{[j,k]}$. Moreover, $\boldsymbol{B}_2^\top \boldsymbol{x}_1$ computes the *curl* of $\boldsymbol{x}_1$, i.e., $[\boldsymbol{B}_2^\top \boldsymbol{x}_1]_t = [\boldsymbol{x}_1]_{[i,j]} - [\boldsymbol{x}_1]_{[i,k]} + [\boldsymbol{x}_1]_{[j,k]}$, which is the net-flow circulation in triangle $t = [i, j, k]$. As illustrated in Fig. 1b, these two computations provide divergent and rotational variation measures of an edge flow, which are analogous to the notions of divergence and curl for vector fields in continuous domains. In the following, we introduce the Hodge decomposition (Hodge, 1989; Lim, 2020) which unfolds an edge flow into three unique characteristic components.

**Lemma 1** ((Lim, 2020)). *We have $\boldsymbol{B}_2^\top \boldsymbol{B}_1^\top = \boldsymbol{0}$, i.e., the curl of the gradient is zero.*

**Theorem 2** (Hodge decomposition). *The $k$-simplicial signal space $\mathbb{R}^{n_k}$ admits a direct sum decomposition*

$$\mathbb{R}^{n_k} = \mathrm{im}(\boldsymbol{B}_k^\top) \oplus \ker(\boldsymbol{L}_k) \oplus \mathrm{im}(\boldsymbol{B}_{k+1}), \;\; \textit{thus,} \;\; \boldsymbol{x}_k = \boldsymbol{x}_{k,\mathrm{G}} + \boldsymbol{x}_{k,\mathrm{H}} + \boldsymbol{x}_{k,\mathrm{C}}, \tag{3}$$

*where $\boldsymbol{x}_{k,\mathrm{G}} = \boldsymbol{B}_k^\top \boldsymbol{x}_{k-1}$ for some $\boldsymbol{x}_{k-1}$, and $\boldsymbol{x}_{k,\mathrm{C}} = \boldsymbol{B}_{k+1} \boldsymbol{x}_{k+1}$ for some $\boldsymbol{x}_{k+1}$. Moreover, we have $\ker(\boldsymbol{B}_{k+1}^\top) = \mathrm{im}(\boldsymbol{B}_k^\top) \oplus \ker(\boldsymbol{L}_k)$ and $\ker(\boldsymbol{B}_k) = \ker(\boldsymbol{L}_k) \oplus \mathrm{im}(\boldsymbol{B}_{k+1})$.*

In the node space, this decomposition is trivial as $\mathbb{R}^{n_0} = \ker(\boldsymbol{L}_0) \oplus \mathrm{im}(\boldsymbol{B}_1)$ where the kernel of $\boldsymbol{L}_0$ contains constant node data and the image of $\boldsymbol{B}_1$ contains nonconstant data. In the edge case, the three subspaces carry more tangible meaning: the *gradient space* $\mathrm{im}(\boldsymbol{B}_1^\top)$ collects edge flows as the gradient of some node signal, which are *curl-free*; the *curl space* $\mathrm{im}(\boldsymbol{B}_2)$ consists of flows cycling around triangles, which are *div-free*; and flows in the *harmonic space* $\ker(\boldsymbol{L}_1)$ are both div- and curl-free. In this paper, we inherit the names

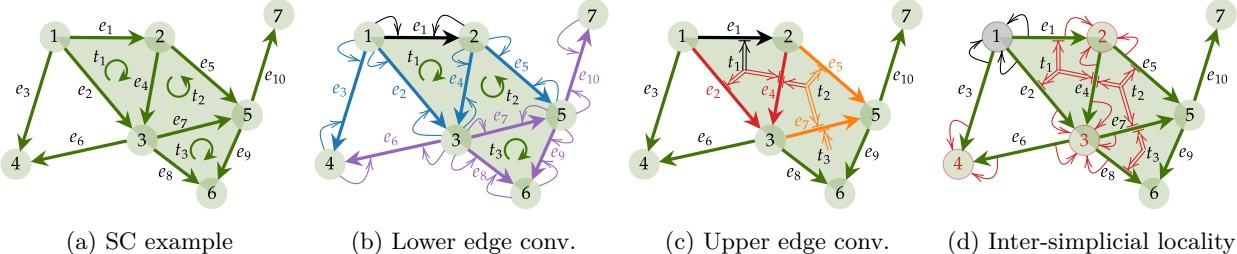

| (a) SC example | (b) Lower edge conv. | (c) Upper edge conv. | (d) Inter-simplicial locality |

Figure 2: (a) An SC where arrows indicate the reference orientations of edges and triangles. 2-simplices are (filled) triangles shaded in green and open triangle $\{1, 3, 4\}$ is not in the SC. (b) Lower convolution via $\boldsymbol{H}_1$ and $\boldsymbol{H}_{1,\mathrm{d}}$ on edge $e_1$. (c) Upper convolution via $\boldsymbol{H}_1$ and $\boldsymbol{H}_{1,\mathrm{u}}$ on $e_1$. (d) Node 1 (in black) contains information from its neighbors $\{2, 3, 4\}$ (nodes in red), and projected information from edges which contribute to these neighbors (denoted by arrows in red from edges to nodes), and from triangles $\{t_1, t_2, t_3\}$ which contribute to those edges (denoted by double arrows in red from triangle centers to edges). This interaction is the coupling between the intra- and the extended inter-simplicial locality.

of three edge subspaces to general $k$-simplices. The above theorem states that any simplicial signal $\boldsymbol{x}_k$ can be uniquely expressed as $\boldsymbol{x}_k = \boldsymbol{x}_{k,\mathrm{G}} + \boldsymbol{x}_{k,\mathrm{H}} + \boldsymbol{x}_{k,\mathrm{C}}$ with the gradient part $\boldsymbol{x}_{k,\mathrm{G}} = \boldsymbol{B}_k^\top \boldsymbol{x}_{k-1}$, the curl part $\boldsymbol{x}_{k,\mathrm{C}} = \boldsymbol{B}_{k+1} \boldsymbol{x}_{k+1}$, for some $\boldsymbol{x}_{k\pm 1}$, and the harmonic part following $\boldsymbol{L}_k \boldsymbol{x}_{k,\mathrm{H}} = \boldsymbol{0}$. Figs. 1c to 1e provide the three Hodge components of the edge flow in Fig. 1b.

## 3 Simplicial Complex CNNs

We first introduce the general convolutional architecture on SCs, then discuss the propreties of SCCNN and study the effects of the three principles from an energy minimization perspective.

In an SC, taking $\boldsymbol{x}_{k-1}^{l-1}, \boldsymbol{x}_k^{l-1}$ and $\boldsymbol{x}_{k+1}^{l-1}$ as inputs, an SCCNN at layer $l = 1, \dots, L$ computes the $k$-simplicial output $\boldsymbol{x}_k^l$ via a map

$$\mathrm{SCCNN}_k^l : \{\boldsymbol{x}_{k-1}^{l-1}, \boldsymbol{x}_k^{l-1}, \boldsymbol{x}_{k+1}^{l-1}\} \to \boldsymbol{x}_k^l, \quad \boldsymbol{x}_k^l = \sigma(\boldsymbol{H}_{k,\mathrm{d}}^l \boldsymbol{x}_{k,\mathrm{d}}^{l-1} + \boldsymbol{H}_k^l \boldsymbol{x}_k^{l-1} + \boldsymbol{H}_{k,\mathrm{u}}^l \boldsymbol{x}_{k,\mathrm{u}}^{l-1}), \tag{4}$$

with

$$\boldsymbol{H}_k = \sum_{t=0}^{T_\mathrm{d}} w_{k,\mathrm{d},t} \boldsymbol{L}_{k,\mathrm{d}}^t + \sum_{t=0}^{T_\mathrm{u}} w_{k,\mathrm{u},t} \boldsymbol{L}_{k,\mathrm{u}}^t, \quad \boldsymbol{H}_{k,\mathrm{d}} = \sum_{t=0}^{T_\mathrm{d}} w_{k,\mathrm{d},t}' \boldsymbol{L}_{k,\mathrm{d}}^t, \quad \boldsymbol{H}_{k,\mathrm{u}} = \sum_{t=0}^{T_\mathrm{u}} w_{k,\mathrm{u},t}' \boldsymbol{L}_{k,\mathrm{u}}^t. \tag{5}$$

Here, $\boldsymbol{H}_k$ denotes a *simplicial convolutional filter* (SCF, (Yang et al., 2022b)) with two sets of learnable coefficients $\{w_{k,\mathrm{d},t}\}, \{w_{k,\mathrm{u},t}\}$, while $\boldsymbol{H}_{k,\mathrm{d}}$ and $\boldsymbol{H}_{k,\mathrm{u}}$ are the lower and upper SCFs, respectively. Moreover, $\boldsymbol{x}_{k,\mathrm{d}}^{l-1} = \boldsymbol{B}_k^\top \boldsymbol{x}_{k-1}^{l-1}$ and $\boldsymbol{x}_{k,\mathrm{u}}^{l-1} = \boldsymbol{B}_{k+1} \boldsymbol{x}_{k+1}^{l-1}$ are the lower and upper projections from $(k \pm 1)$-simplices via incidence relations to $k$-simplices, respectively, and $\sigma(\cdot)$ is an elementwise nonlinearity. The convolution operations in this SCCNN can be understood as follows: 1) The previous $k$-simplicial output $\boldsymbol{x}_k^{l-1}$ is passed to an SCF $\boldsymbol{H}_k^l$ of orders $T_\mathrm{d}, T_\mathrm{u}$, which performs a linear combination of the signals from the lower-adjacent (up to $T_\mathrm{d}$-hop) and upper-adjacent (up to $T_\mathrm{u}$-hop) simplices. 2) The previous $k \pm 1$-simplicial outputs $\boldsymbol{x}_{k\pm 1}^{l-1}$ are first projected to $k$-simplices, which are then convolved using a lower SCF and an upper SCF, respectively.
*Example* 3. In Fig. 2, we provide an example of SCCNN for the edge case $k = 1$. We focus on edge $e_1$ and consider the cases $T_\mathrm{d} = T_\mathrm{u} = 2$. On edge $e_1$, the SCF $\boldsymbol{H}_1$ linearly combines the signals from its direct lower neighbors (edges in blue) and two-hop lower neighbors (edges in purple), as shown in Fig. 2b. It also combines the signals from the direct upper neighbors (edges in red) and two-hop upper neighbors (edges in orange), as shown in Fig. 2c. At the same time, the signals on nodes are projected on the edges, denoted by arrows in blue and purple from nodes to edges in Fig. 2b, which are then combined to edge $e_1$ by the lower SCF $\boldsymbol{H}_{1,\mathrm{d}}$. The signals on triangles are projected on the edges as well, denoted by double arrows in red and orange in Fig. 2c, which are combined to edge $e_1$ by the upper SCF $\boldsymbol{H}_{1,\mathrm{u}}$.

This architecture subsumes the convolutional learning methods on SCs in Bunch et al. (2020); Ebli et al. (2020); Roddenberry et al. (2021); Yang et al. (2022a); Chen et al. (2022b); Yang et al. (2022c). We refer

to Appendix C for a detailed discussion. Particularly, we here emphasize on the key three principles of an SCCNN layer:

(P1) It uncouples the lower and upper parts in the Hodge Laplacian. This leads to an independent treatment of the lower and upper adjacencies, achieved by using two sets of learnable weights. We shall see in Section 4 that how this relates to the independent and expressive learning in the Hodge subspaces given in Theorem 2.

(P2) It accounts for the inter-simplicial couplings via the incidence relations. The projections $\boldsymbol{x}_{k,\mathrm{d}}$ and $\boldsymbol{x}_{k,\mathrm{u}}$ carry nontrivial information contained in the faces and cofaces of simplices (by Theorem 2).

(P3) It performs higher-order convolutions. We consider $T_\mathrm{d}, T_\mathrm{u} \geq 1$ in SCFs which leads to a multi-hop receptive field on SCs.

In short, each SCCNN layer propagates information across SCs based on two simplicial adjacencies and two incidences in a multi-hop fashion.

## 3.1 Properties

**Simplicial locality** The simplicial convolutions admit an intra-simplicial locality where the output $\boldsymbol{H}_k\boldsymbol{x}_k$ is localized in $T_\mathrm{d}$-hop lower and $T_\mathrm{u}$-hop upper $k$-simplicial neighborhoods (Yang et al., 2022b). An SCCNN preserves such property as $\sigma(\cdot)$ does not alter the information locality. It also admits an inter-simplicial locality between $k$- and $(k \pm 1)$-simplices due to the inter-simplicial couplings. This further extends to simplices of orders $k \pm l$ if $L \geq l$ because $\boldsymbol{B}_k\sigma(\boldsymbol{B}_{k+1}) \neq \boldsymbol{0}$ (Schaub et al., 2021). Moreover, the intra- and inter-simplicial localities are coupled in a multi-hop way through higher-order convolutions such that, for example, a node not only interacts with its incident edges and the triangles including it, but also with those further hops away, as shown in Fig. 2d. We refer to Appendix B.1 for a more formal discussion.

**Complexity** An SCCNN layer has a parameter complexity of order $\mathcal{O}(T_\mathrm{d} + T_\mathrm{u})$ and a computational complexity $\mathcal{O}(k(n_k + n_{k+1}) + n_k m_k(T_\mathrm{d} + T_\mathrm{u}))$, which are linear in the simplex dimensions. Here, $m_k$ is the maximum number of neighbors for $k$-simplices. We refer to Appendix B.2 for more details.

**Equivariance** SCCNNs are permutation-equivairant, which allows us to list simplices in any order. They are also orientation-equivariant if the activation function $\sigma(\cdot)$ is odd, which gives us the freedom to choose reference orientations. In Appendix B.3, we provide a more formal discussion on such equivariances and why permutations form a symmetry group of an SC and orientation changes are symmetries of data spaces but not SCs.

## 3.2 A simplicial Dirichlet energy perspective

Here we analyze the convolution architecture in Eq. (4) from an energy minimization perspective. First, we extend the notion of Dirichlet energy from graphs to SCs.

**Definition 4.** The *Dirichlet energy* of a $k$-simplicial signal $\boldsymbol{x}_k$ is

$$D(\boldsymbol{x}_k) = D_\mathrm{d}(\boldsymbol{x}_k) + D_\mathrm{u}(\boldsymbol{x}_k) := \|\boldsymbol{B}_k\boldsymbol{x}_k\|_2^2 + \|\boldsymbol{B}_{k+1}^\top\boldsymbol{x}_k\|_2^2. \tag{6}$$

This Dirichlet energy returns the graph Dirichlet energy when $k = 0$. In this case, $D(\boldsymbol{x}_0) = \|\boldsymbol{B}_1^\top\boldsymbol{x}_0\|_2^2 = \sum_i \sum_j \|x_{0,i} - x_{0,j}\|^2$ is the $\ell_2$-norm of the *gradient* of the node signal $\boldsymbol{x}_0$. For edge flow $\boldsymbol{x}_1$, $D(\boldsymbol{x}_1)$ consists of two parts, $\|\boldsymbol{B}_1\boldsymbol{x}_1\|_2^2$ and $\|\boldsymbol{B}_2^\top\boldsymbol{x}_1\|_2^2$, which measure the total divergence and curl of $\boldsymbol{x}_1$, respectively, i.e., the edge flow variations w.r.t. nodes and triangles. In the general case, $D_\mathrm{d}(\boldsymbol{x}_k)$ and $D_\mathrm{u}(\boldsymbol{x}_k)$ measure the lower and upper $k$-simplicial signal variations w.r.t. the faces and cofaces, respectively. A harmonic $k$-simplicial signal $\boldsymbol{x}_k$ has zero Dirichlet energy, e.g., a constant node signal and a div- and curl-free edge flow.

**Simplicial shifting as Hodge Laplacian smoothing**  Bunch et al. (2020); Yang et al. (2022c) considered $\boldsymbol{H}_k$ to be a weighted variant of $\boldsymbol{I} - \boldsymbol{L}_k$, generalizing the graph convolutional network (GCN) (Kipf & Welling, 2017). This is necessarily a Hodge Laplacian smoothing as in Schaub et al. (2021)—given an initial signal $\boldsymbol{x}_k^0$, we consider the Dirichlet energy minimization:

$$\min_{\boldsymbol{x}_k} \|\boldsymbol{B}_k \boldsymbol{x}_k\|_2^2 + \gamma\|\boldsymbol{B}_{k+1}^\top \boldsymbol{x}_k\|_2^2, \ \ \gamma > 0,$$
$$\text{gradient descent: } \boldsymbol{x}_{k,\text{gd}}^{l+1} = (\boldsymbol{I} - \eta\boldsymbol{L}_{k,\text{d}} - \eta\gamma\boldsymbol{L}_{k,\text{u}})\boldsymbol{x}_k^l \tag{7}$$

with step size $\eta > 0$. The simplicial shifting $\boldsymbol{x}_k^{l+1} = w_0(\boldsymbol{I} - \boldsymbol{L}_k)\boldsymbol{x}_k^l$ is a gradient descent step with $\eta = \gamma = 1$ and weighted by $w_0$. A minimizer of Eq. (7) with $\gamma = 1$ is in fact in the harmonic space $\ker(\boldsymbol{L}_k)$. Thus, a neural network composed of simplicial shifting layers may generate an output with an exponentially decreasing Dirichlet energy as it deepens, formulated by the following proposition. We refer to this as *simplicial oversmoothing*, a notion that generalizes the oversmoothing of a GCN and its variants (Nt & Maehara, 2019; Cai & Wang, 2020; Rusch et al., 2023).

**Proposition 5.** *If* $w_0^2\|\boldsymbol{I} - \boldsymbol{L}_k\|_2^2 < 1$, $D(\boldsymbol{x}_k^{l+1})$ *in a neural network of simplicial shifting layers exponentially converges to zero.*

However, when uncoupling the lower and upper parts of $\boldsymbol{L}_k$ in this shifting, associated to the case $\gamma \neq 1$, the decrease of $D(\boldsymbol{x}_k)$ can slow down or cease because the objective function in Eq. (7) instead looks for a solution primarily in either $\ker(\boldsymbol{B}_k)$ (for $\gamma \ll 1$) or $\ker(\boldsymbol{B}_{k+1}^\top)$ (for $\gamma \gg 1$), not necessarily in $\ker(\boldsymbol{L}_k)$, as we shall corroborate in Section 6.

**Inter-simplicial couplings as sources**  Given some nontrivial $\boldsymbol{x}_{k-1}$ and $\boldsymbol{x}_{k+1}$, we consider the optimization

$$\min_{\boldsymbol{x}_k} \|\boldsymbol{B}_k \boldsymbol{x}_k - \boldsymbol{x}_{k-1}\|_2^2 + \|\boldsymbol{B}_{k+1}^\top \boldsymbol{x}_k - \boldsymbol{x}_{k+1}\|_2^2,$$
$$\text{gradient descent: } \boldsymbol{x}_{k,\text{gd}}^{l+1} = (\boldsymbol{I} - \eta\boldsymbol{L}_k)\boldsymbol{x}_k^l + \eta(\boldsymbol{x}_{k,\text{d}} + \boldsymbol{x}_{k,\text{u}}) \tag{8}$$

with step size $\eta > 0$. It resembles the convolutional layer, $\boldsymbol{x}_k^{l+1} = w_0(\boldsymbol{I} - \boldsymbol{L}_k)\boldsymbol{x}_k^l + w_1\boldsymbol{x}_{k,\text{d}} + w_2\boldsymbol{x}_{k,\text{u}}$ with some learnable weights, in Bunch et al. (2020); Yang et al. (2022c).

**Proposition 6.** *We have the following bounds for the Dirichlet energy:* $\|\boldsymbol{x}_{k-1}\|_2^2 + \|x_{k+1}\|_2^2 \leq D(\boldsymbol{x}_k^{l+1}) \leq w_0^2\|\boldsymbol{I} - \boldsymbol{L}_k\|_2^2 D(\boldsymbol{x}_k^l) + w_1^2\lambda_{\max}(\boldsymbol{L}_{k,\text{d}})\|\boldsymbol{x}_{k,\text{d}}\|_2^2 + w_2^2\lambda_{\max}(\boldsymbol{L}_{k,\text{u}})\|\boldsymbol{x}_{k,\text{u}}\|_2^2.$

The signal projections from the lower and upper simplices act as energy sources for $\boldsymbol{x}_k^l$, and also the objective function in Eq. (8) looks for an $\boldsymbol{x}_k$ in the image spaces of $\boldsymbol{B}_{k+1}$ and $\boldsymbol{B}_k^\top$, instead of $\ker(\boldsymbol{L}_k)$. This ensues that $D(\boldsymbol{x}_k^{l+1})$ may not converge to zero, but rather to a nontrivial value $\|\boldsymbol{x}_{k-1}\|_2^2 + \|x_{k+1}\|_2^2$. Thus, inter-simplicial couplings pay a role in mitigating the oversmoothing.

Here we showed that simply generalzing GCNs to simplices will inherit its oversmoothing risks. However, by uncoupling the lower and upper Laplacians and accounting for the inter-simplicial couplings we could mitigate this issue. This can also be explained by means of a diffusion process on SCs (Ziegler et al., 2022), which we discuss in Appendix B.4.

## 4  From convolutional to Hodge-aware

In this section, we first introduce the *Hodge-invariant operator*, which is an operator such that the three Hodge subspaces are invariant under it. Then, we show that the SCF is such an operator and SCCNN, guided by the three principles (P1-P3), performs *Hodge-invariant learning*, allowing for rational and effective learning on SCs while remaining expressive. Throughout the exposition, we rely on the simplicial spectral theory (Barbarossa & Sardellitti, 2020; Yang et al., 2021; 2022b), which also allows us to characterize the expressive power of SCCNNs. We refer to the detailed derivations and proofs in Appendix E.

**Definition 7** (Invariant subspace). Let $V$ be a finite-dimensional vector space over $\mathbb{R}$ with $\dim(V) \geq 1$, and let $T : V \rightarrow V$ be an operator in V. A subspace $U \subset V$ is an *invariant subspace* under $T$ if $Tu \in U$ for all $u \in U$, i.e., the image of every vector in $U$ under $T$ remains within $U$. We denote this as $T|_U : U \rightarrow U$ where $T|_U$ is the restriction of $T$ on $U$.

Given the notion of invariant subspace, we then define the Hodge-invariant operators.

**Definition 8** (Hodge-invariant operator)**.** Let $\square \in \{\mathrm{im}(\boldsymbol{B}_k^\top), \mathrm{im}(\boldsymbol{B}_{k+1}), \ker(\boldsymbol{L}_k)\}$ be any Hodge subspace of $\mathbb{R}^{n_k}$. A linear transformation $F : \mathbb{R}^{n_k} \to \mathbb{R}^{n_k}$ is a *Hodge-invariant operator* if for all $\boldsymbol{x}_k \in \square$ it holds that $F(\boldsymbol{x}_k) \in \square$. That is, any simplicial signal in a certain Hodge subspace remains in that subspace under $F$.

**Definition 9** ((Barbarossa & Sardelitti, 2020))**.** The *simplicial Fourier transform* (SFT) of $\boldsymbol{x}_k$ is $\tilde{\boldsymbol{x}}_k = \boldsymbol{U}_k^\top \boldsymbol{x}_k$ where the eigenbasis $\boldsymbol{U}_k$ of $\boldsymbol{L}_k$ acts as the simplicial Fourier basis and the eigenvalues in $\boldsymbol{\Lambda}_k = \mathrm{diag}(\boldsymbol{\lambda}_k)$ are *simplicial frequencies*.

**Proposition 10** (Yang et al. (2022b))**.** *The SFT basis can be found as* $\boldsymbol{U}_k = [\boldsymbol{U}_{k,\mathrm{H}}\ \boldsymbol{U}_{k,\mathrm{G}}\ \boldsymbol{U}_{k,\mathrm{C}}]$ *where*

- $\boldsymbol{U}_{k,\mathrm{H}}$ *is the eigenvector matrix associated to the zero eigenvalues* $\boldsymbol{\Lambda}_{k,\mathrm{H}} = \mathrm{diag}(\boldsymbol{\lambda}_{k,\mathrm{H}})$, *named as harmonic frequencies,*
- $\boldsymbol{U}_{k,\mathrm{G}}$ *is associated to the nonzero eigenvalues in* $\boldsymbol{\Lambda}_{k,\mathrm{G}} = \mathrm{diag}(\boldsymbol{\lambda}_{k,\mathrm{G}})$ *of* $\boldsymbol{L}_{k,\mathrm{d}}$, *named as* gradient *frequencies, and*
- $\boldsymbol{U}_{k,\mathrm{C}}$ *is associated to the nonzero eigenvalues in* $\boldsymbol{\Lambda}_{k,\mathrm{C}} = \mathrm{diag}(\boldsymbol{\lambda}_{k,\mathrm{C}})$, *named as* curl *frequencies.*

*Moreover, they span the Hodge subspaces:*

$$\mathrm{span}(\boldsymbol{U}_{k,\mathrm{H}}) = \ker(\boldsymbol{L}_k),\ \ \mathrm{span}(\boldsymbol{U}_{k,\mathrm{G}}) = \mathrm{im}(\boldsymbol{B}_k^\top),\ \ \mathrm{span}(\boldsymbol{U}_{k,\mathrm{C}}) = \mathrm{im}(\boldsymbol{B}_{k+1}) \tag{9}$$

*where* $\mathrm{span}(\bullet)$ *denotes all possible linear combinations of columns of* $\bullet$.

*Remark* 11. The frequency notion in general carries the physical meaning of signal variations. In the simplicial case, gradient frequencies reflect the degree of lower variations $D_{\mathrm{d}}(\boldsymbol{u}_{k,\mathrm{G}})$ of the associated gradient Fourier basis, and curl frequencies reflect the degree of upper variations $D_{\mathrm{u}}(\boldsymbol{u}_{k,\mathrm{C}})$ of the associated curl basis. Harmonic frequencies (zeros) correspond to the basis having zero lower and upper variations. In the edge case, the gradient and curl frequencies, respectively, correspond to the total divergence and total curl, measuring how divergent and rotational the associated basis is (Yang et al., 2022b).

**Proposition 12.** *The SCF* $\boldsymbol{H}_k$ *is a Hodge-invariant operator. That is, for any* $\boldsymbol{x}_k \in \square$, *we have* $\boldsymbol{H}_k\boldsymbol{x}_k \in \square$, *for* $\square \in \{\mathrm{im}(\boldsymbol{B}_k^\top), \mathrm{im}(\boldsymbol{B}_{k+1}), \ker(\boldsymbol{L}_k)\}$. *Moreover, the SCF operation can be implicitly written as*

$$\boldsymbol{H}_k\boldsymbol{x}_k = \boldsymbol{H}_k|_{\mathrm{im}(\boldsymbol{B}_k^\top)}\boldsymbol{x}_{k,\mathrm{G}} + \boldsymbol{H}_k|_{\ker(\boldsymbol{L}_k)}\boldsymbol{x}_{k,\mathrm{H}} + \boldsymbol{H}_k|_{\mathrm{im}(\boldsymbol{B}_{k+1})}\boldsymbol{x}_{k,\mathrm{C}} \tag{10}$$

*where* $\boldsymbol{H}_k|_{\mathrm{im}(\boldsymbol{B}_k^\top)} = \sum_{t=1}^{T_{\mathrm{d}}} w_{k,\mathrm{d},t}\boldsymbol{L}_{k,\mathrm{d}}^t + (w_{k,\mathrm{d},0} + w_{k,\mathrm{u},0})\boldsymbol{I}$ *is the restriction of* $\boldsymbol{H}_k$ *on the gradient space* $\mathrm{im}(\boldsymbol{B}_k^\top)$, $\boldsymbol{H}_k|_{\ker(\boldsymbol{L}_k)} = (w_{k,\mathrm{d},0} + w_{k,\mathrm{u},0})\boldsymbol{I}$ *is the restriction of* $\boldsymbol{H}_k$ *on the harmonic space, and* $\boldsymbol{H}_k|_{\mathrm{im}(\boldsymbol{B}_{k+1})} = \sum_{t=0}^{T_{\mathrm{u}}} w_{k,\mathrm{u},t}\boldsymbol{L}_{k,\mathrm{u}}^t + (w_{k,\mathrm{d},0} + w_{k,\mathrm{u},0})\boldsymbol{I}$ *is the restriction on the curl space.*

Provided with the Hodge-invariance of $\boldsymbol{H}_k$ and the SFT, we can perform a spectral analysis, which is of interest to further understand the SCCNN since simplicial frequencies reflect the variation characteristics of simplicial signals.

### 4.1 Spectral analysis

Consider the SFT $\tilde{\boldsymbol{x}}_k = [\tilde{\boldsymbol{x}}_{k,\mathrm{H}}^\top, \tilde{\boldsymbol{x}}_{k,\mathrm{G}}^\top, \tilde{\boldsymbol{x}}_{k,\mathrm{C}}^\top]^\top$ of $\boldsymbol{x}_k$ where each component is the intensity of $\boldsymbol{x}_k$ at a certain simplicial frequency. We can understand how an SCCNN convolutional layer $\boldsymbol{y}_k = \boldsymbol{H}_{k,\mathrm{d}}\boldsymbol{x}_{k,\mathrm{d}} + \boldsymbol{H}_k\boldsymbol{x}_k + \boldsymbol{H}_{k,\mathrm{u}}\boldsymbol{x}_{k,\mathrm{u}}$ regulates/learns from the simplicial signals at different frequencies by performing the SFT

$$\begin{cases} \tilde{\boldsymbol{y}}_{k,\mathrm{H}} = \tilde{\boldsymbol{h}}_{k,\mathrm{H}} \odot \tilde{\boldsymbol{x}}_{k,\mathrm{H}} \\ \tilde{\boldsymbol{y}}_{k,\mathrm{G}} = \tilde{\boldsymbol{h}}_{k,\mathrm{d}} \odot \tilde{\boldsymbol{x}}_{k,\mathrm{d}} + \tilde{\boldsymbol{h}}_{k,\mathrm{G}} \odot \tilde{\boldsymbol{x}}_{k,\mathrm{G}} \\ \tilde{\boldsymbol{y}}_{k,\mathrm{C}} = \tilde{\boldsymbol{h}}_{k,\mathrm{C}} \odot \tilde{\boldsymbol{x}}_{k,\mathrm{C}} + \tilde{\boldsymbol{h}}_{k,\mathrm{u}} \odot \tilde{\boldsymbol{x}}_{k,\mathrm{u}}, \end{cases} \tag{11}$$

with $\odot$ the elementwise multiplication. The $n_k$-dimensional vector $\tilde{\boldsymbol{h}}_k = \mathrm{diag}(\boldsymbol{U}_k^\top \boldsymbol{H}_k \boldsymbol{U}_k) = [\tilde{\boldsymbol{h}}_{k,\mathrm{H}}^\top\ \tilde{\boldsymbol{h}}_{k,\mathrm{G}}^\top\ \tilde{\boldsymbol{h}}_{k,\mathrm{C}}^\top]^\top$ is the frequency response vector of $\boldsymbol{H}_k$ with

$$\tilde{\boldsymbol{h}}_{k,\mathrm{H}} = (w_{k,\mathrm{d},0} + w_{k,\mathrm{u},0})\mathbf{1}, \quad \tilde{\boldsymbol{h}}_{k,\mathrm{G}} = \sum_{t=0}^{T_{\mathrm{d}}} w_{k,\mathrm{d},t}\boldsymbol{\lambda}_{k,\mathrm{G}}^{\odot t} + w_{k,\mathrm{u},0}\mathbf{1}, \quad \tilde{\boldsymbol{h}}_{k,\mathrm{C}} = \sum_{t=0}^{T_{\mathrm{u}}} w_{k,\mathrm{u},t}\boldsymbol{\lambda}_{k,\mathrm{C}}^{\odot t} + w_{k,\mathrm{d},0}\mathbf{1}, \tag{12}$$

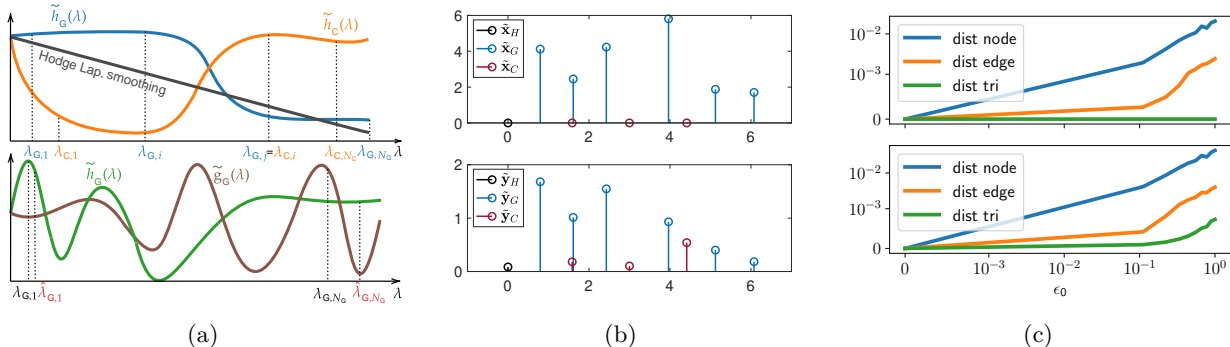

Figure 3: (a) *(top)*: Independent gradient and curl learning responses. *(bottom)*: Stability-selectivity tradeoff of SCFs where $\tilde{h}_{\mathrm{G}}$ has better stability but smaller selectivity than $\tilde{g}_{\mathrm{G}}$. (b) Information spillage of nonlinearity. *(top)*: the SFT of an input with only gradient components. *(bottom)*: the SFT of the output shows that after applying a nonlinearity the output also contains information in non-gradient frequencies. (c) The distance between the perturbed outputs and true when node adjacencies are perturbed. *(top)*: $L = 1$, triangle output remains clean. *(bottom)*: $L = 2$, triangle output is perturbed.

where $\cdot^{\odot t}$ is the elementwise $t$-th power of a vector. Likewise,

$$\tilde{\boldsymbol{h}}_{k,\mathrm{d}} = \sum_{t=0}^{T_{\mathrm{d}}} w'_{k,\mathrm{d},t} \boldsymbol{\lambda}_{k,\mathrm{G}}^{\odot t} + w'_{k,\mathrm{u},0} \mathbf{1}, \text{ and } \tilde{\boldsymbol{h}}_{k,\mathrm{u}} = \sum_{t=0}^{T_{\mathrm{u}}} w'_{k,\mathrm{u},t} \boldsymbol{\lambda}_{k,\mathrm{C}}^{\odot t} + w'_{k,\mathrm{d},0} \mathbf{1} \tag{13}$$

are the frequency response vectors of $\boldsymbol{H}_{k,\mathrm{d}}$ and $\boldsymbol{H}_{k,\mathrm{u}}$. The spectral relation in Eq. (11) shows that the gradient SFT $\tilde{\boldsymbol{x}}_{k,\mathrm{G}}$ is learned by a gradient response $\tilde{\boldsymbol{h}}_{k,\mathrm{G}}$, while the curl SFT $\tilde{\boldsymbol{x}}_{k,\mathrm{C}}$ is learned by a curl response $\tilde{\boldsymbol{h}}_{k,\mathrm{C}}$. The two learnable responses are independent and they only coincide at the trivial harmonic frequency, as shown by the two individual curves in Fig. 3a. Moreover, the lower and upper projections are independently learned by $\tilde{\boldsymbol{h}}_{k,\mathrm{d}}$ and $\tilde{\boldsymbol{h}}_{k,\mathrm{u}}$, respectively.

The elementwise nonlinearity induces an *information spillage* that one type of spectra could be spread over other types. As illustrated in Fig. 3b, the top figure shows the SFT of an input with only gradient components, and the bottom figure plots the SFT of $\sigma(\boldsymbol{y}_k)$, showing that it also contains information in harmonic or curl subspaces. This results from the nonlinearity, since applying a linear SCF leads to an output with only gradient components. In the following, we characterize the expressive power of SCCNN.

**Proposition 13.** *An SCCNN layer with inputs $\boldsymbol{x}_{k,\mathrm{d}}, \boldsymbol{x}_k, \boldsymbol{x}_{k,\mathrm{u}}$ is at most expressive as an MLP $\sigma(\boldsymbol{G}'_{k,\mathrm{d}}\boldsymbol{x}_{k,\mathrm{d}} + \boldsymbol{G}_k \boldsymbol{x}_k + \boldsymbol{G}'_{k,\mathrm{u}}\boldsymbol{x}_{k,\mathrm{u}})$ with $\boldsymbol{G}_k = \boldsymbol{G}_{k,\mathrm{d}} + \boldsymbol{G}_{k,\mathrm{u}}$ where $\boldsymbol{G}_{k,\mathrm{d}}$ and $\boldsymbol{G}'_{k,\mathrm{d}}$ are analytical matrix functions of $\boldsymbol{L}_{k,\mathrm{d}}$, while $\boldsymbol{G}_{k,\mathrm{u}}$ and $\boldsymbol{G}'_{k,\mathrm{u}}$ are analytical matrix functions of $\boldsymbol{L}_{k,\mathrm{u}}$. This expressivity can be achieved by setting $T_{\mathrm{d}} = T'_{\mathrm{d}} = n_{k,\mathrm{G}}$ and $T_{\mathrm{u}} = T'_{\mathrm{u}} = n_{k,\mathrm{C}}$ in Eq. (4) with $n_{k,\mathrm{G}}$ the number of distinct gradient frequencies and $n_{k,\mathrm{C}}$ the number of distinct curl frequencies.*

The proof follows from Cayley-Hamilton theorem (Horn & Johnson, 2012). This expressive power can be better understood from the spectral perspective — The gradient SFT $\tilde{\boldsymbol{x}}_{k,\mathrm{G}}$ can be learned as expressive as by an analytical vector-valued function $\tilde{\boldsymbol{g}}_{k,\mathrm{G}}$, which collects the eigenvalues of $\boldsymbol{G}_{k,\mathrm{d}}$ at gradient frequencies. The curl SFT $\tilde{\boldsymbol{x}}_{k,\mathrm{C}}$ can be learned as expressive as by another analytical vector-valued function $\tilde{\boldsymbol{g}}_{k,\mathrm{C}}$, which collects the eigenvalues of $\boldsymbol{G}_{k,\mathrm{u}}$ at curl frequencies. These two functions need only to coincide at the harmonic frequency. In addition, the SFTs of lower and upper projections can be learned as expressive as by two independent analytical vector-valued functions as well.

## 4.2 Hodge-aware learning

Given the expressive power in Proposition 13 and the spectral relation in Eq. (11), we show that SCCNN performs a Hodge-aware learning in the following sense, which comes with advantages over the existing approaches.

**Theorem 14.** *An SCCNN is Hodge-aware: 1) The SCF $\boldsymbol{H}_k$ is a Hodge-invariant learning operator. Specifically, three Hodge subspaces are invariant under $\boldsymbol{H}_k$; 2) The lower SCF $\boldsymbol{H}_{k,\mathrm{d}}$ and upper SCF $\boldsymbol{H}_{k,\mathrm{u}}$ are,*

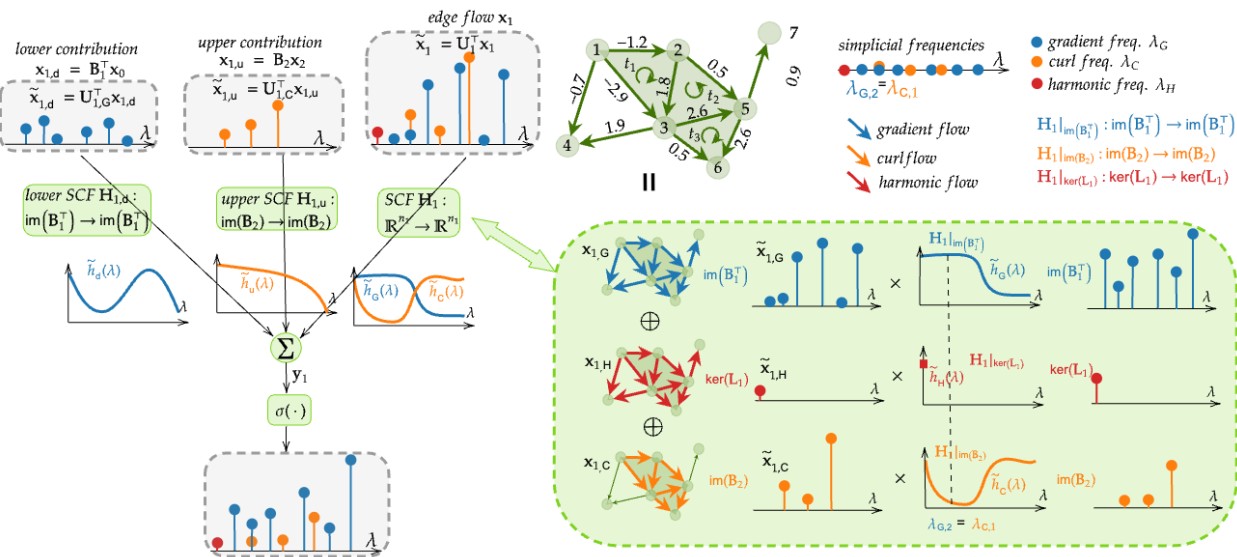

Figure 4: An illustration of the Hodge-aware learning of an SCCNN. We show that how an edge flow $\boldsymbol{x}_1$, together with the lower and upper projections $\boldsymbol{x}_{1,\mathrm{d}}$ and $\boldsymbol{x}_{1,\mathrm{u}}$, are transformed by an SCCNN in the spectral domain. The *implicit* operation $\boldsymbol{H}_1 \boldsymbol{x}_1$ (in the dashed box on the right) reflects the Hodge-aware learning: 1) $\boldsymbol{H}_1$ is Hodge invariant: each component is learned within their own subspace, and $\boldsymbol{H}_1$ does not *mix up* the three subspaces; 2) The learning in the gradient and curl subspaces are independent where features at shared frequencies $\lambda_{G,2}$ and $\lambda_{C,1}$ can be separately learned; and 3) The learning operators are expressive in the sense that the spectral responses are as expressive as any analytical functions in the gradient and curl frequencies.

*respectively, gradient- and curl-invariant learning operators; 3) The learnings in the gradient and curl spaces are **independent**; And 4) the learnings in the gradient and curl spaces are **expressive** as in* Proposition 13.

This theorem shows that an SCCNN performs an expressive and independent learning in the gradient and curl subspaces from the three inputs while preserving the three subspaces to be invariant w.r.t its learnable SCFs. This allows for the *rational and effective learning* on SCs, as illustrated in Fig. 4, from the two aspects. These three-fold properties of an SCCNN, respectively, come from the convolutional architecture choice, the uncoupling of the lower and upper adjacencies, and the higher-order convolutions in the SCCNN.

On the one hand, Proposition 12 shows that the operation of $\boldsymbol{H}_k$ on the simplicial signal space is equivalent to a summation of its restrictions $\boldsymbol{H}_k|_{\square}$ on three smaller subspaces $\square$. This Hodge-invariant nature of the learnable SCFs substantially shrinks the learning space of an SCCNN and allows for an effective learning. On the other hand, simplicial signals often present implicit or explicit properties that different Hodge subspaces can capture. For example, water flows, traffic flows, electrical currents (Grady & Polimeni, 2010; Jia et al., 2019) follow flow conservation, i.e., being div-free in the gradient space $\ker(\boldsymbol{B}_1)$, while exchange rates can be modelled as curl-free edge flows (Jiang et al., 2011). Owing to the Hodge-invariance of $\boldsymbol{H}_1$ and its independent learning in the nontrivial subspaces, an SCCNN can capture such characteristics of real-world edge flows effectively. When it comes to regression tasks on SCs, an SCCNN can generate outputs respecting these physical laws.

*Remark* 15 (Relation to message passing networks). Message-passing simplicial networks (MPSNs) (Bodnar et al., 2021b) using MLP to aggregate and update are non-Hodge-aware. Their learning functions pursue direct mappings between the much larger signal space $\mathbb{R}^{n_k}$, thus, requiring more training data for accurate learning, as well as a larger computational complexity. Moreover, MPSN does not preserve the Hodge subspaces, i.e., it is not Hodge-invariant. Thus, they might generate outputs with small losses (e.g., mean-squared-errors) in regression tasks, yet not respecting the physical laws being either div- or curl-free properties such as the above simplicial signals. We shall corroborate this in Appendix G.

*Remark* 16 (Relation to other convolutional methods)*.* While most convolutional networks on SCs use Hodge-invariant learning operators, they are not strictly Hodge-aware, resulting in practical limits. For example, Ebli et al. (2020) considered $\boldsymbol{H}_k = \sum_i w_i \boldsymbol{L}_k^i$, which preserves the Hodge subspaces yet does not uncouple the lower and upper parts of $\boldsymbol{L}_k$. This makes it *strictly less expressive* and non-Hodge-aware. Consider two frequencies $\lambda_G = \lambda_C$ which share a common value but correspond to the gradient and curl subspaces, respectively. The simplicial signal components at these two frequencies are always learned in the same fashion, which induces contradicting issues when the underlying component in one subspace should be diminished while the one in the other subspace should be preserved. This underlines the importance of uncoupling the two adjacencies because the lower and upper Laplacians operate in different subspaces. Roddenberry et al. (2021) applied $\boldsymbol{H}_k$ with $T_\text{d} = T_\text{u} = 1$. Spatially, this limits the receptive field of each simplex to its direct neighbors. Spectrally, it leads to a linear learning frequency response. A similar treatment was considered in Bunch et al. (2020); Yang et al. (2022c) which simply generalized the GCN without uncoupling the two adjacencies, and gave a limited low-pass linear spectral response, as shown in Fig. 3a and discussed in Section 3.2.

## 5    How robust are SCCNNs to domain perturbations?

In practice, an SCCNN is often built on a weighted SC to capture the strengths of simplicial adjacencies and incidences. We defer the explicit formulations in Appendix F.1 since it has the same form as Eq. (4) in this case, except for that the Hodge Laplacians are weighted, as well as the incidence matrices. These matrices are often defined following Grady & Polimeni (2010); Horak & Jost (2013); Guglielmi et al. (2023). For example, Bunch et al. (2020); Yang et al. (2022c) considered a particular random walk formulation (Schaub et al., 2020). They can also be learned from data, e.g., via an attention method (Goh et al., 2022; Giusti et al., 2022). For the weighted incidence matrices $\boldsymbol{B}_k^\top, \boldsymbol{B}_{k+1}$, we use operators $\boldsymbol{R}_{k,\text{d}}, \boldsymbol{R}_{k,\text{u}}$ in this section.

To highlight the need for a stability analysis, note that, on the one hand, we may lack the true underlying topologies in SCs as they are often estimated from noisy data; and we may undergo adversarial attacks on the topologies. On the other hand, we want to characterize the stability-selectivity tradeoff of SCCNN, in analogy to the study for CNNs (Bruna & Mallat, 2013; Qiu et al., 2018; Bietti & Mairal, 2017) and GNNs (Gama et al., 2019b; 2020a; Kenlay et al., 2021; Parada-Mayorga et al., 2022).

This motivates us to investigate the stability of SCCNN: *how far are the outputs of an SCCNN before and after perturbations are applied to SCs?* We consider the following relative perturbation model, generalizing the graph perturbation model in Gama et al. (2019b)

**Definition 17** (Relative perturbation)**.** Consider some perturbation matrix of an appropriate dimension. For the weighted Hodge Laplacian $\boldsymbol{L}_{k,\text{d}}$, its relative perturbed version is $\widehat{\boldsymbol{L}}_{k,\text{d}} = \boldsymbol{L}_{k,\text{d}} + \boldsymbol{E}_{k,\text{d}} \boldsymbol{L}_{k,\text{d}} + \boldsymbol{L}_{k,\text{d}} \boldsymbol{E}_{k,\text{d}}$ with perturbation $\boldsymbol{E}_{k,\text{d}}$; likewise for $\widehat{\boldsymbol{L}}_{k,\text{u}}$ by $\boldsymbol{E}_{k,\text{u}}$. For the weighted incidence matrix $\boldsymbol{R}_{k,\text{d}}$, its relative perturbed version is $\widehat{\boldsymbol{R}}_{k,\text{d}} = \boldsymbol{R}_{k,\text{d}} + \boldsymbol{J}_{k,\text{d}} \boldsymbol{R}_{k,\text{d}}$ with perturbation $\boldsymbol{J}_{k,\text{d}}$; likewise for $\widehat{\boldsymbol{R}}_{k,\text{u}}$ by $\boldsymbol{J}_{k,\text{u}}$.

This models the *domain perturbations* on the strengths of adjacent and incident relations, e.g., a large weight is applied when two edges are weakly or not adjacent, or data on a node is projected on an edge not incident to it. Moreover, this quantifies the relative perturbations with respect to the local simplicial topology in the sense that weaker connections in an SC are deviated by perturbations proportionally less than stronger connections. We further define the integral Lipschitz property of spectral filters to measure the variability of spectral response functions of $\boldsymbol{H}_k$.

**Definition 18** (Intergral Lipschitz SCF)**.** An SCF $\boldsymbol{H}_k$ is *integral Lipschitz* with constants $c_{k,\text{d}}, c_{k,\text{u}} \geq 0$ if the derivatives of its spectral response functions $\tilde{h}_{k,\text{G}}(\lambda)$ and $\tilde{h}_{k,\text{C}}(\lambda)$ follow that $|\lambda \tilde{h}'_{k,\text{G}}(\lambda)| \leq c_{k,\text{d}}$ and $|\lambda \tilde{h}'_{k,\text{C}}(\lambda)| \leq c_{k,\text{u}}$.

This property provides a stability-selectivity tradeoff of SCFs independently in the gradient and curl frequencies. A spectral response can have both a good selectivity and stability in small frequencies (a large $|\tilde{h}'_{k,\cdot}|$ for $\lambda \to 0$), while it tends to be flat for having better stability at the cost of selectivity (a small variability for large $\lambda$) in large frequencies, as shown in Fig. 3a. As of the polynomial nature of responses, all SCFs of SCCNN are integral Lipschitz. We also denote the integral Lipschitz constant for the lower SCFs $\boldsymbol{H}_{k,\text{d}}$ by $c_{k,\text{d}}$ and for the upper SCFs $\boldsymbol{H}_{k,\text{u}}$ by $c_{k,\text{u}}$ without loss of generality.

Under the following assumptions, we now characterize the stability bound of SCCNN.

**Assumption 19.** *The perturbations are small such that* $\|\boldsymbol{E}_{k,\mathrm{d}}\|_2 \leq \epsilon_{k,\mathrm{d}}, \|\boldsymbol{J}_{k,\mathrm{d}}\|_2 \leq \varepsilon_{k,\mathrm{d}}, \|\boldsymbol{E}_{k,\mathrm{u}}\|_2 \leq \epsilon_{k,\mathrm{u}}$ *and* $\|\boldsymbol{J}_{k,\mathrm{u}}\|_2 \leq \varepsilon_{k,\mathrm{u}}$, *where* $\|\boldsymbol{A}\|_2 = \max_{|\boldsymbol{x}|_1=1}\|\boldsymbol{Ax}\|_2$ *is the operator norm (spectral radius) of a matrix* $\boldsymbol{A}$.

**Assumption 20.** *The SCFs* $\boldsymbol{H}_k$ *of an SCCNN have a normalized bounded frequency response (for simplicity, though unnecessary), likewise for* $\boldsymbol{H}_{k,\mathrm{d}}$ *and* $\boldsymbol{H}_{k,\mathrm{u}}$.

**Assumption 21.** *The lower and upper projections are finite such that* $\|\boldsymbol{R}_{k,\mathrm{d}}\|_2 \leq r_{k,\mathrm{d}}$ *and* $\|\boldsymbol{R}_{k,\mathrm{u}}\|_2 \leq r_{k,\mathrm{u}}$.

**Assumption 22.** *The nonlinearity* $\sigma(\cdot)$, *e.g.,* $\mathrm{relu}, \mathrm{tanh}, \mathrm{sigmoid}$, *is* $c_\sigma$-*Lipschitz with* $c_\sigma \geq 0$.

**Assumption 23.** *The initial inputs* $\boldsymbol{x}_k^0$, *for all* $k$, *are finite, such that* $\|\boldsymbol{x}_k^0\|_2 \leq [\boldsymbol{\beta}]_k$. *We collect them in* $\boldsymbol{\beta} = [\beta_0, \ldots, \beta_K]^\top$.

**Theorem 24.** *Let* $\boldsymbol{x}_k^L$ *be the* $k$-*simplicial signal output of an* $L$-*layer SCCNN on a weighted SC. Let* $\hat{\boldsymbol{x}}_k^L$ *be the output of the same SCCNN but on a relatively perturbed SC. Define* $\delta_{k,\mathrm{d}} = (\|\boldsymbol{V}_{k,\mathrm{d}} - \boldsymbol{U}_k\| + 1)^2 - 1$ *and* $\delta_{k,\mathrm{u}} = (\|\boldsymbol{V}_{k,\mathrm{u}} - \boldsymbol{U}_k\| + 1)^2 - 1$, *with* $\boldsymbol{V}_{k,\mathrm{d}}$ *and* $\boldsymbol{V}_{k,\mathrm{u}}$ *the eigenvectors of* $\boldsymbol{E}_{k,\mathrm{d}}$ *and* $\boldsymbol{E}_{k,\mathrm{u}}$, *which measure the eigenvector misalignments between the perturbations and Laplacians. Under Assumptions 19 to 23, the Euclidean distance between the two outputs is finite and upper-bounded*

$$\|\hat{\boldsymbol{x}}_k^L - \boldsymbol{x}_k^L\|_2 \leq [\boldsymbol{d}]_k \text{ with } \boldsymbol{d} = c_\sigma^L \sum_{l=1}^{L} \widehat{\boldsymbol{Z}}^{l-1}\boldsymbol{T}\boldsymbol{Z}^{L-l}\boldsymbol{\beta}, \tag{14}$$

*where for* $K = 2$,

$$\boldsymbol{T} = \begin{bmatrix} t_0 & t_{0,\mathrm{u}} & \\ t_{1,\mathrm{d}} & t_1 & t_{1,\mathrm{u}} \\ & t_{2,\mathrm{d}} & t_2 \end{bmatrix}, \boldsymbol{Z} = \begin{bmatrix} 1 & r_{0,\mathrm{u}} & \\ r_{1,\mathrm{d}} & 1 & r_{1,\mathrm{u}} \\ & r_{2,\mathrm{d}} & 1 \end{bmatrix}, \widehat{\boldsymbol{Z}} = \begin{bmatrix} 1 & \hat{r}_{0,\mathrm{u}} & \\ \hat{r}_{1,\mathrm{d}} & 1 & \hat{r}_{1,\mathrm{u}} \\ & \hat{r}_{2,\mathrm{d}} & 1 \end{bmatrix}, \tag{15}$$

*with* $\hat{r}_{k,\mathrm{d}} = r_{k,\mathrm{d}}(1+\varepsilon_{k,\mathrm{d}})$ *and* $\hat{r}_{k,\mathrm{u}} = r_{k,\mathrm{u}}(1+\varepsilon_{k,\mathrm{u}})$. *Notice that* $\boldsymbol{T}, \boldsymbol{Z}$ *and* $\widehat{\boldsymbol{Z}}$ *are tridiagonal and follow a similar structure for a general* $K$. *The diagonal entries of* $\boldsymbol{T}$ *are* $t_k = c_{k,\mathrm{d}}\Delta_{k,\mathrm{d}}\epsilon_{k,\mathrm{d}} + c_{k,\mathrm{u}}\Delta_{k,\mathrm{u}}\epsilon_{k,\mathrm{u}}$. *The off-diagonal entries are* $t_{k,\mathrm{d}} = r_{k,\mathrm{d}}\varepsilon_{k,\mathrm{d}} + c_{k,\mathrm{d}}\Delta_{k,\mathrm{d}}\epsilon_{k,\mathrm{d}}r_{k,\mathrm{d}}$ *and* $t_{k,\mathrm{u}} = r_{k,\mathrm{u}}\varepsilon_{k,\mathrm{u}} + c_{k,\mathrm{u}}\Delta_{k,\mathrm{u}}\epsilon_{k,\mathrm{u}}r_{k,\mathrm{u}}$, *where* $\Delta_{k,\mathrm{d}} = 2(1+\delta_{k,\mathrm{d}})\sqrt{n_k}$ *and* $\Delta_{k,\mathrm{u}} = 2(1 + \delta_{k,\mathrm{u}})\sqrt{n_k}$.

We refer to Appendix F.2 for a two-step proof. This result bounds the outputs of an SCCNN on all simplicial levels, showing that they are stable to small perturbations on the simplicial adjacencies and incidences. Specifically, we make two observations from the complex expression. First, the stability bound depends on i) the degree of perturbations including their magnitudes $\epsilon_{k,\cdot}$ and $\varepsilon_{k,\cdot}$, and the eigenspace misalignment $\delta_{k,\cdot}$; ii) the number of simplices $n_k$; iii) the integral Lipschitz properties $c_{k,\cdot}$ of SCFs; and, iv) the degree of projections $r_{k,\cdot}$. Second, the stability of the $k$-output depends on the factors related to not only $k$-simplices, but also simplices of adjacent orders due to inter-simplicial couplings. For example, when $L = 1$, the node output bound $d_0$ is affected by factors in the node space, as well as the edge space factored by the projection degree. As the layer deepens, this mutual dependence expands further. When $L = 2$, the factors in the triangle space also affect the stability of the node output $d_0$, as we observe in Fig. 3c.

More importantly, this stability bound provides intuitive practical implications for convolutional learning on SCs. While inter-simplicial couplings may be beneficial, SCCNN becomes less stable as the number of layers increases due to the mutual dependence between outputs on different simplicial levels. Thus, to maintain the expressive power, we expect to use higher-order SCFs in exchange for shallow layers. This yet does not harm the stability in the following two aspects:

- First, high-frequency components can be spread over the low frequencies due to the information spillage of nonlinearity [cf. Fig. 3b] where the spectral responses are more selective and have better stability. If the signal has large high gradient frequency components, we need the SCCNN to be selective in high gradient frequencies. However, to guarantee the stability, the frequency response should be smooth (less selective) in these frequencies, as illustrated by $\tilde{h}_\mathrm{G}$ in Fig. 3a *(bottom)*. This selectivity-stability tradeoff can be mitigated by the nonlinearity — the information in high gradient frequencies could spill over in lower frequencies, where the spectral responses are more selective and have better discriminating ability.

- Second, higher-order SCFs are easier to be learned with smaller integral Lipschitz constants than lower-order ones due to the increased degree of freedom, thus, leading to an increased stability. This can be easily seen by comparing one-order and two-order cases. We experimentally investigate this in Section 6.4. Moreover, we introduce the following regularizer to the loss function during training so to promote the integral Lipschitz property.

$$r_{\text{IL}} = \|\lambda_{k,\text{G}}\tilde{h}'_{k,\text{G}}(\lambda_{k,\text{G}})\| + \|\lambda_{k,\text{C}}\tilde{h}'_{k,\text{C}}(\lambda_{k,\text{C}})\| = \left\|\sum_{t=0}^{T_{\text{d}}} t w_{k,\text{d},t}\lambda_{k,\text{G}}^t\right\| + \left\|\sum_{t=0}^{T_{\text{u}}} t w_{k,\text{u},t}\lambda_{k,\text{C}}^t\right\|, \quad (16)$$

for $\lambda_{k,\text{G}} \in \{\lambda_{k,\text{G},i}\}_{i=1}^{n_{k,\text{G}}}$ and $\lambda_{k,\text{C}} \in \{\lambda_{k,\text{C},i}\}_{i=1}^{n_{k,\text{C}}}$, which are the gradient and curl frequencies. To avoid computing the eigendecomposition of the Hodge Laplacian, we can approximate the true frequencies by sampling certain number of points in the frequency band $(0, \lambda_{k,\text{G},\text{m}}]$ and $(0, \lambda_{k,\text{C},\text{m}}]$ where the maximal gradient and curl frequencies can be computed by efficient algorithms, e.g., power iteration.

## 6 Experiments

The goal of this section is to answer the following four research questions with experiments on various simplicial-level regression and classification tasks:

**RQ 1** What are the effects of the three principles of SCCNN, i.e., uncoupling the lower and upper parts of Hodge Laplacians (P1), the inter-simplicial couplings (P2), and higher-order convolutions (P3)?

**RQ 2** How do the uncoupling of the lower and upper parts of Hodge Laplacians and the inter-simplicial couplings affect the simplicial oversmoothing?

**RQ 3** How does the Hodge-aware property of SCCNN play a role in different tasks on SCs, compared to non-Hodge-aware methods?

**RQ 4** How do different factors affect the stability of SCCNN, and how can we maintain the stability while keeping the expressive power?

For comparison, we consider the following learning methods on single-level simplices:

- simplicial neural network (SNN) (Ebli et al., 2020), which does not respect P1 and P2 and is non-Hodge-aware;
- principled simplicial neural network (PSNN) (Roddenberry et al., 2021), which does not respect P2 and P3 and is non-Hodge-aware;
- simplicial convolutional neural networks (SCNN)[1] (Yang et al., 2022a), which does not respect P2 but is Hodge-aware;

and the following learning methods on simplicial complexes:

- Bunch (Bunch et al., 2020), which does not respect P1 and P2 and is non-Hodge-aware;
- MPSN (Bodnar et al., 2021b), which is based on message-passing and not Hodge-aware.

We also considered the MLP and standard GNN (Defferrard et al., 2016) as baselines to highlight the effect of SC topology on simplicial-level tasks. We refer to Appendix C for the detailed comparisons between these methods and SCCNN, as well as to Appendix G for the full experimental details.

### 6.1 Foreign currency exchange (RQs 1, 3)

In forex problems, to build a fair market, the *arbitray-free* condition implies that for any currencies $i, j, k$, it follows that $r^{i/j}r^{j/k} = r^{i/k}$ where $r^{i/j}$ is the exchange rate between $i$ and $j$. That is, the exchange path

---

[1]Note that the difference between SCCNN and SCNN lies in that the latter does not include the inter-layer projections, as detailed in Appendix C thus, we refer to our method, simplicial complex CNN.

Table 1: Forex results (nmse|total arbitrage, ↓).

| Methods | Random Noise | Curl Noise | Interpolation |
|---|---|---|---|
| Input | $0.119_{\pm 0.004}\|29.19_{\pm 0.874}$ | $0.552_{\pm 0.027}\|122.4_{\pm 5.90}$ | $0.717_{\pm .030}\|106.4_{\pm 0.902}$ |
| Baseline ($\ell_2$ regularizer) | $0.036_{\pm 0.005}\|2.29_{\pm 0.079}$ | $0.050_{\pm 0.002}\|11.12_{\pm 0.537}$ | $0.534_{\pm 0.043}\|9.67_{\pm 0.082}$ |
| SNN | $0.110_{\pm 0.005}\|23.24_{\pm 1.03}$ | $0.446_{\pm 0.017}\|86.95_{\pm 2.20}$ | $0.702_{\pm 0.033}\|104.74_{\pm 1.04}$ |
| PSNN | $0.008_{\pm 0.001}\|0.984_{\pm 0.170}$ | $\mathbf{0.000_{\pm 0.000}}\|\mathbf{0.000_{\pm 0.000}}$ | $0.009_{\pm 0.001}\|1.13_{\pm 0.329}$ |
| MPSN | $0.039_{\pm 0.004}\|7.74_{\pm 0.88}$ | $0.076_{\pm 0.012}\|14.92_{\pm 2.49}$ | $0.117_{\pm 0.063}\|23.15_{\pm 11.7}$ |
| SCCNN, id | $0.027_{\pm 0.005}\|0.000_{\pm 0.000}$ | $\mathbf{0.000_{\pm 0.000}}\|\mathbf{0.000_{\pm 0.000}}$ | $0.265_{\pm 0.036}\|0.000_{\pm 0.000}$ |
| SCCNN, tanh | $\mathbf{0.002_{\pm 0.000}}\|\mathbf{0.325_{\pm 0.082}}$ | $0.000_{\pm 0.000}\|0.003_{\pm 0.003}$ | $\mathbf{0.003_{\pm 0.002}}\|\mathbf{0.279_{\pm 0.151}}$ |

$i \to j \to k$ provides no profit or loss over a direct exchange $i \to k$. Following Jiang et al. (2011), we model exchange rates as edge flows in an SC of order two, specifically, via $[\boldsymbol{x}_1]_{[i,j]} = \log(r^{i/j})$. This conveniently translates the arbitrage-free condition into $\boldsymbol{x}_1$ being curl-free, i.e., $[\boldsymbol{x}_1]_{[i,j]} + [\boldsymbol{x}_1]_{[j,k]} - [\boldsymbol{x}_1]_{[i,k]} = 0$ in any triangle $[i, j, k]$. We consider a real-world forex market at three timestamps, which contains certain degree of arbitrage (Jia et al., 2019; Yang et al., 2024). We focus on recovering a fair market in two scenarios, first, denoising from noisy exchange rates where random noise and noise only in the curl space modelling random arbitrage ("curl noise") are added, and second, *interpolation*, when only 50% of the total rates are observed. To evaluate the performance, we measure the normalized mean squared error (nmse) and total arbitrage (total curl), both equally important for achieving a fair market.

From Table 1, we make the following observations on the impacts of P1 and P3, as well as the Hodge-awareness.

1) MPSN performs poorly at this task: although it reduces the nmse, it outputs unfair rates with large arbitrage, against the forex principle, because it is not Hodge-aware and unable to capture the arbitrage-free property with small amount of data (cf. Remark 15).

2) SNN performs poorly as well: as discussed in Remark 16, it restricts the gradient and curl spaces to be always learned in the same fashion and makes it impossible to perform disjoint learning in two subspaces. However, since there are eigenvalues which share a common value but live in different subspaces in this SC, it requires preserving the gradient component while removing the curl one here.

3) PSNN can reconstruct relatively fair forex rates with small nmse. The reconstruction from curl noise is perfect, while in the other two cases, the nmse and arbitrage are three times larger than the proposed SCCNN due to the limited expressivity of linear learning responses.

4) SCCNN performs the best in both reducing the total error and the total arbitrage, ultimately, corroborating the impact of performing Hodge-aware learning.

We notice that with an identity activation function ($\sigma = \text{id}$), the arbitrage-free rule is fully learned by an SCCNN. However, it has relatively large errors in the random noise and interpolation cases due to its limited linear expressive power. With a nonlinearity $\sigma = \tanh$, an SCCNN can tackle these more challenging cases, finding a good compromise between overall errors and data characteristics.

## 6.2 Simplicial oversmoothing analysis (RQ 2)

We use simplicial shifting layers (i.e., Eq. (7) composed with $\sigma = \tanh$) to illustrate the evolution of Dirichlet energies of the outputs on nodes, edges and triangles in an SC of order two with respect to the number of layers. The corresponding inputs are randomly sampled from a uniform distribution $\mathcal{U}([-5, 5])$. Fig. 5 (the dashed lines) shows that simply generalizing the GCN on SCs as in Bunch method could lead to oversmoothing on simplices of all orders. This aligns with our theoretical results in Section 3.2. However, uncoupling the lower and upper parts of $\boldsymbol{L}_1$ (e.g., by setting $\gamma = 2$ in Eq. (7)) could mitigate

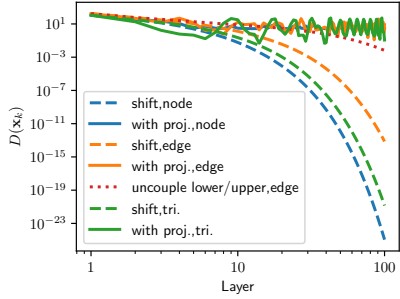

Figure 5: Simplicial oversmoothing.

Table 2: Simplex prediction (AUC, ↑).

|  | MLP | GNN | SNN | PSNN | SCNN | Bunch | MPSN | SCCNN |
|---|---|---|---|---|---|---|---|---|
| 2-simplex | 68.5±1.6 | 93.9±1.0 | 92.0±1.8 | 95.6±1.3 | 96.5±1.5 | **98.3±0.5** | **98.1±0.5** | **98.7±0.5** |
| 3-simplex | 69.0±2.2 | 96.6±0.5 | 95.1±1.2 | 98.1±0.5 | 98.3±0.4 | **98.5±0.5** | **99.2±0.3** | **99.4±0.3** |

Table 3: Ablation study on different components in an SCCNN: the best results with the corresponding hyperparameters, and the results under the same hyperparameters.

| Missing component | 2-simplex | Hyper Params. | 2-simplex | Hyper Params. |
|---|---|---|---|---|
| — | 98.7±0.5 | $L = 2, T = 2$ | 98.7±0.5 | $L = 2, T = 2$ |
| Edge-to-Node | 93.9±1.0 | $L = 5, T = 2$ | 89.8±1.9 | $L = 2, T = 2$ |
| Node-to-Node | 98.7±0.4 | $L = 4, T = 2$ | 96.1±0.9 | $L = 2, T = 2$ |
| Edge-to-Edge | 98.5±1.0 | $L = 3, T = 2$ | 97.4±0.8 | $L = 2, T = 2$ |
| Node-to-Edge | 98.8±0.3 | $L = 4, T = 2$ | 98.5±0.5 | $L = 2, T = 2$ |
| Node input | 98.2±0.5 | $L = 2, T = 4$ | 98.1±0.7 | $L = 2, T = 2$ |
| Edge input | 98.1±0.4 | $L = 2, T = 3$ | 97.2±0.9 | $L = 2, T = 2$ |

the oversmoothing on edges, as shown by the dotted line. Lastly, when we account for the inter-simplicial coupling, as shown by the solid lines (where we applied Eq. (8)), it could almost prevent the oversmoothing, since it provides energy sources. We refer to Appendix G.1 for more results.

### 6.3 Simplex prediction (RQs 1, 3-4)

We consider the prediction task of 2- and 3-simplices which extends the link (1-simplex) prediction in graphs. Our approach is to first learn the representations of lower-order simplices and then use an MLP with their concatenation as inputs to identify if a simplex is closed or open, which generalizes the link prediction method of Zhang & Chen (2018). Considering a coauthorship dataset (Ammar et al., 2018), we built an SC following Ebli et al. (2020) where nodes represent authors and $(k-1)$-simplices thus represent the collaborations of $k$-authors. The input simplicial signals are the numbers of citations, e.g., $\boldsymbol{x}_1$ and $\boldsymbol{x}_2$ are those of dyadic and triadic collaborations. Thus, 2-simplex (3-simplex) prediction amounts to predicting triadic (tetradic) collaborations. We evaluate the AUC (area under the curve) performance.

From Table 2, we make three observations on the effect of the three key principles. 1) SCCNN, MPSN and Bunch methods outperform the ones without inter-simplicial couplings. This highlights that accounting for contributions from faces and cofaces increases the representation power of the network. 2) SCNN performs better than an SNN, which shows that uncoupling the lower and upper parts in $\boldsymbol{L}_k$ improves the representation learning. 3) SCCNN performs better than Bunch (similarly, SCNN better than PSNN), showing that higher-order convolution further improves predictions. 4) While MPSN performs similar to SCCNN, it has three times more parameters than an SCCNN (Appendix G.3.6) under the settings of the best results.

**Ablation study** We then perform an ablation study to investigate the roles of different components in an SCCNN. As reported in Table 3, we remove certain simplicial relations in the SCCNN and evaluate the prediction performance. Without the edge-to-node incidence, when inputting the node features to the MLP predictor, it is equivalent to a GNN, which has a poor performance. When removing other adjacencies or incidences, the best performance remains similar but with an increased model complexity (more layers required). This however is not preferred, because the stability decreases as the architecture deepens and the model gets influenced by factors in other simplicial spaces, as discussed in Section 5 and shown in Fig. 3c. When keeping the hyperparameters fixed, the performance decreases more severely. We also consider the case with limited input where the input on nodes or on edges is missing. The best performance of an SCCNN only slightly drops with an increase of the convolution order.

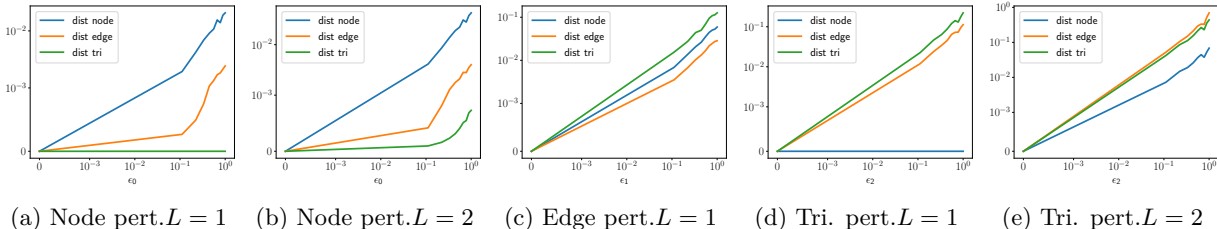

(a) Node pert.$L = 1$    (b) Node pert.$L = 2$    (c) Edge pert.$L = 1$    (d) Tri. pert.$L = 1$    (e) Tri. pert.$L = 2$

Figure 7: The relative difference of SCCNN outputs on simplices of different orders when different levels of perturbations are applied to only nodes, edges and triangles and the number of layers varies.

### 6.3.1 Stability analysis (RQ 4)

**Stability bounds** To investigate the stability bound in Eq. (14), we add perturbations to relatively shift the eigenvalues of the Hodge Laplacians and the singular values of the projection matrices by $\epsilon \in [0, 1]$ (cf. Assumption 19). We compare the bound in Eq. (14) to the experimental $\ell_2$ distance on each simplex level. As shown in Fig. 6 where the dashed lines are the theoretical stability bounds whereas the solid ones are the experimental stability bounds, we see the bounds become tighter as perturbation increases.

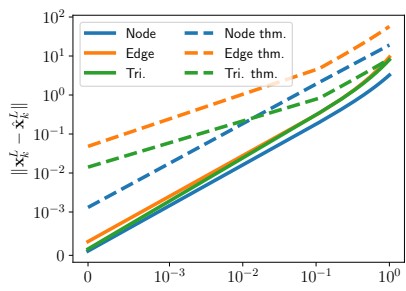

Figure 6: Stability bounds.

**Stability dependence across simplices** For 2-simplex prediction of $K = 2$, we measure the distance between the simplicial outputs of SCCNN with and without perturbations on nodes, edges, and triangles, i.e., $\|\boldsymbol{x}_k^L - \hat{\boldsymbol{x}}_k^L\|/\|\boldsymbol{x}_k^L\|$, for $k = 0, 1, 2$. Fig. 7 shows that overall the stabilities of different simplicial outputs are dependent on each other. Specifically, we see that the triangle output is not influenced by the perturbation on node weights until $L = 2$; likewise, the node output is not influenced by the perturbations on triangle weights when $L = 1$. Also, perturbations on the edge weights will perturbe the outputs on nodes, edges, triangles when $L = 1$. This corroborates our discussions in Section 5.

**Effect of number of simplices.** We see that the same level of perturbations added to different simplices leads to different degrees of instability, owing to the effect of $n_k$ on stability in Eq. (14). Since $n_0 < n_1 < n_2$, the perturbations on node weights cause less instability than those on edge and triangle weights.

**Effect of number of layers.** As the number of layers increases, Fig. 7 also shows that the stability of SCCNN degrades, which corresponds to our analysis of using shallow layers.

### 6.4 Trajectory prediction (RQ 1, 4)

We consider the task of predicting trajectories in a synthetic SC and ocean drifters from Schaub et al. (2020), following Roddenberry et al. (2021). From Table 4, we first observe that the SCCNN and Bunch methods do not always perform better than those without inter-simplicial couplings. This is because zero inputs are applied on nodes and triangles following Roddenberry et al. (2021), which makes inter-simplicial couplings inconsequential. Secondly, an SCCNN performs better than Bunch on average, and SCNN better than PSNN, showing the advantages of higher-order convolutions. Note that the prediction here aims to find the best candidate from the neighborhood of the end node, which depends on the node degree. Since the average node degree of the synthetic SC is 5.24 and that in the ocean drifter data is 4.81, a random guess has around 20% accuracy. The high standard derivations may result from the limited ocean drifter data size.

### 6.4.1 Stability analysis (RQ 4)

Different from Section 6.3.1, we further investigate the stability in terms of the integral Lipschitz properties and convolutional orders. We consider SCNNs (Yang et al., 2022a) with orders $T_{\mathrm{d}} = T_{\mathrm{u}} \in \{1, 3, 5\}$ and train them with regularizations on the integral Lipschitz constants. As shown in Fig. 8, the higher-order case has

Table 4: Trajectory prediction (accuracy, ↑).

| Methods | Synthetic trajectories | Ocean drifters |
|---------|------------------------|----------------|
| SNN     | 65.5±2.4               | 52.5±6.0       |
| PSNN    | 63.1±3.1               | 49.0±8.0       |
| SCNN    | 67.7±1.7               | 53.0±7.8       |
| Bunch   | 62.3±4.0               | 46.0±6.2       |
| SCCNN   | 65.2±4.1               | 54.5±7.9       |

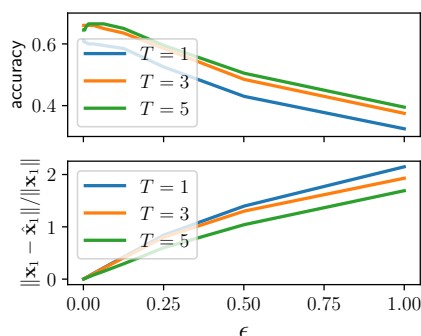

Figure 8: Stability and accuracy of SCCNN versus convolutional orders.

better stability (smaller $\ell_2$ distance between the outputs without and with perturbations) and consistent better accuracy, compared to the lower-order case. This is because the additional flexibility in the higher-order case allows the filters to have better integral Lipschitz properties, thus, better stability, while maintaining the accuracy. Meanwhile, we also study the effect of the regularizer in Eq. (16) on improving the stability while not losing the performance. We refer to Appendix G.4.4 for the detailed experimental analysis where we numerically measure the improved integral Lipschitz properties with this regularizer.

## 7 Related Works, Discussions and Conclusion

**Related works** Our work is mainly related to learning methods on SCs. Roddenberry & Segarra (2019) first used $\boldsymbol{L}_{1,\mathrm{d}}$ to build neural networks on edges in a graph setting without the upper edge adjacency. Ebli et al. (2020) then generalized convolutional GNNs (Kipf & Welling, 2017; Defferrard et al., 2016) to simplices by using the Hodge Laplacian. Roddenberry et al. (2021); Yang et al. (2022a) instead uncoupled the lower and upper Laplacians to perform one- and multi-order convolutions, to which Goh et al. (2022); Giusti et al. (2022); Lee et al. (2022) added attention schemes. Keros et al. (2022) considered a variant of Roddenberry et al. (2021) to identify topological "holes" and Chen et al. (2022b) combined shifting on nodes and edges for link prediction. These works learned within a simplicial level and did not consider the incidence relations (inter-simplicial couplings) in SCs, which was included by Bunch et al. (2020); Yang et al. (2022c). These works considered convolutional-type methods, which can be subsumed by SCCNNs. Meanwhile, Bodnar et al. (2021b); Hajij et al. (2021) generalized the message passing on graphs (Xu et al., 2018a) to SCs, relying on both adjacencies and incidences. Most of these works focused on extending GNNs to SCs by varying the information propagation on SCs with limited theoretical insights into their components. Among them, Roddenberry et al. (2021) discussed the equivariance of PSNN to permutation and orientation, which SCCNNs admit as well. Bodnar et al. (2021b) studied the messgae-passing on SCs in terms of WL test of SCs built by completing cliques in a graph. The more closely related work is Yang et al. (2022a), which gave only a spectral formulation based on SCFs but not SCCNNs. We refer to Papamarkou et al. (2024); Besta et al. (2024) for an overview of the current progress on learning on SCs.

**Discussions** In our opinion, the advantage of SCs is not only about them being able to model higher-order network structures, but also support simplicial data, which can be both human-generated data like coauthorship, and physical data like flow-type data. This is why we approached the analysis from the perspectives of both simplicial structures and the simplicial data, i.e., the Hodge theory and spectral simplicial theory (Hodge, 1989; Lim, 2020; Yang et al., 2021; Barbarossa & Sardellitti, 2020; Steenbergen, 2013; Yang et al., 2022b; Govek et al., 2018). We provided insights into why the three principles (P1-P3) are needed and how they can guide the effective and rational learning from simplicial data. As we have practically found, SCCNNs perform well in applications where data exhibits properties characterized by the Hodge decomposition due to the Hodge-awareness, while non-Hodge-aware learners fail at giving rational results. In cases where data does not possess such properties, SCCNNs have better or comparable performance than the ones which violate or do not respect the three principles.

Concurrently, there are works on more general cell complexes, e.g., (Hajij et al., 2020; 2022; Sardellitti et al., 2021; Roddenberry et al., 2022; Bodnar et al., 2021a), where 2-cells inlcude not only triangles, but also general polygon faces. We focus on SCs because a regular cell complex can be subdivided into an SC (Lundell et al., 1969; Grady & Polimeni, 2010) to which the analysis in this paper applies, or we can generalize our analysis by allowing $B_2$ to include 2-cells. This is however informal and does not exploit the power of cell complexes, which relies on cellular sheaves, as studied in (Hansen & Ghrist, 2019; Bodnar et al., 2022).

**Conclusion** We proposed three principles (P1-P3) for convolutional learning on SCs, summarized in a general architecture, SCCNNs. Our analysis showed this architecture, guided by the three principles, demonstrates an awareness of the Hodge decomposition and performs rational, effective and expressive learning from simplicial data. Furthermore, our study reveals that SCCNNs exhibit stability and robustness against perturbations in the strengths of simplicial connections. Experimental results validate the benefits of respecting the three principles and the Hodge-awareness, as well as the stability results. Overall, our work establishes a solid theoretical fundation for convolutional learning on SCs, highlighting the importance of the Hodge theorem.

### Reproducibility Statement

We refer to Learning_on_SCs for the reproducibility of our experiments. We also note that the proposed architecture is implemented in the TopoModelX framework (Hajij et al., 2024).

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

## Contents

# A    Illustration for Background

Ths paper relies on the Hodge decomposition and the spectral simplicial theory. To ease the exposition, we illustrate them for the edge flow space. We refer to Barbarossa & Sardellitti (2020); Yang et al. (2021; 2022b) for more details.

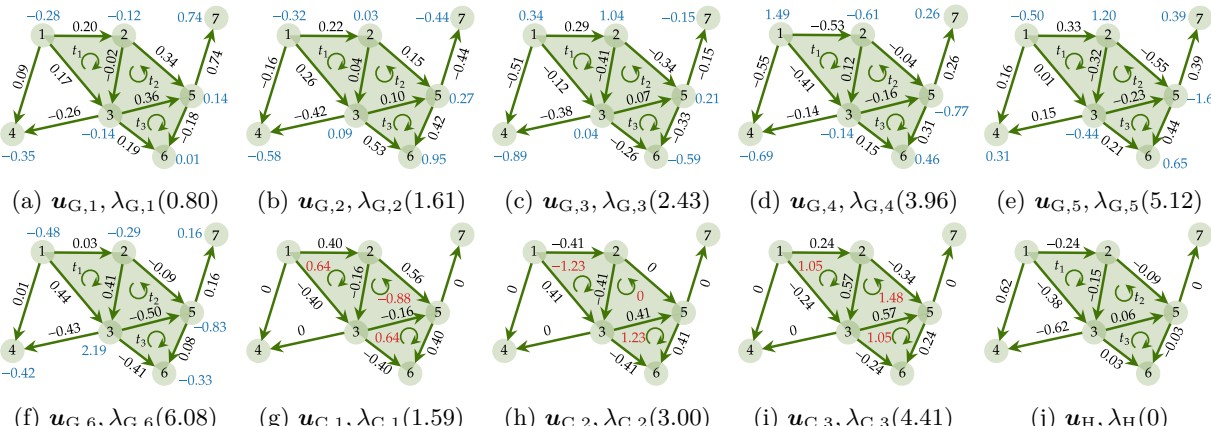

Figure 9: (a)-(f) Six gradient frequencies and the corresponding Fourier basis. We also annotate their divergences, and we see that these eigenvectors with a small eigenvalue have a small magnitude of total divergence, i.e., the edge flow variation in terms of the nodes. Gradient frequencies reflect the nodal variations. (g)-(i) Three curl frequencies and the corresponding Fourier basis. We annotate their curls and we see that these eigenvectors with a small eigenvalue have a small magnitude of total curl, i.e., the edge flow variation in terms of the triangles. Curl frequencies reflect the rotational variations. (j) Harmonic basis with a zero frequency, which has a zero nodal and zero rotational variation.

## A.1    Spectral simplicial theory

Here we show how the eigenvalues of $L_k$ carry the notion of simplicial frequency Yang et al. (2022b). Specifically, we show for $k = 1$ an eigenvalue measures the total divergence or curl of the eigenvector.

- *Gradient Frequency:* the nonzero eigenvalues associated with the eigenvectors $U_{1,G}$ of $L_{1,d}$, which span the gradient space $\text{im}(B_1^\top)$, admit $L_{1,d}u_{1,G} = \lambda_{1,G}u_{1,G}$ for any eigenpair $u_{1,G}$ and $\lambda_{1,G}$. Thus, we have $\lambda_{1,G} = u_{1,G}^\top L_{1,d}u_{1,G} = u_{1,G}^\top B_1^\top B_1 u_{1,G} = \|B_1 u_{1,G}\|_2^2$, which is an Euclidean norm of the divergence, i.e., the total nodal variation of $u_{1,G}$. If an eigenvector has a larger eigenvalue, it has a larger total divergence. For the SFT of an edge flow, if the gradient embedding $\tilde{x}_{1,G}$ has a large weight on such an eigenvector, it contains components with a large divergence, and we say it has a large gradient frequency. Thus, we call such eigenvalues associated with $U_{1,G}$ gradient frequencies.

- *Curl Frequency:* the nonzero eigenvalues associated with the eigenvectors $U_{1,C}$ of $L_{1,u}$, which span the curl space $\text{im}(B_2)$, admit $L_{1,u}u_{1,C} = \lambda_{1,C}u_{1,C}$ for any eigenpair $u_{1,C}$ and $\lambda_{1,C}$. Thus, we have $\lambda_{1,C} = u_{1,C}^\top L_{1,u}u_{1,C} = u_{1,C}^\top B_2 B_2^\top u_{1,C} = \|B_2^\top u_{1,C}\|_2^2$, which is an Euclidean norm of the curl, i.e., the total rotational variation of $u_{1,C}$. If an eigenvector has a larger eigenvalue, it has a larger total curl. For the SFT of an edge flow, if the curl embedding $\tilde{x}_{1,C}$ has a large weight on such an eigenvector, it contains components with a large curl, and we say it has a large curl frequency. Thus, we call such eigenvalues associated with $U_{1,C}$ curl frequencies.

- *Harmonic Frequency:* the zero eigenvalues associated with the eigenvectors $U_{1,H}$, which span the harmonic space $\ker(L_1)$, admit $L_1 u_{1,H} = 0$ for any eigenpair $u_{1,H}$ and $\lambda_{1,H} = 0$. From the definition of $L_1$, we have $B_1 u_{1,H} = B_2^\top u_{1,H} = 0$. That is, the eigenvector $u_{1,H}$ is divergence- and curl-free. We also say such an eigenvector has zero signal variation in terms of the nodes and triangles. This resembles the constant graph signal in the node space. We call such zero eigenvalues as harmonic frequencies.

[Fig. 9](#) shows the simplicial Fourier basis and the corresponding simplicial frequencies of the SC, from which we see how the eigenvalues of $\boldsymbol{L}_1$ can be interpreted as the simplicial frequencies.

For $k = 0$, the eigenvalues of $\boldsymbol{L}_0$ carry the notion of graph frequency, which measures the graph (node) signal smoothness w.r.t. the upper adjacent simplices, i.e., edges. Thus, the curl frequency of $k = 0$ coincides with the graph frequency and a constant graph signal has only harmonic frequency component. For a more general $k$, there exist these three types of simplicial frequencies, which measure the $k$-simplicial signal total variations in terms of faces and cofaces.

## B Simplicial 2-Complex CNNs and Details on Properties

We give two examples where the first is a SCCNN on a SC of order two, and the second is the form of SCCNN with multi-features.

*Example* 25. For $k = 2$, a SCCNN layer reads as

$$
\begin{aligned}
\boldsymbol{x}_0^l &= \sigma(\boldsymbol{H}_0^l \boldsymbol{x}_0^{l-1} + \boldsymbol{H}_{0,\mathrm{u}}^l \boldsymbol{B}_1 \boldsymbol{x}_1^{l-1}), \\
\boldsymbol{x}_1^l &= \sigma(\boldsymbol{H}_{1,\mathrm{d}}^l \boldsymbol{B}_1^\top \boldsymbol{x}_0^{l-1} + \boldsymbol{H}_1^l \boldsymbol{x}_1^{l-1} + \boldsymbol{H}_{1,\mathrm{u}}^l \boldsymbol{B}_2 \boldsymbol{x}_2^{l-1}), \\
\boldsymbol{x}_2^l &= \sigma(\boldsymbol{H}_{2,\mathrm{d}}^l \boldsymbol{B}_2^\top \boldsymbol{x}_1^{l-1} + \boldsymbol{H}_2^l \boldsymbol{x}_2^{l-1}).
\end{aligned}
\tag{17}
$$

Recursively, we see that a SCCNN layer takes as inputs $\{\boldsymbol{x}_0^{l-1}, \boldsymbol{x}_0^{l-2}, \boldsymbol{x}_1^{l-2}, \boldsymbol{x}_2^{l-2}\}$ to compute $\boldsymbol{x}_0^l$. One may find this familar as some type of skip connections in GNNs [Xu et al. (2018b)](#).

*Example* 26 (Multi-Feature SCCNN). A multi-feature SCCNN at layer $l$ takes $\{\boldsymbol{X}_{k-1}^{l-1}, \boldsymbol{X}_k^{l-1}, \boldsymbol{X}_{k+1}^{l-1}\}$ as inputs, each of which has $F_{l-1}$ features, and generates an output $\boldsymbol{X}_k^l$ with $F_l$ features as

$$
\boldsymbol{X}_k^l = \sigma \left( \sum_{t=0}^{T_\mathrm{d}} \boldsymbol{L}_{k,\mathrm{d}}^t \boldsymbol{B}_k^\top \boldsymbol{X}_{k-1}^{l-1} \boldsymbol{W}_{k,\mathrm{d},t}'^l + \sum_{t=0}^{T_\mathrm{d}} \boldsymbol{L}_{k,\mathrm{d}}^t \boldsymbol{X}_k^{l-1} \boldsymbol{W}_{k,\mathrm{d},t}^l + \sum_{t=0}^{T_\mathrm{u}} \boldsymbol{L}_{k,\mathrm{u}}^t \boldsymbol{X}_k^{l-1} \boldsymbol{W}_{k,\mathrm{u},t}^l + \sum_{t=0}^{T_\mathrm{u}} \boldsymbol{L}_{k,\mathrm{u}}^t \boldsymbol{B}_{k+1} \boldsymbol{X}_{k+1}^{l-1} \boldsymbol{W}_{k,\mathrm{u},t}'^l \right)
\tag{18}
$$

where $\boldsymbol{L}^t$ indicates the matrix $t$-power of $\boldsymbol{L}$, while superscript $l$ indicates the layer index.

### B.1 Simplicial locality in details

The construction of SCFs has an intra-simplicial locality. $\boldsymbol{H}_k \boldsymbol{x}_k$, which consists of basic operations $\boldsymbol{L}_{k,\mathrm{d}} \boldsymbol{x}_k$ and $\boldsymbol{L}_{k,\mathrm{u}} \boldsymbol{x}_k$. They are given, on simplex $s_i^k$, by

$$
[\boldsymbol{L}_{k,\mathrm{d}} \boldsymbol{x}_k]_i = \sum_{j \in \mathcal{N}_{i,\mathrm{d}}^k \cup \{i\}} [\boldsymbol{L}_{k,\mathrm{d}}]_{ij} [\boldsymbol{x}_k]_j, \quad [\boldsymbol{L}_{k,\mathrm{u}} \boldsymbol{x}_k]_i = \sum_{j \in \mathcal{N}_{i,\mathrm{u}}^k \cup \{i\}} [\boldsymbol{L}_{k,\mathrm{u}}]_{ij} [\boldsymbol{x}_k]_j,
\tag{19}
$$

where $s_i^k$ aggregates signals from its lower and upper neighbors, $\mathcal{N}_{i,\mathrm{d}}^k$ and $\mathcal{N}_{i,\mathrm{u}}^k$. We can compute the $t$-step shifting recursively as $\boldsymbol{L}_{k,\mathrm{d}}^t \boldsymbol{x}_k = \boldsymbol{L}_{k,\mathrm{d}}(\boldsymbol{L}_{k,\mathrm{d}}^{t-1} \boldsymbol{x}_k)$, a one-step shifting of the $(t-1)$-shift result; likewise for $\boldsymbol{L}_{k,\mathrm{u}}^t \boldsymbol{x}_k$. A SCF linearly combines such multi-step simplicial shiftings based on lower and upper adjacencies. Thus, the output $\boldsymbol{H}_k \boldsymbol{x}_k$ is localized in $T_\mathrm{d}$-hop lower and $T_\mathrm{u}$-hop upper $k$-simplicial neighborhoods ([Yang et al., 2022b](#)). SCCNNs preserve such *intra-simplicial locality* as the elementwise nonlinearity does not alter the information locality, shown in [Figs. 2b](#) and [2c](#).

A SCCNN takes the data on $k$- and $(k \pm 1)$-simplices at layer $l-1$ to compute $\boldsymbol{x}_k^l$, causing interactions between $k$-simplices and their (co)faces when all SCFs are identity. In turn, $\boldsymbol{x}_{k-1}^{l-1}$ contains information on $(k-2)$-simplices from layer $l-2$. Likewise for $\boldsymbol{x}_{k+1}^{l-1}$, thus, $\boldsymbol{x}_k^l$ also contains information up to $(k \pm 2)$-simplices if $L \geq 2$, because $\boldsymbol{B}_k \sigma(\boldsymbol{B}_{k+1}) \neq \boldsymbol{0}$. Accordingly, this *inter-simplicial locality* extends to the whole SC if $L \geq K$, unlike linear filters in a SC where the locality happens up to the adjacent simplices ([Schaub et al., 2021](#); [Isufi & Yang, 2022](#)), which limits its expressive power. This locality is further coupled with the intra-locality through three SCFs such that a node not only interacts with the edges incident to it and direct triangles including it, but also edges and triangles further hops away which contribute to the neighboring nodes, as shown in [Fig. 2d](#).

## B.2 Complexity

In a SCCNN layer for computing $\boldsymbol{x}_k^l$, there are $2 + T_\mathrm{d} + T_\mathrm{u}$ filter coefficients for the SCF $\boldsymbol{H}_k^l$, and $1 + T_\mathrm{d}$ and $1 + T_\mathrm{u}$ for $\boldsymbol{H}_{k,\mathrm{d}}^l$ and $\boldsymbol{H}_{k,\mathrm{u}}^l$, respectively, which gives the parameter complexity of order $\mathcal{O}(T_\mathrm{d} + T_\mathrm{u})$. This complexity will increase by $F_l F_{l-1}$ fold for the multi-feature case, and likewise for the computational complexity. Given the inputs $\{\boldsymbol{x}_{k-1}^{l-1}, \boldsymbol{x}_k^{l-1}, \boldsymbol{x}_{k+1}^{l-1}\}$, we discuss the computation complexity of $\boldsymbol{x}_k^l$ in Eq. (4).

First, consider the SCF operation $\boldsymbol{H}_k^l \boldsymbol{x}_k^{l-1}$. As discussed in the localities, it is a composition of $T_\mathrm{d}$-step lower and $T_\mathrm{u}$-step upper simplicial shiftings. Each simplicial shifting has a computational complexity of order $\mathcal{O}(n_k m_k)$ dependent on the number of neighbors $m_k$ where $n_k$ is the number of $k$-simplices. Thus, this operation has a complexity of order $\mathcal{O}(n_k m_k (T_\mathrm{d} + T_\mathrm{u}))$.

Second, consider the lower SCF operation $\boldsymbol{H}_{k,\mathrm{d}}^l \boldsymbol{B}_k^\top \boldsymbol{x}_{k-1}^{l-1}$. As incidence matrix $\boldsymbol{B}_k$ is sparse, it has $n_k(k+1)$ nonzero entries as each $k$-simplex has $k+1$ faces. This leads to a complexity of order $O(n_k k)$ for operation $\boldsymbol{B}_k^\top \boldsymbol{x}_{k-1}^{l-1}$. Followed by a lower SCF operation, i.e., a $T_\mathrm{d}$-step lower simplicial shifting, thus, a complexity of order $\mathcal{O}(kn_k + n_k m_k T_\mathrm{d})$ is needed.

Third, consider the upper SCF operation $\boldsymbol{H}_{k,\mathrm{u}}^l \boldsymbol{B}_{k+1} \boldsymbol{x}_{k+1}^{l-1}$. Likewise, incidence matrix $\boldsymbol{B}_{k+1}$ has $n_{k+1}(k+2)$ nonzero entries. This leads to a complexity of order $\mathcal{O}(n_{k+1} k)$ for the projection operation $\boldsymbol{B}_{k+1} \boldsymbol{x}_{k+1}^{l-1}$. Followed by an upper SCF operation, i.e., a $T_\mathrm{u}$-step upper simplicial shifting, thus, a complexity of order $\mathcal{O}(kn_{k+1} + n_k m_k T_\mathrm{u})$ is needed.

Finally, we have a computational complexity of order $\mathcal{O}(k(n_k + n_{k+1}) + N_k M_k (T_\mathrm{d} + T_\mathrm{u}))$ in total.

*Remark* 27. The lower SCF operation $\boldsymbol{H}_{k,\mathrm{d}}^l \boldsymbol{B}_k^\top \boldsymbol{x}_{k-1}^{l-1}$ can be further reduced if $n_{k-1} \ll n_k$. Note that we have

$$\boldsymbol{H}_{k,\mathrm{d}}^l \boldsymbol{B}_k^\top \boldsymbol{x}_{k-1}^{l-1} = \sum_{t=0}^{T_\mathrm{d}} w_{k,\mathrm{d},t}'^l \boldsymbol{L}_{k,\mathrm{d}}^t \boldsymbol{B}_k^\top \boldsymbol{x}_{k-1}^{l-1} = \boldsymbol{B}_k^\top \sum_{t=0}^{T_\mathrm{d}} w_{k,\mathrm{d},t}'^l \boldsymbol{L}_{k-1,\mathrm{u}}^t \boldsymbol{x}_{k-1}^{l-1}, \tag{20}$$

where the second equality comes from that $\boldsymbol{L}_{k,\mathrm{d}} \boldsymbol{B}_k^\top = \boldsymbol{B}_k^\top \boldsymbol{B}_k \boldsymbol{B}_k^\top = \boldsymbol{B}_k^\top \boldsymbol{L}_{k-1,\mathrm{u}}$, $\boldsymbol{L}_{k,\mathrm{d}}^2 \boldsymbol{B}_k^\top = (\boldsymbol{B}_k^\top \boldsymbol{B}_k)(\boldsymbol{B}_k^\top \boldsymbol{B}_k) \boldsymbol{B}_k^\top = \boldsymbol{B}_k^\top (\boldsymbol{B}_k \boldsymbol{B}_k^\top)(\boldsymbol{B}_k \boldsymbol{B}_k^\top) = \boldsymbol{B}_k^\top \boldsymbol{L}_{k-1,\mathrm{u}}$ and likewise for general $t$. Using the RHS of Eq. (20) where the simplicial shifting is performed in the $(k-1)$-simplicial space, we have a complexity of order $\mathcal{O}(kn_k + n_{k-1} m_{k-1} T_\mathrm{d})$. Similarly, we have

$$\boldsymbol{H}_{k,\mathrm{u}}^l \boldsymbol{B}_{k+1} \boldsymbol{x}_{k+1}^{l-1} = \sum_{t=0}^{T_\mathrm{u}} w_{k,\mathrm{u},t}'^l \boldsymbol{L}_{k,\mathrm{u}}^t \boldsymbol{B}_{k+1} \boldsymbol{x}_{k+1}^{l-1} = \boldsymbol{B}_{k+1} \sum_{t=0}^{T_\mathrm{u}} w_{k,\mathrm{u},t}'^l \boldsymbol{L}_{k+1,\mathrm{d}}^t \boldsymbol{x}_{k+1}^{l-1} \tag{21}$$

where the simplicial shifting is performed in the $(k+1)$-simplicial space. If it follows that $n_{k+1} \ll n_k$, we have a smaller complexity of $\mathcal{O}(kn_{k+1} + n_{k+1} m_{k+1} T_\mathrm{u})$ by using the RHS of Eq. (21).

## B.3 Symmetries of SCs and simplicial data, Equivariance of SCCNNs

**Permutation symmetry of SCs.** There exists a permutation group $P_{n_k}$ for each set $\mathcal{S}^k$ in a SC of order $K$. For $K = 0$, this gives the graph permutation group. We can combine these groups for different simplex orders by a group product to form a larger permutation group $P = \times_k P_{n_k}$, which is a symmetry group of SCs and simplicial data, assuming vertices in each simplex are consistently ordered. That is, we have, for $p = (p_0, p_1, \ldots, p_K) \in P$, $[p \cdot \boldsymbol{L}_k]_{ij} = [\boldsymbol{L}_k]_{p_k^{-1}(i) p_k^{-1}(j)}$, $[p \cdot \boldsymbol{B}_k]_{ij} = [\boldsymbol{B}_k]_{p_{k-1}^{-1}(i) p_k^{-1}(j)}$, and $[p \cdot \boldsymbol{x}_k]_i = [\boldsymbol{x}_k]_{p_k^{-1}(i)}$. This permutation symmetry of SCs gives us the freedom to list simplices in any order.

**Orientation symmetry of simplicial data.** The orientation of a simplex is an equivalence class that two orientations are equivalent if they differ by an even permutation Lim (2020); Munkres (2018). Thus, for a simplex $s_i^k = \{i_0, \ldots, i_k\}$ with $k > 0$, we have an *orientation symmetry* group $O_{k,i} = \{o_{k,i}^+, o_{k,i}^-\}$ by a group homomorphism which maps all the even permutations of $\{i_0, \ldots, i_k\}$ to the identity element $o_{k,i}^+$ and all the odd permutations to the reverse operation $o_{k,i}^-$.

We can further combine the orientation groups of all simplices in a SC as $O = \times_{i,k} O_{k,i}$ by using a group product. This however is not a symmetry group of an oriented SC because $o_{k,i}^- \cdot \boldsymbol{L}_k$ changes the signs of

$\boldsymbol{L}_k$ elements in $i$th column and row, and $o_{k,i}^- \cdot \boldsymbol{B}_k$ changes the $i$th row, resulting in a different SC topology. Instead, it is a symmetry group of the data space, due to its alternating nature w.r.t. simplices. For $o \in O$ we have $[o \cdot \boldsymbol{x}_k]_i = o_{k,i} \cdot f_k(s_i^k) = f_k(o_{k,i}^{-1} \cdot s_i^k)$, i.e., $[\boldsymbol{x}_k]_i$ remains unchanged w.r.t. the changed orientation of $s_i^k$. This gives us the freedom to choose reference orientations of simplices when working with simplicial data.

**Theorem 28** (Permutation Equivariance)**.** *A SCCNN in* Eq. (4) *is P-equivariant. For all $p \in P$, we have* $p \cdot \mathrm{SCCNN}_k : \{p_{k-1} \cdot \boldsymbol{x}_{k-1}, p_k \cdot \boldsymbol{x}_k, p_{k+1} \cdot \boldsymbol{x}_{k+1}\} \to p_k \boldsymbol{x}_k$.

**Theorem 29** (Orientation Equivariance)**.** *A SCCNN in* Eq. (4) *is O-equivariant if $\sigma(\cdot)$ is odd. For all $o \in O$, we have* $o \cdot \mathrm{SCCNN}_k : \{o_{k-1} \cdot \boldsymbol{x}_{k-1}, o_k \cdot \boldsymbol{x}_k, o_{k+1} \cdot \boldsymbol{x}_{k+1}\} \to o_k \cdot \boldsymbol{x}_k$.

*Proof.* (informal) Both the permutation group and orientation group have linear matrix representations. By following the same procedure in (Bodnar et al., 2021b, Appendix D) or Roddenberry et al. (2021), we can prove the equivariance. □

## B.4 Diffusion process on SCs

Diffusion process on graphs can be generalized to SCs to characterize the evolution of simplicial data over the SC, in analogy to data diffusion on nodes Anand et al. (2022); Ziegler et al. (2022); Grady & Polimeni (2010). Here we provide an informal treatment of how discretizing diffusion equations on SCs can give resemblances of simplicial shifting layers. Consider diffusion equation and its Euler discretization with a unit time step

$$\dot{\boldsymbol{x}}_k(t) = -\boldsymbol{L}_k \boldsymbol{x}_k(t), \text{ Euler step: } \boldsymbol{x}_k(t+1) = \boldsymbol{x}_k(t) - \boldsymbol{L}_k \boldsymbol{x}_k(t) = (\boldsymbol{I} - \boldsymbol{L}_k)\boldsymbol{x}_k(t) \tag{22}$$

with an initial condition $\boldsymbol{x}_k(t) = \boldsymbol{x}_k^0$. The solution of this diffusion is $\boldsymbol{x}_k(t) = \exp(-\boldsymbol{L}_k t)\boldsymbol{x}_k^0$. As the time increases, the simplicial data reaches to a steady state $\dot{\boldsymbol{x}}_k(t) = \boldsymbol{0}$, which lies in the harmonic space $\ker(\boldsymbol{L}_k)$. The simplicial shifting layer resembles this Euler step with a weight and nonlinearity when viewing the time step as layer index. Thus, a NN composed of simplicial shifting layers can suffer from oversmoothing on SCs, giving outputs with decreasing Dirichlet energies as the number of layers increases.

Now let us consider the case where the two Laplacians have different coefficients

$$\dot{\boldsymbol{x}}_k(t) = -\boldsymbol{L}_{k,\mathrm{d}}\boldsymbol{x}_k(t) - \gamma\boldsymbol{L}_{k,\mathrm{u}}\boldsymbol{x}_k(t), \text{ Euler step: } \boldsymbol{x}_k(t) = (\boldsymbol{I} - \boldsymbol{L}_{k,\mathrm{d}} - \gamma\boldsymbol{L}_{k,\mathrm{u}})\boldsymbol{x}_k(t). \tag{23}$$

The steady state of this diffusion equation follows $(\boldsymbol{L}_{k,\mathrm{d}} + \gamma\boldsymbol{L}_{k,\mathrm{u}})\boldsymbol{x}_k(t) = \boldsymbol{0}$, where $\boldsymbol{x}_k(t)$ would be in the kernal space of $\boldsymbol{L}_k$ still. However, before reaching this state, when the time increases, $\boldsymbol{x}_k(t)$ would primarily approach to the kernel of $\boldsymbol{B}_{k+1}^\top$ if $\gamma \gg 1$, in which the lower part of the Dirichlet energy remains, i.e., the decrease of $D(\boldsymbol{x}(t))$ slows down.

When accounting for inter-simplicial couplings, consider there are nontrivial $\boldsymbol{x}_{k-1}$ and $\boldsymbol{x}_{k+1}$ and the diffusion equation becomes

$$\dot{\boldsymbol{x}}_k(t) = -\boldsymbol{L}_k \boldsymbol{x}_k(t) + \boldsymbol{B}_k^\top \boldsymbol{x}_{k-1} + \boldsymbol{B}_{k+1}\boldsymbol{x}_{k+1}, \tag{24}$$

which has source terms $\boldsymbol{B}_k^\top \boldsymbol{x}_{k-1} + \boldsymbol{B}_{k+1}\boldsymbol{x}_{k+1}$. Consider a steady state $\dot{\boldsymbol{x}}_k = 0$. We have $\boldsymbol{L}_k \boldsymbol{x}_k(t) = \boldsymbol{x}_{k,\mathrm{d}} + \boldsymbol{x}_{k,\mathrm{u}}$, where $\boldsymbol{x}_k$ is not in the kernel space of $\boldsymbol{L}_k$. The Euler discretization gives

$$\boldsymbol{x}_k(t+1) = (\boldsymbol{I} - \boldsymbol{L}_k)\boldsymbol{x}_k(t) + \boldsymbol{x}_{k,\mathrm{d}} + \boldsymbol{x}_{k,\mathrm{u}}. \tag{25}$$

The layer in Bunch et al. (2020) $\boldsymbol{x}_k^{l+1} = w_0(\boldsymbol{I} - \boldsymbol{L}_k)\boldsymbol{x}_k^l + w_1\boldsymbol{x}_{k,\mathrm{d}} + w_2\boldsymbol{x}_{k,\mathrm{u}}$ is a weighted variant of above step when viewing time steps as layers.

# C Related works

We first compare SCCNN with other architectures on if they respect the three principles in Section 3 in Table 5. We then describe how the SCCNN in Eq. (18) generalize other NNs on graphs and SCs in Table 6. For simplicity, we use $\boldsymbol{Y}$ and $\boldsymbol{X}$ to denote the output and input, respectively, without the index $l$. Note that for GNNs, $\boldsymbol{L}_{0,\mathrm{d}}$ is not defined.

Table 5: Comparisons between SCCNN and other architectures on if they respect the three principles.

| Methods | Scheme | P1 | P2 | P3 |
|---|---|---|---|---|
| MPSN Bodnar et al. (2021b) | message-passing | yes | yes | no, only direct neighborhoods |
| Eq. (11) of MPSN, or Bunch et al. (2020) | convolutional | no | yes | no, only direct neighborhoods |
| Eq. (27) of MPSN | convolutional | yes | yes | no, only direct neighborhoods |
| SNN Ebli et al. (2020) | convolutional | no | no | yes |
| PSNN Roddenberry et al. (2021) | convolutional | yes | no | no, only direct neighborhoods |
| SCNN Yang et al. (2022a) | convolutional | yes | no | yes |
| SCCNN | convolutional | yes | yes | yes |

Table 6: SCCNNs generalize other convolutional architectures on SCs.

| Methods | Parameters (n.d. denotes "not defined") |
|---|---|
| Ebli et al. (2020) | $w_{k,\mathrm{d},t}^l = w_{k,\mathrm{u},t}^l, \boldsymbol{H}_{k,\mathrm{d}}^l, \boldsymbol{H}_{k,\mathrm{u}}^l$ n.d. |
| Roddenberry et al. (2021) | $T_\mathrm{d} = T_\mathrm{u} = 1, \boldsymbol{H}_{k,\mathrm{d}}^l, \boldsymbol{H}_{k,\mathrm{u}}^l$ n.d. |
| Yang et al. (2022a) | $\boldsymbol{H}_{k,\mathrm{d}}^l, \boldsymbol{H}_{k,\mathrm{u}}^l$ n.d. |
| Bunch et al. (2020) | $T_\mathrm{d} = T_\mathrm{u} = 1, \boldsymbol{H}_{k,\mathrm{d}}^l = \boldsymbol{H}_{k,\mathrm{u}}^l = \boldsymbol{I}$ |
| Bodnar et al. (2021b) | $T_\mathrm{d} = T_\mathrm{u} = 1, \boldsymbol{H}_{k,\mathrm{d}}^l = \boldsymbol{H}_{k,\mathrm{u}}^l = \boldsymbol{I}$ |

Gama et al. (2020a) proposed to build a GNN layer with the form

$$\boldsymbol{Y}_0 = \sigma\left( \sum_{t=0}^{T_\mathrm{u}} \boldsymbol{L}_0^t \boldsymbol{X}_0 \boldsymbol{W}_{0,\mathrm{u},t} \right) \tag{26}$$

where the convolution step is performed via a graph filter (Sandryhaila & Moura, 2013; 2014; Gama et al., 2019a; 2020b). This GNN can be easily built as a special SCCNN without contributions from edges. Furthermore, Defferrard et al. (2016) considered a fast implementation of this GNN via a Chebyshev polynomial, while Wu et al. (2019) simplified this by setting $\boldsymbol{W}_{0,t,\mathrm{u}}$ as zeros for $t < T_\mathrm{u}$. Kipf & Welling (2017) further simplified this by setting $T_\mathrm{u} = 1$, namely, GCN.

Yang et al. (2022a) proposed a simplicial convolutional neural network (SCNN) to learn from $k$-simplicial signals

$$\boldsymbol{Y}_k = \sigma\left( \sum_{t=0}^{T_\mathrm{d}} \boldsymbol{L}_{k,\mathrm{d}}^t \boldsymbol{X}_k \boldsymbol{W}_{k,\mathrm{d},t} + \sum_{t=0}^{T_\mathrm{u}} \boldsymbol{L}_{k,\mathrm{u}}^t \boldsymbol{X}_k \boldsymbol{W}_{k,\mathrm{u},t} \right) \tag{27}$$

where the linear operation is also defined as a simplicial convolution filter in Yang et al. (2022b). This is a special SCCNN with a focus on one simplex level without taking into the lower and upper contributions consideration. The simplicial neural network (SNN) of Ebli et al. (2020) did not differentiate the lower and the upper convolutions with a form of $\boldsymbol{Y}_k = \sigma(\sum_{t=0}^{T} \boldsymbol{L}_k^t \boldsymbol{X}_k \boldsymbol{W}_{k,t})$, which leads to a joint processing in the gradient and curl subspaces as analyzed in Section 4.

While Roddenberry et al. (2021) proposed an architecture (referred to as PSNN)of a particular form of Eq. (27) with $T_\mathrm{d} = T_\mathrm{u} = 1$, performing only a one-step simplicial shifting Eq. (19). Keros et al. (2022) also performs a one-step simplicial shifting but with an inverted Hodge Laplacian to localize the homology group in an SC. An attention mechanism was added to both SCNNs and PSNNs by Giusti et al. (2022) and Goh et al. (2022), respectively. Battiloro et al. (2023) added the attention mechanism to SCCNNs.

To account for the information from adjacent simplices, Bunch et al. (2020) proposed a simplicial 2-complex CNN (S2CCNN)

$$\begin{aligned} \boldsymbol{Y}_0 &= \sigma\big( \boldsymbol{L}_0 \boldsymbol{X}_0 \boldsymbol{W}_{0,\mathrm{u},1} + \boldsymbol{B}_1 \boldsymbol{X}_1 \boldsymbol{W}'_{0,\mathrm{u},0} \big) \\ \boldsymbol{Y}_1 &= \sigma\big( \boldsymbol{B}_1^\top \boldsymbol{X}_0 \boldsymbol{W}_{1,\mathrm{d},0} + \boldsymbol{L}_1 \boldsymbol{X}_1 \boldsymbol{W}_{1,1} + \boldsymbol{B}_2 \boldsymbol{X}_2 \boldsymbol{W}'_{1,\mathrm{u},0} \big) \\ \boldsymbol{Y}_2 &= \sigma\big( \boldsymbol{B}_2^\top \boldsymbol{X}_1 \boldsymbol{W}_{2,\mathrm{d},0} + \boldsymbol{L}_{2,\mathrm{u}} \boldsymbol{X}_2 \boldsymbol{W}_{2,\mathrm{u},1} \big) \end{aligned} \tag{28}$$

which is limited to SCs of order two. Note that instead of Hodge Laplacians, simplicial adjacency matrices with self-loops are used in Bunch et al. (2020), which encode equivalent information as setting all filter orders in SCCNNs as one. It is a particular form of the SCCNN where the SCF is a one-step simplicial shifting operation without differentiating the lower and upper shifting, and the lower and upper contributions are simply added, not convolved or shifted by lower and upper SCFs. That is, Bunch et al. (2020) can be obtained from Eq. (4) by setting lower and upper SCFs as identity, $\boldsymbol{H}_{k,\mathrm{d}} = \boldsymbol{H}_{k,\mathrm{u}} = \boldsymbol{I}$, and setting $w_{k,\mathrm{d},t} = w_{k,\mathrm{u},t}$ and $T_{\mathrm{d}} = T_{\mathrm{u}} = 1$ for the SCF $\boldsymbol{H}_k$. The convolution in Yang et al. (2022c, Eq. 3) is the same as Bunch et al. (2020) though it was performed in a block matrix fashion.

The combination of graph shifting and edge shifting in Chen et al. (2022b) can be again seen as a special S2CCNN, where the implementation was performed in a block matrix fashion. Bodnar et al. (2021b) proposed a message passing scheme which collects information from one-hop simplicial neighbors and direct faces and cofaces as Bunch et al. (2020) and Yang et al. (2022c), but replacing the one-step shifting and projections from (co)faces by some learnable functions. The same message passing was applied for simplicial representation learning by Hajij et al. (2021).

Lastly, there are works on signal processing and NNs on cell complexes. For example, Sardellitti et al. (2021); Roddenberry et al. (2022) generalized the signal processing techniques from SCs to cell complexes, Bodnar et al. (2021a); Hajij et al. (2020) performed message passing on cell complexes as in SCs and Hajij et al. (2022) added the attention mechanism. Cell complexes are a more general model compared to SCs, where $k$-cells compared to $k$-simplices contain any shapes homeomorphic to a $k$-dimensional closed balls in Euclidean space, e.g., a filled polygon is a 2-cell while only triangles are 2-simplices. We refer to Hansen & Ghrist (2019) for a more formal definition of cell complexes. Despite cell complexes are more powerful to model real-world higher-order structures, SCCNNs can be easily generalized to cell complexes by considering any $k$-cells instead of only $k$-simplices in the algebraic representations, and the theoretical analysis in this paper can be adapted to cell complexes as well.

# D   Proofs for Section 3

## D.1   Dirichlet energy minimization perspective

**Hodge Laplacian smoothing.**   We can find the gradient of problem Eq. (7) as $\frac{\partial D}{\partial \boldsymbol{x}_k} = \boldsymbol{B}_k^\top \boldsymbol{B}_k \boldsymbol{x}_k + \gamma \boldsymbol{B}_{k+1} \boldsymbol{B}_{k+1}^\top \boldsymbol{x}_k$, thus, a gradient descent step follows as Eq. (7) with a step size $\eta$.

*Proof of Proposition 5.* Consider $\eta = 1$.

$$
\begin{aligned}
D(\boldsymbol{x}_k^{l+1}) &= w_0^2 \|\boldsymbol{B}_k(\boldsymbol{I} - \boldsymbol{L}_{k,\mathrm{d}} - \gamma \boldsymbol{L}_{k,\mathrm{u}})\boldsymbol{x}_k^l\|_2^2 + w_0^2 \|\boldsymbol{B}_{k+1}^\top(\boldsymbol{I} - \boldsymbol{L}_{k,\mathrm{d}} - \gamma \boldsymbol{L}_{k,\mathrm{u}})\boldsymbol{x}_k^l\|_2^2 \\
&= w_0^2 \|(\boldsymbol{I} - \boldsymbol{L}_{k-1,\mathrm{u}})\boldsymbol{B}_k \boldsymbol{x}_k^l\|_2^2 + w_0^2 \|(\boldsymbol{I} - \gamma \boldsymbol{L}_{k+1,\mathrm{d}})\boldsymbol{B}_{k+1}^\top \boldsymbol{x}_k^l\|_2^2 \\
&\leq w_0^2 \|(\boldsymbol{I} - \boldsymbol{L}_{k-1,\mathrm{u}})\|_2^2 \|\boldsymbol{B}_k \boldsymbol{x}_k^l\|_2^2 + w_0^2 \|(\boldsymbol{I} - \gamma \boldsymbol{L}_{k+1,\mathrm{d}})\|_2^2 \|\boldsymbol{B}_{k+1}^\top \boldsymbol{x}_k^l\|_2^2
\end{aligned}
\tag{29}
$$

which follows from triangle inequality. By definition, we have $\|\boldsymbol{I} - \boldsymbol{L}_{k-1,\mathrm{u}}\|_2^2 = \|\boldsymbol{I} - \boldsymbol{L}_{k,\mathrm{d}}\|_2^2$ and $\|\boldsymbol{I} - \boldsymbol{L}_{k,\mathrm{u}}\|_2^2 = \|\boldsymbol{I} - \boldsymbol{L}_{k+1,\mathrm{d}}\|_2^2$. Also, we have $\|\boldsymbol{I} - \boldsymbol{L}_k\|_2^2 = \max\{\|\boldsymbol{I} - \boldsymbol{L}_{k,\mathrm{d}}\|_2^2, \|\boldsymbol{I} - \boldsymbol{L}_{k,\mathrm{u}}\|_2^2\}$ Thus, we have $D(\boldsymbol{x}_k^{l+1}) \leq w_0^2 \|\boldsymbol{I} - \boldsymbol{L}_k\|_2^2 D(\boldsymbol{x}_k^l)$ when $\gamma = 1$. When $w_0^2 \|\boldsymbol{I} - \boldsymbol{L}_k\|_2^2 < 1$, Dirichlet energy $D(\boldsymbol{x}_k^l)$ will exponentially decrease as $l$ increases. $\qquad\square$

When $\gamma \neq 1$, from Eq. (29), we have $D(\boldsymbol{x}_k^{l+1}) = D_{\mathrm{d}}(\boldsymbol{x}_k^{l+1}) + D_{\mathrm{u}}(\boldsymbol{x}_k^{l+1})$, which follows

$$
D_{\mathrm{d}}(\boldsymbol{x}_k^{l+1}) \leq w_0^2 \|(\boldsymbol{I} - \boldsymbol{L}_{k,\mathrm{d}})\|_2^2 D_{\mathrm{d}}(\boldsymbol{x}_k^l) \text{ and } D_{\mathrm{u}}(\boldsymbol{x}_k^{l+1}) \leq w_0^2 \|(\boldsymbol{I} - \gamma \boldsymbol{L}_{k,\mathrm{u}})\|_2^2 D_{\mathrm{u}}(\boldsymbol{x}_k^l)
\tag{30}
$$

When $\gamma = 1$, the oversmoothing condition is $\|\boldsymbol{I} - \boldsymbol{L}_k\|_2^2 = \max\{\|\boldsymbol{I} - \boldsymbol{L}_{k,\mathrm{d}}\|_2^2, \|\boldsymbol{I} - \boldsymbol{L}_{k,\mathrm{u}}\|_2^2\} < \frac{1}{w_0^2}$. If $\|\boldsymbol{I} - \boldsymbol{L}_k\|_2^2 = \|\boldsymbol{I} - \boldsymbol{L}_{k,\mathrm{d}}\|_2^2$, under the oversmoothing condition, by not restricting $\gamma$ to be 1, $w_0^2 \|(\boldsymbol{I} - \gamma \boldsymbol{L}_{k,\mathrm{u}})\|_2^2$ can be larger than 1 depending on the choice, which means $D_{\mathrm{u}}(\boldsymbol{x}_k^l)$ does not necessarily decrease, so does not $D(\boldsymbol{x}_k^l)$.

**Hodge Laplacian smoothing with sources.** The gradient of the objective in Eq. (8) is given by $\boldsymbol{L}_k\boldsymbol{x}_k^l - \boldsymbol{B}_k^\top\boldsymbol{x}_{k-1} - \boldsymbol{B}_{k+1}\boldsymbol{x}_{k+1}$, which gives the gradient descent update in Eq. (8) with a step size $\eta$.

Consider the layer in Bunch et al. (2020) $\boldsymbol{x}_k^{l+1} = w_0(\boldsymbol{I} - \boldsymbol{L}_k)\boldsymbol{x}_k^l + w_1\boldsymbol{x}_{k,\mathrm{d}} + w_2\boldsymbol{x}_{k,\mathrm{u}}$ with some weights. By triangle inequality, we have $D(\boldsymbol{x}_k^{l+1}) \le w_0^2\|\boldsymbol{I} - \boldsymbol{L}_k\|_2^2 D(\boldsymbol{x}_k^l) + w_1^2\lambda_{\max}(\boldsymbol{L}_{k,\mathrm{d}})\|\boldsymbol{x}_{k,\mathrm{d}}\|_2^2 + w_2^2\lambda_{\max}(\boldsymbol{L}_{k,\mathrm{u}})\|\boldsymbol{x}_{k,\mathrm{u}}\|_2^2$. If the weight $w_0$ is small enough following the condition in Proposition 5, the contribution from the projections, controled by weights $w_1$ and $w_2$, can compromise the decrease by $w_0$, maintaining the Dirichlet energy.

## E Proofs for Section 4

### E.1 The SCF is Hodge-invariant in Proposition 12

*Proof.* We first give the following lemma.

**Lemma 30.** *Any finite set of eigenfunctions of a linear operator spans an invariant subspace.*

Then, the proof follows from Lemma 30 and Proposition 10. □

### E.2 A derivation of the spectral frequency response in Eq. (11)

**SFT of $\boldsymbol{x}_k$.** First, the SFT of $\boldsymbol{x}_k$ is given by $\tilde{\boldsymbol{x}}_k = [\tilde{\boldsymbol{x}}_{k,\mathrm{H}}^\top, \tilde{\boldsymbol{x}}_{k,\mathrm{G}}^\top, \tilde{\boldsymbol{x}}_{k,\mathrm{C}}^\top]^\top$ with the *harmonic embedding* $\tilde{\boldsymbol{x}}_{k,\mathrm{H}} = \boldsymbol{U}_{k,\mathrm{H}}^\top\boldsymbol{x}_k = \boldsymbol{U}_{k,\mathrm{H}}^\top\boldsymbol{x}_{k,\mathrm{H}}$ in the zero frequencies, the *gradient embedding* $\tilde{\boldsymbol{x}}_{k,\mathrm{G}} = \boldsymbol{U}_{k,\mathrm{G}}^\top\boldsymbol{x}_k = \boldsymbol{U}_{k,\mathrm{G}}^\top\boldsymbol{x}_{k,\mathrm{G}}$ in the gradient frequencies, and the *curl embedding* $\tilde{\boldsymbol{x}}_{k,\mathrm{C}} = \boldsymbol{U}_{k,\mathrm{C}}^\top\boldsymbol{x}_k = \boldsymbol{U}_{k,\mathrm{C}}^\top\boldsymbol{x}_{k,\mathrm{C}}$ in the curl frequencies.

**SFT of $\boldsymbol{H}_k\boldsymbol{x}_k$.** By diagonalizing an SCF $\boldsymbol{H}_k$ with $\boldsymbol{U}_k$, we have

$$\boldsymbol{H}_k\boldsymbol{x}_k = \boldsymbol{U}_k\widetilde{\boldsymbol{H}}_k\boldsymbol{U}_k^\top\boldsymbol{x}_k = \boldsymbol{U}_k(\tilde{\boldsymbol{h}}_k \odot \tilde{\boldsymbol{x}}_k) \tag{31}$$

where $\widetilde{\boldsymbol{H}}_k = \mathrm{diag}(\tilde{\boldsymbol{h}}_k)$. Here, $\tilde{\boldsymbol{h}}_k = [\tilde{\boldsymbol{h}}_{k,\mathrm{H}}^\top, \tilde{\boldsymbol{h}}_{k,\mathrm{G}}^\top, \tilde{\boldsymbol{h}}_{k,\mathrm{C}}^\top]^\top$ is the *frequency response*, given by

$$\begin{cases} \textit{harmonic response} : \tilde{\boldsymbol{h}}_{k,\mathrm{H}} = (w_{k,\mathrm{d},0} + w_{k,\mathrm{u},0})\boldsymbol{1}, \\ \textit{gradient response} : \tilde{\boldsymbol{h}}_{k,\mathrm{G}} = \sum_{t=0}^{T_\mathrm{d}} w_{k,\mathrm{d},t}\boldsymbol{\lambda}_{k,\mathrm{G}}^{\odot t} + w_{k,\mathrm{u},0}\boldsymbol{1}, \\ \textit{curl response} : \tilde{\boldsymbol{h}}_{k,\mathrm{C}} = \sum_{t=0}^{T_\mathrm{u}} w_{k,\mathrm{u},t}\boldsymbol{\lambda}_{k,\mathrm{C}}^{\odot t} + w_{k,\mathrm{d},0}\boldsymbol{1}, \end{cases}$$

with $(\cdot)^{\odot t}$ the elementwise $t$-th power of a vector. Thus, we can express $\tilde{\boldsymbol{h}}_k \odot \tilde{\boldsymbol{x}}_k$ as

$$[(\tilde{\boldsymbol{h}}_{k,\mathrm{H}} \odot \tilde{\boldsymbol{x}}_{k,\mathrm{H}})^\top, (\tilde{\boldsymbol{h}}_{k,\mathrm{G}} \odot \tilde{\boldsymbol{x}}_{k,\mathrm{G}})^\top, (\tilde{\boldsymbol{h}}_{k,\mathrm{C}} \odot \tilde{\boldsymbol{x}}_{k,\mathrm{C}})^\top]^\top. \tag{32}$$

**SFT of projections.** Second, the lower projection $\boldsymbol{x}_{k,\mathrm{d}} \in \mathrm{im}(\boldsymbol{B}_k^\top)$ has only a nonzero gradient embedding $\tilde{\boldsymbol{x}}_{k,\mathrm{d}} = \boldsymbol{U}_{k,\mathrm{G}}^\top\boldsymbol{x}_{k,\mathrm{d}}$. Likewise, the upper projection $\boldsymbol{x}_{k,\mathrm{u}} \in \mathrm{im}(\boldsymbol{B}_{k+1})$ contains only a nonzero curl embedding $\tilde{\boldsymbol{x}}_{k,\mathrm{u}} = \boldsymbol{U}_{k,\mathrm{C}}^\top\boldsymbol{x}_{k,\mathrm{u}}$. The lower SCF $\boldsymbol{H}_{k,\mathrm{d}}$ has $\tilde{\boldsymbol{h}}_{k,\mathrm{d}} = \sum_{t=0}^{T_\mathrm{d}} w_{k,\mathrm{d},t}'\boldsymbol{\lambda}_{k,\mathrm{G}}^{\odot t}$ as the frequency response that modulates the gradient embedding of $\boldsymbol{x}_{k,\mathrm{d}}$ and the upper SCF $\boldsymbol{H}_{k,\mathrm{u}}$ has $\tilde{\boldsymbol{h}}_{k,\mathrm{u}} = \sum_{t=0}^{T_\mathrm{u}} w_{k,\mathrm{u},t}'\boldsymbol{\lambda}_{k,\mathrm{C}}^{\odot t}$ as the frequency response that modulates the curl embedding of $\boldsymbol{x}_{k,\mathrm{u}}$.

**SFT of $\boldsymbol{y}_k$.** For the output $\boldsymbol{y}_k = \boldsymbol{H}_{k,\mathrm{d}}\boldsymbol{x}_{k,\mathrm{d}} + \boldsymbol{H}_k\boldsymbol{x}_k + \boldsymbol{H}_{k,\mathrm{u}}\boldsymbol{x}_{k,\mathrm{u}}$, we have

$$\begin{cases} \tilde{\boldsymbol{y}}_{k,\mathrm{H}} = \tilde{\boldsymbol{h}}_{k,\mathrm{H}} \odot \tilde{\boldsymbol{x}}_{k,\mathrm{H}}, \\ \tilde{\boldsymbol{y}}_{k,\mathrm{G}} = \tilde{\boldsymbol{h}}_{k,\mathrm{d}} \odot \tilde{\boldsymbol{x}}_{k,\mathrm{d}} + \tilde{\boldsymbol{h}}_{k,\mathrm{G}} \odot \tilde{\boldsymbol{x}}_{k,\mathrm{G}}, \\ \tilde{\boldsymbol{y}}_{k,\mathrm{C}} = \tilde{\boldsymbol{h}}_{k,\mathrm{C}} \odot \tilde{\boldsymbol{x}}_{k,\mathrm{C}} + \tilde{\boldsymbol{h}}_{k,\mathrm{u}} \odot \tilde{\boldsymbol{x}}_{k,\mathrm{u}}. \end{cases} \tag{33}$$

### E.3 Expressive power in Proposition 13

*Proof.* From the Cayley-Hamilton theorem Horn & Johnson (2012), we know that an analytical function $f(\boldsymbol{A})$ of a matrix $\boldsymbol{A}$ can be expressed as a matrix polynomial of degree at most its minimal polynomial degree, which equals to the number of distinct eigenvalues if $\boldsymbol{A}$ is positive semi-definite.

Consider an analytical function $\boldsymbol{G}_{k,\mathrm{d}}$ of $\boldsymbol{L}_{k,\mathrm{d}}$, defined on the spectrum of $\boldsymbol{L}_{k,\mathrm{d}}$ via analytical function $g_{k,\mathrm{G}}(\lambda)$ where $\lambda$ is in the set of zero and the gradient frequencies. Then, $\boldsymbol{G}_{k,\mathrm{d}}$ can be implemented by a matrix polynomial of $\boldsymbol{L}_{k,\mathrm{d}}$ of order up to $n_{k,\mathrm{G}}$ where $n_{k,\mathrm{G}}$ is the number of nonzero eigenvalues of $\boldsymbol{L}_{k,\mathrm{d}}$, i.e., the number of distinct gradient frequencies. Likewise, any analytical function $\boldsymbol{G}_{k,\mathrm{u}}$ of $\boldsymbol{L}_{k,\mathrm{u}}$ can be implemented by a matrix polynomial of $\boldsymbol{L}_{k,\mathrm{u}}$ of order up to $n_{k,\mathrm{C}}$, which is the number of nonzero eigenvalues of $\boldsymbol{L}_{k,\mathrm{u}}$, i.e., the number of distinct curl frequencies.

Thus, as of the matrix polynomial definition of SCFs in a SCCNN, the expressive power of $\boldsymbol{H}_{k,\mathrm{d}}\boldsymbol{x}_{k,\mathrm{d}} + \boldsymbol{H}_k\boldsymbol{x}_k + \boldsymbol{H}_{k,\mathrm{u}}\boldsymbol{x}_{k,\mathrm{u}}$ is at most $\boldsymbol{G}'_{k,\mathrm{d}}\boldsymbol{x}_{k,\mathrm{d}} + (\boldsymbol{G}_{k,\mathrm{d}} + \boldsymbol{G}_{k,\mathrm{u}})\boldsymbol{x}_k + \boldsymbol{G}'_{k,\mathrm{u}}\boldsymbol{x}_{k,\mathrm{u}}$, when the matrix polynomial orders (convolution orders) follow $T_{k,\mathrm{d}} = T'_{k,\mathrm{d}} = n_{k,\mathrm{G}}$ and $T_{k,\mathrm{u}} = T'_{k,\mathrm{u}} = n_{k,\mathrm{C}}$. $\qquad\square$

### E.4 Hodge-aware of SCCNN in Theorem 14

*Proof.* Consider a linear mapping $T : V \to V$. An invariant subspace $W$ of $T$ has the property that all vectors $\boldsymbol{v} \in W$ are transformed by $T$ into vectors also contained in $W$, i.e., $\boldsymbol{v} \in W \implies T(\boldsymbol{v}) \in W$. For an input $\boldsymbol{x} \in \mathrm{im}(\boldsymbol{B}_k^\top)$, the output $\boldsymbol{H}_k\boldsymbol{x}$ is in $\mathrm{im}(\boldsymbol{B}_k^\top)$ too, because of

$$\boldsymbol{H}_k\boldsymbol{x} = \sum_t \boldsymbol{L}_{k,\mathrm{d}}^t\boldsymbol{x} + \sum_t \boldsymbol{L}_{k,\mathrm{u}}^t\boldsymbol{x} = \sum_t \boldsymbol{L}_{k,\mathrm{d}}^t\boldsymbol{x} \in \mathrm{im}(\boldsymbol{B}_k^\top) \tag{34}$$

where the second equality comes from the orthogonality between $\mathrm{im}(\boldsymbol{B}_k^\top)$ and $\mathrm{im}(\boldsymbol{B}_{k+1})$. Similarly, we can show that for $\boldsymbol{x} \in \mathrm{im}(\boldsymbol{B}_{k+1})$, the output $\boldsymbol{H}_k\boldsymbol{x} \in \mathrm{im}(\boldsymbol{B}_{k+1})$; for $\boldsymbol{x} \in \ker(\boldsymbol{L}_k)$, the output $\boldsymbol{H}_k\boldsymbol{x} \in \ker(\boldsymbol{L}_k)$. This essentially says the three subspaces of the Hodge decomposition are invariant with respect to the SCF $\boldsymbol{H}_k$. Likewise, the gradient space is invariant with respect to the lower SCF $\boldsymbol{H}_{k,\mathrm{d}}$, which says any lower projection remains in the gradient space after passed by $\boldsymbol{H}_{k,\mathrm{d}}$; and the curl space is invariant with respect to the upper SCF $\boldsymbol{H}_{k,\mathrm{u}}$.

Lastly, through the spectral relation in Eq. (11), the learning operator $\boldsymbol{H}_k$ in the gradient space is controlled by the learnable weights $\{w_{k,\mathrm{d},t}\}$, which is independent of the learnable weights $\{w_{k,\mathrm{u},t}\}$, associated to the learning of $\boldsymbol{H}_k$ in the curl space. Likewise, the lower SCF learns in the gradient space as well but with another set of learnable weights $\{w'_{k,\mathrm{d},t}\}$, and the upper SCF learns in the curl space with learnable weights $\{w'_{k,\mathrm{u},t}\}$. From the spectral expressive power, we see that above four independent learning in the two subspaces can be as expressive as any analytical functions of the corresponding frequencies (spectrum). This concludes the independent and expressive learning in the gradient and curl spaces. $\qquad\square$

## F Proofs for Section 5

We first give the formulation of SCCNNs on weighted SCs, then we proceed the stability proof.

### F.1 SCCNN on weighted SCs

A weighted SC can be defined through specifying the weights of simplices. We give the definition of a commonly used weighted SC with weighted Hodge Laplacians in Grady & Polimeni (2010); Horak & Jost (2013).

**Definition 31** (Weighted SC and Hodge Laplacians)**.** In an oriented and weighted SC, we have diagonal weighting matrices $\boldsymbol{M}_k$ with $[\boldsymbol{M}]_{ii}$ measuring the weight of $i$th $k$-simplex. A weighted $k$th Hodge Laplacian is given by

$$\tilde{\boldsymbol{L}}_k = \tilde{\boldsymbol{L}}_{k,\mathrm{d}} + \tilde{\boldsymbol{L}}_{k,\mathrm{u}} = \boldsymbol{M}_k\boldsymbol{B}_k^\top\boldsymbol{M}_{k-1}^{-1}\boldsymbol{B}_k + \boldsymbol{B}_{k+1}\boldsymbol{M}_{k+1}\boldsymbol{B}_{k+1}^\top\boldsymbol{M}_k^{-1}. \tag{35}$$

where $\boldsymbol{L}_{k,\mathrm{d}}$ and $\boldsymbol{L}_{k,\mathrm{u}}$ are the weighted lower and upper Laplacians. A symmetric version follows $\boldsymbol{L}_k^s = \boldsymbol{M}_k^{-1/2}\boldsymbol{L}_k\boldsymbol{M}_k^{1/2}$, and likewise, we have $\boldsymbol{L}_{k,\mathrm{d}}^s = \boldsymbol{M}_k^{1/2}\boldsymbol{B}_k^\top\boldsymbol{M}_{k-1}^{-1}\boldsymbol{B}_k\boldsymbol{M}_k^{1/2}$ and $\boldsymbol{L}_{k,\mathrm{u}}^s = \boldsymbol{M}_k^{-1/2}\boldsymbol{B}_{k+1}\boldsymbol{M}_{k+1}\boldsymbol{B}_{k+1}^\top\boldsymbol{M}_k^{-1/2}$, with the weighted incidence matrix is $\boldsymbol{M}_{k-1}^{-1/2}\boldsymbol{B}_k\boldsymbol{M}_k^{1/2}$ (Horak & Jost, 2013; Guglielmi et al., 2023; Schaub et al., 2020).

**SCCNNs in weighted SC.** The SCCNN layer defined in a weighted SC is of form

$$\boldsymbol{x}_k^l = \sigma(\boldsymbol{H}_{k,\mathrm{d}}^l\boldsymbol{R}_{k,\mathrm{d}}\boldsymbol{x}_{k-1}^{l-1} + \boldsymbol{H}_k^l\boldsymbol{x}_k^{l-1} + \boldsymbol{H}_{k,\mathrm{u}}^l\boldsymbol{R}_{k,\mathrm{u}}\boldsymbol{x}_{k+1}^{l-1}) \tag{36}$$

where the three SCFs are defined based on the weighted Laplacians Eq. (35), and the lower and upper contributions $\boldsymbol{x}_{k,\mathrm{d}}^l$ and $\boldsymbol{x}_{k,\mathrm{u}}^l$ are obtained via projection matrices $\boldsymbol{R}_{k,\mathrm{d}} \in \mathbb{R}^{n_k \times n_{k-1}}$ and $\boldsymbol{R}_{k,\mathrm{u}} \in \mathbb{R}^{n_k \times n_{k+1}}$, instead of $\boldsymbol{B}_k^\top$ and $\boldsymbol{B}_{k+1}$. For example, Bunch et al. (2020) considered $\boldsymbol{R}_{1,\mathrm{d}} = \boldsymbol{M}_1 \boldsymbol{B}_1^\top \boldsymbol{M}_0^{-1}$ and $\boldsymbol{R}_{1,\mathrm{u}} = \boldsymbol{B}_2 \boldsymbol{M}_2$.

### F.2 Proof of Stability of SCCNNs in Theorem 24

For a SCCNN in Eq. (36) in a weighted SC $\mathcal{S}$, we consider its perturbed version in a perturbed SC $\widehat{\mathcal{S}}$ at layer $l$, given by

$$\hat{\boldsymbol{x}}_k^l = \sigma(\widehat{\boldsymbol{H}}_{k,\mathrm{d}}^l \widehat{\boldsymbol{R}}_{k,\mathrm{d}} \hat{\boldsymbol{x}}_{k-1}^{l-1} + \widehat{\boldsymbol{H}}_k^l \hat{\boldsymbol{x}}_k^{l-1} + \widehat{\boldsymbol{H}}_{k,\mathrm{u}}^l \widehat{\boldsymbol{R}}_{k,\mathrm{u}} \hat{\boldsymbol{x}}_{k+1}^{l-1}) \tag{37}$$

which is defined based on perturbed Laplacians with the same set of filter coefficients, and the perturbed projection operators following relativ perturbation model.

Given the initial input $\boldsymbol{x}_k^0$ for $k = 0, 1, \ldots, K$, our goal is to upper bound the Euclidean distance between the outputs $\boldsymbol{x}_k^l$ and $\hat{\boldsymbol{x}}_k^l$ for $l = 1, \ldots, L$,

$$\begin{aligned}
\|\hat{\boldsymbol{x}}_k^l - \boldsymbol{x}_k^l\|_2 = \|\sigma(&\widehat{\boldsymbol{H}}_{k,\mathrm{d}}^l \widehat{\boldsymbol{R}}_{k,\mathrm{d}} \hat{\boldsymbol{x}}_{k-1}^{l-1} - \boldsymbol{H}_{k,\mathrm{d}}^l \boldsymbol{R}_{k,\mathrm{d}} \boldsymbol{x}_{k-1}^{l-1} \\
&+ \widehat{\boldsymbol{H}}_k^l \hat{\boldsymbol{x}}_k^{l-1} - \boldsymbol{H}_k^l \boldsymbol{x}_k^{l-1} + \widehat{\boldsymbol{H}}_{k,\mathrm{u}}^l \widehat{\boldsymbol{R}}_{k,\mathrm{u}} \hat{\boldsymbol{x}}_{k+1}^{l-1} - \boldsymbol{H}_{k,\mathrm{u}}^l \boldsymbol{R}_{k,\mathrm{u}} \boldsymbol{x}_{k+1}^{l-1})\|_2.
\end{aligned} \tag{38}$$

We proceed the proof in two steps: first, we analyze the operator norm $\|\widehat{\boldsymbol{H}}_k^l - \boldsymbol{H}_k^l\|_2$ of a SCF $\boldsymbol{H}_k^l$ and its perturbed version $\widehat{\boldsymbol{H}}_k^l$; then we look for the bound of the output distance for a general $L$-layer SCCNN. To ease notations, we omit the subscript such that $\|\boldsymbol{A}\| = \max_{\|\boldsymbol{x}\|_2=1} \|\boldsymbol{A}\boldsymbol{x}\|_2$ is the operator norm (spectral radius) of a matrix $\boldsymbol{A}$, and $\|\boldsymbol{x}\|$ is the Euclidean norm of a vector $\boldsymbol{x}$.

In the first step we omit the indices $k$ and $l$ for simplicity since they hold for general $k$ and $l$. We first give a useful lemma.

**Lemma 32.** *Given the ith eigenvector $\boldsymbol{u}_i$ of $\boldsymbol{L} = \boldsymbol{U}\boldsymbol{\Lambda}\boldsymbol{U}^\top$, for lower and upper perturbations $\boldsymbol{E}_\mathrm{d}$ and $\boldsymbol{E}_\mathrm{u}$, we have*

$$\boldsymbol{E}_\mathrm{d} \boldsymbol{u}_i = q_{\mathrm{d}i} \boldsymbol{u}_i + \boldsymbol{E}_1 \boldsymbol{u}_i, \quad \boldsymbol{E}_\mathrm{u} \boldsymbol{u}_i = q_{\mathrm{u}i} \boldsymbol{u}_i + \boldsymbol{E}_2 \boldsymbol{u}_i \tag{39}$$

*with eigendecompositions $\boldsymbol{E}_\mathrm{d} = \boldsymbol{V}_\mathrm{d} \boldsymbol{Q}_\mathrm{d} \boldsymbol{V}_\mathrm{d}^\top$ and $\boldsymbol{E}_\mathrm{u} = \boldsymbol{V}_\mathrm{u} \boldsymbol{Q}_\mathrm{u} \boldsymbol{V}_\mathrm{u}^\top$ where $\boldsymbol{V}_\mathrm{d}$, $\boldsymbol{V}_\mathrm{u}$ collect the eigenvectors and $\boldsymbol{Q}_\mathrm{d}$, $\boldsymbol{Q}_\mathrm{u}$ the eigenvalues. It holds that $\|\boldsymbol{E}_1\| \leq \epsilon_\mathrm{d} \delta_\mathrm{d}$ and $\|\boldsymbol{E}_2\| \leq \epsilon_\mathrm{u} \delta_\mathrm{u}$, with $\delta_\mathrm{d} = (\|\boldsymbol{V}_\mathrm{d} - \boldsymbol{U}\|+1)^2 - 1$ and $\delta_\mathrm{u} = (\|\boldsymbol{V}_\mathrm{u} - \boldsymbol{U}\|+1)^2 - 1$ measuring the eigenvector misalignments.*

*Proof.* We first prove that $\boldsymbol{E}_\mathrm{d} \boldsymbol{u}_i = q_{\mathrm{d}i} \boldsymbol{u}_i + \boldsymbol{E}_1 \boldsymbol{u}_i$. The perturbation matrix on the lower Laplacian can be written as $\boldsymbol{E}_\mathrm{d} = \boldsymbol{E}_\mathrm{d}' + \boldsymbol{E}_1$ with $\boldsymbol{E}_\mathrm{d}' = \boldsymbol{U} \boldsymbol{Q}_\mathrm{d} \boldsymbol{U}^\top$ and $\boldsymbol{E}_1 = (\boldsymbol{V}_\mathrm{d} - \boldsymbol{U}) \boldsymbol{Q}_\mathrm{d} (\boldsymbol{V}_\mathrm{d} - \boldsymbol{U})^\top + \boldsymbol{U} \boldsymbol{Q}_\mathrm{d} (\boldsymbol{V}_\mathrm{d} - \boldsymbol{U})^\top + (\boldsymbol{V}_\mathrm{d} - \boldsymbol{U}) \boldsymbol{Q}_\mathrm{d} \boldsymbol{U}^\top$. For the $i$th eigenvector $\boldsymbol{u}_i$, we have that

$$\boldsymbol{E}_\mathrm{d} \boldsymbol{u}_i = \boldsymbol{E}_\mathrm{d}' \boldsymbol{u}_i + \boldsymbol{E}_1 \boldsymbol{u}_i = q_{\mathrm{d}i} \boldsymbol{u}_i + \boldsymbol{E}_1 \boldsymbol{u}_i \tag{40}$$

where the second equality follows from $\boldsymbol{E}_\mathrm{d}' \boldsymbol{u}_i = q_{\mathrm{d}i} \boldsymbol{u}_i$. Since $\|\boldsymbol{E}_\mathrm{d}\| \leq \epsilon_\mathrm{d}$, it follows that $\|\boldsymbol{Q}_\mathrm{d}\| \leq \epsilon_\mathrm{d}$. Then, applying the triangle inequality, we have that

$$\begin{aligned}
\|\boldsymbol{E}_1\| &\leq \|(\boldsymbol{V}_\mathrm{d} - \boldsymbol{U}) \boldsymbol{Q}_\mathrm{d} (\boldsymbol{V}_\mathrm{d} - \boldsymbol{U})^\top\| + \|\boldsymbol{U} \boldsymbol{Q}_\mathrm{d} (\boldsymbol{V}_\mathrm{d} - \boldsymbol{U})^\top\| + \|(\boldsymbol{V}_\mathrm{d} - \boldsymbol{U}) \boldsymbol{Q}_\mathrm{d} \boldsymbol{U}\| \\
&\leq \|\boldsymbol{V}_\mathrm{d} - \boldsymbol{U}\|^2 \|\boldsymbol{Q}_\mathrm{d}\| + 2\|\boldsymbol{V}_\mathrm{d} - \boldsymbol{U}\| \|\boldsymbol{Q}_\mathrm{d}\| \|\boldsymbol{U}\| \leq \epsilon_\mathrm{d} \|\boldsymbol{V}_\mathrm{d} - \boldsymbol{U}\|^2 + 2\epsilon_\mathrm{d} \|\boldsymbol{V}_\mathrm{d} - \boldsymbol{U}\| \\
&= \epsilon_\mathrm{d}((\|\boldsymbol{V}_\mathrm{d} - \boldsymbol{U}\|+1)^2 - 1) = \epsilon_\mathrm{d} \delta_\mathrm{d},
\end{aligned} \tag{41}$$

which completes the proof for the lower perturbation matrix. Likewise, we can prove for $\boldsymbol{E}_\mathrm{u} \boldsymbol{u}_i$. $\qquad \square$

#### F.2.1 Step I: Stability of the SCF

*Proof.* **1. Low-order approximation of $\widehat{\boldsymbol{H}} - \boldsymbol{H}$.** Given a SCF $\boldsymbol{H} = \sum_{t=0}^{T_\mathrm{d}} w_{\mathrm{d},t} \boldsymbol{L}_\mathrm{d}^t + \sum_{t=0}^{T_\mathrm{u}} w_{\mathrm{u},t} \boldsymbol{L}_\mathrm{u}^t$, we denote its perturbed version by $\widehat{\boldsymbol{H}} = \sum_{t=0}^{T_\mathrm{d}} w_{\mathrm{d},t} \widehat{\boldsymbol{L}}_\mathrm{d}^t + \sum_{t=0}^{T_\mathrm{u}} w_{\mathrm{u},t} \widehat{\boldsymbol{L}}_\mathrm{u}^t$, where the filter coefficients are the same. The difference between $\boldsymbol{H}$ and $\widehat{\boldsymbol{H}}$ can be expressed as

$$\widehat{\boldsymbol{H}} - \boldsymbol{H} = \sum_{t=0}^{T_\mathrm{d}} w_{\mathrm{d},t} (\widehat{\boldsymbol{L}}_\mathrm{d}^t - \boldsymbol{L}_\mathrm{d}^t) + \sum_{t=0}^{T_\mathrm{u}} w_{\mathrm{u},t} (\widehat{\boldsymbol{L}}_\mathrm{u}^t - \boldsymbol{L}_\mathrm{u}^t), \tag{42}$$

in which we can compute the first-order Taylor expansion of $\widehat{\boldsymbol{L}}_{\mathrm{d}}^t$ as

$$\widehat{\boldsymbol{L}}_{\mathrm{d}}^t = (\boldsymbol{L}_{\mathrm{d}} + \boldsymbol{E}_{\mathrm{d}}\boldsymbol{L}_{\mathrm{d}} + \boldsymbol{L}_{\mathrm{d}}\boldsymbol{E}_{\mathrm{d}})^t = \boldsymbol{L}_{\mathrm{d}}^t + \boldsymbol{D}_{\mathrm{d},t} + \boldsymbol{C}_{\mathrm{d}} \tag{43}$$

with $\boldsymbol{D}_{\mathrm{d},t} := \sum_{r=0}^{t-1}(\boldsymbol{L}_{\mathrm{d}}^r\boldsymbol{E}_{\mathrm{d}}\boldsymbol{L}_{\mathrm{d}}^{t-r} + \boldsymbol{L}_{\mathrm{d}}^{r+1}\boldsymbol{E}_{\mathrm{d}}\boldsymbol{L}_{\mathrm{d}}^{t-r-1})$ parameterized by $t$ and $\boldsymbol{C}_{\mathrm{d}}$ following $\|\boldsymbol{C}_{\mathrm{d}}\| \le \sum_{r=2}^t \binom{t}{r}\|\boldsymbol{E}_{\mathrm{d}}\boldsymbol{L}_{\mathrm{d}} + \boldsymbol{L}_{\mathrm{d}}\boldsymbol{E}_{\mathrm{d}}\|^r\|\boldsymbol{L}_{\mathrm{d}}\|^{t-r}$. Likewise, we can expand $\widehat{\boldsymbol{L}}_{\mathrm{u}}^t$ as

$$\widehat{\boldsymbol{L}}_{\mathrm{u}}^t = (\boldsymbol{L}_{\mathrm{u}} + \boldsymbol{E}_{\mathrm{u}}\boldsymbol{L}_{\mathrm{d}} + \boldsymbol{L}_{\mathrm{d}}\boldsymbol{u})^t = \boldsymbol{L}_{\mathrm{u}}^t + \boldsymbol{D}_{\mathrm{u},t} + \boldsymbol{C}_{\mathrm{u}} \tag{44}$$

with $\boldsymbol{D}_{\mathrm{u},t} := \sum_{r=0}^{t-1}(\boldsymbol{L}_{\mathrm{u}}^r\boldsymbol{E}_{\mathrm{u}}\boldsymbol{L}_{\mathrm{u}}^{t-r} + \boldsymbol{L}_{\mathrm{u}}^{r+1}\boldsymbol{E}_{\mathrm{u}}\boldsymbol{L}_{\mathrm{u}}^{t-r-1})$ parameterized by $t$ and $\boldsymbol{C}_{\mathrm{u}}$ following $\|\boldsymbol{C}_{\mathrm{u}}\| \le \sum_{r=2}^t \binom{t}{r}\|\boldsymbol{E}_{\mathrm{u}}\boldsymbol{L}_{\mathrm{u}} + \boldsymbol{L}_{\mathrm{u}}\boldsymbol{E}_{\mathrm{u}}\|^r\|\boldsymbol{L}_{\mathrm{u}}\|^{t-r}$. Then, by substituting Eq. (43) and Eq. (44) into Eq. (42), we have

$$\widehat{\boldsymbol{H}} - \boldsymbol{H} = \sum_{t=0}^{T_{\mathrm{d}}} w_{\mathrm{d},t}\boldsymbol{D}_{\mathrm{d},t} + \sum_{t=0}^{T_{\mathrm{u}}} w_{\mathrm{u},t}\boldsymbol{D}_{\mathrm{u},t} + \boldsymbol{F}_{\mathrm{d}} + \boldsymbol{F}_{\mathrm{u}} \tag{45}$$

with negligible terms $\|\boldsymbol{F}_{\mathrm{d}}\| = \mathcal{O}(\|\boldsymbol{E}_{\mathrm{d}}\|^2)$ and $\|\boldsymbol{F}_{\mathrm{u}}\| = \mathcal{O}(\|\boldsymbol{E}_{\mathrm{u}}\|^2)$ because perturbations are small and the coefficients of higher-order power terms are the derivatives of analytic functions $\tilde{h}_{\mathrm{G}}(\lambda)$ and $\tilde{h}_{\mathrm{C}}(\lambda)$, which are bounded [cf. Definition 18].

**2. Spectrum of $(\widehat{\boldsymbol{H}} - \boldsymbol{H})\boldsymbol{x}$.** Consider a simplicial signal $\boldsymbol{x}$ with an SFT $\tilde{\boldsymbol{x}} = \boldsymbol{U}^\top\boldsymbol{x} = [\tilde{x}_1, \ldots, \tilde{x}_n]^\top$, thus, $\boldsymbol{x} = \sum_{i=1}^n \tilde{x}_i\boldsymbol{u}_i$. Then, we study the effect of the difference of the SCFs on a simplicial signal from the spectral perspective via

$$(\widehat{\boldsymbol{H}} - \boldsymbol{H})\boldsymbol{x} = \sum_{i=1}^n \tilde{x}_i \sum_{t=0}^{T_{\mathrm{d}}} w_{\mathrm{d},t}\boldsymbol{D}_{\mathrm{d},t}^t\boldsymbol{u}_i + \sum_{i=1}^n \tilde{x}_i \sum_{t=0}^{T_{\mathrm{d}}} w_{\mathrm{u},t}\boldsymbol{D}_{\mathrm{u},t}^t\boldsymbol{u}_i + \boldsymbol{F}_{\mathrm{d}}\boldsymbol{x} + \boldsymbol{F}_{\mathrm{u}}\boldsymbol{x} \tag{46}$$

where we have

$$\boldsymbol{D}_{\mathrm{d},t}^t\boldsymbol{u}_i = \sum_{r=0}^{t-1}(\boldsymbol{L}_{\mathrm{d}}^r\boldsymbol{E}_{\mathrm{d}}\boldsymbol{L}_{\mathrm{d}}^{t-r} + \boldsymbol{L}_{\mathrm{d}}^{r+1}\boldsymbol{E}_{\mathrm{d}}\boldsymbol{L}_{\mathrm{d}}^{t-r-1})\boldsymbol{u}_i, \text{ and } \boldsymbol{D}_{\mathrm{u},t}^t\boldsymbol{u}_i = \sum_{r=0}^{t-1}(\boldsymbol{L}_{\mathrm{u}}^r\boldsymbol{E}_{\mathrm{u}}\boldsymbol{L}_{\mathrm{u}}^{t-r} + \boldsymbol{L}_{\mathrm{u}}^{r+1}\boldsymbol{E}_{\mathrm{u}}\boldsymbol{L}_{\mathrm{u}}^{t-r-1})\boldsymbol{u}_i. \tag{47}$$

Since the lower and upper Laplacians admit the eigendecompositions for an eigenvector[2] $\boldsymbol{u}_i$

$$\boldsymbol{L}_{\mathrm{d}}\boldsymbol{u}_i = \lambda_{\mathrm{d}i}\boldsymbol{u}_i, \quad \boldsymbol{L}_{\mathrm{u}}\boldsymbol{u}_i = \lambda_{\mathrm{u}i}\boldsymbol{u}_i, \tag{48}$$

we can express the terms in Eq. (46) as

$$\boldsymbol{L}_{\mathrm{d}}^r\boldsymbol{E}_{\mathrm{d}}\boldsymbol{L}_{\mathrm{d}}^{t-r}\boldsymbol{u}_i = \boldsymbol{L}_{\mathrm{d}}^r\boldsymbol{E}_{\mathrm{d}}\lambda_{\mathrm{d}i}^{t-r}\boldsymbol{u}_i = \lambda_{\mathrm{d}i}^{t-r}\boldsymbol{L}_{\mathrm{d}}^r(q_{\mathrm{d}i}\boldsymbol{u}_i + \boldsymbol{E}_1\boldsymbol{u}_i) = q_{\mathrm{d}i}\lambda_{\mathrm{d}i}^{t-r}\boldsymbol{u}_i + \lambda_{\mathrm{d}i}^{t-r}\boldsymbol{L}_{\mathrm{d}}^r\boldsymbol{E}_1\boldsymbol{u}_i, \tag{49}$$

where the second equality holds from Lemma 32. Thus, we have

$$\boldsymbol{L}_{\mathrm{d}}^{r+1}\boldsymbol{E}_{\mathrm{d}}\boldsymbol{L}_{\mathrm{d}}^{t-r-1}\boldsymbol{u}_i = q_{\mathrm{d}i}\lambda_{\mathrm{d}i}^t\boldsymbol{u}_i + \lambda_{\mathrm{d}i}^{t-r-1}\boldsymbol{L}_{\mathrm{d}}^{r+1}\boldsymbol{E}_1\boldsymbol{u}_i. \tag{50}$$

With the results in Eq. (49) and Eq. (50), we can write the first term in Eq. (46) as

$$\sum_{i=1}^n \tilde{x}_i \sum_{t=0}^{T_{\mathrm{d}}} w_{\mathrm{d},t}\boldsymbol{D}_{\mathrm{d},t}^t\boldsymbol{u}_i = \underbrace{\sum_{i=1}^n \tilde{x}_i \sum_{t=0}^{T_{\mathrm{d}}} w_{\mathrm{d},t} \sum_{r=0}^{t-1} 2q_{\mathrm{d}i}\lambda_{\mathrm{d}i}^t\boldsymbol{u}_i}_{\text{term 1}}$$

$$+ \underbrace{\sum_{i=1}^n \tilde{x}_i \sum_{t=0}^{T_{\mathrm{d}}} w_{\mathrm{d},t} \sum_{r=0}^{t-1}(\lambda_{\mathrm{d}i}^{t-r}\boldsymbol{L}_{\mathrm{d}}^r\boldsymbol{E}_1\boldsymbol{u}_i + \lambda_{\mathrm{d}i}^{t-r-1}\boldsymbol{L}_{\mathrm{d}}^{r+1}\boldsymbol{E}_1\boldsymbol{u}_i)}_{\text{term 2}}. \tag{51}$$

---

[2]Note that they can be jointly diagonalized.

Term 1 can be further expanded as

$$\text{term } 1 = 2\sum_{i=1}^{n} \tilde{x}_i q_{\mathrm{d}i} \sum_{t=0}^{T_{\mathrm{d}}} t w_{\mathrm{d},t} \lambda_{\mathrm{d}i}^t \boldsymbol{u}_i = 2\sum_{i=1}^{n} \tilde{x}_i q_{\mathrm{d}i} \lambda_{\mathrm{d}i} \tilde{h}_{\mathrm{G}}'(\lambda_{\mathrm{d}i}) \boldsymbol{u}_i \tag{52}$$

where we used the fact that $\sum_{t=0}^{T_{\mathrm{d}}} t w_{\mathrm{d},t} \lambda_{\mathrm{d}i}^t = \lambda_{\mathrm{d}i} \tilde{h}_{\mathrm{G}}'(\lambda_{\mathrm{d}i})$. Using $\boldsymbol{L}_{\mathrm{d}} = \boldsymbol{U}\boldsymbol{\Lambda}_{\mathrm{d}}\boldsymbol{U}^{\top}$ we can write term 2 in Eq. (51) as

$$\text{term } 2 = \sum_{i=1}^{n} \tilde{x}_i \boldsymbol{U}\,\mathrm{diag}(\boldsymbol{g}_{\mathrm{d}i})\boldsymbol{U}^{\top}\boldsymbol{E}_1\boldsymbol{u}_i \tag{53}$$

where $\boldsymbol{g}_{\mathrm{d}i} \in \mathbb{R}^n$ has the $j$th entry

$$[\boldsymbol{g}_{\mathrm{d}i}]_j = \sum_{t=0}^{T_{\mathrm{d}}} w_{\mathrm{d},t} \sum_{r=0}^{t-1} \left( \lambda_{\mathrm{d}i}^{t-r}[\boldsymbol{\Lambda}_{\mathrm{d}}]_j^r + \lambda_{\mathrm{d}i}^{t-r-1}[\boldsymbol{\Lambda}_{\mathrm{d}}]_j^{r+1} \right) = \begin{cases} 2\lambda_{\mathrm{d}i}\tilde{h}_{\mathrm{G}}'(\lambda_{\mathrm{d}i}) & \text{for } j = i, \\ \frac{\lambda_{\mathrm{d}i}+\lambda_{\mathrm{d}j}}{\lambda_{\mathrm{d}i}-\lambda_{\mathrm{d}j}}(\tilde{h}_{\mathrm{G}}(\lambda_{\mathrm{d}i}) - \tilde{h}_{\mathrm{G}}(\lambda_{\mathrm{d}j})) & \text{for } j \neq i. \end{cases} \tag{54}$$

Now, substituting Eq. (52) and Eq. (53) into Eq. (51), we have

$$\sum_{i=1}^{n} \tilde{x}_i \sum_{t=0}^{T_{\mathrm{d}}} w_{\mathrm{d},t} \boldsymbol{D}_{\mathrm{d},t}^t \boldsymbol{u}_i = 2\sum_{i=1}^{n} \tilde{x}_i q_{\mathrm{d}i} \lambda_{\mathrm{d}i} \tilde{h}_{\mathrm{G}}'(\lambda_{\mathrm{d}i}) \boldsymbol{u}_i + \sum_{i=1}^{n} \tilde{x}_i \boldsymbol{U}\,\mathrm{diag}(\boldsymbol{g}_{\mathrm{d}i})\boldsymbol{U}^{\top}\boldsymbol{E}_1\boldsymbol{u}_i. \tag{55}$$

By following the same steps as in Eq. (51)-Eq. (54), we can express also the second term in Eq. (46) as

$$\sum_{i=1}^{n} \tilde{x}_i \sum_{t=0}^{T_{\mathrm{d}}} w_{\mathrm{u},t} \boldsymbol{D}_{\mathrm{u},t}^t \boldsymbol{u}_i = 2\sum_{i=1}^{n} \tilde{x}_i q_{\mathrm{u}i} \lambda_{\mathrm{u}i} \tilde{h}_{\mathrm{C}}'(\lambda_{\mathrm{u}i}) \boldsymbol{u}_i + \sum_{i=1}^{n} \tilde{x}_i \boldsymbol{U}\,\mathrm{diag}(\boldsymbol{g}_{\mathrm{u}i})\boldsymbol{U}^{\top}\boldsymbol{E}_2\boldsymbol{u}_i \tag{56}$$

where $\boldsymbol{g}_{\mathrm{u}i} \in \mathbb{R}^n$ is defined as

$$[\boldsymbol{g}_{\mathrm{u}i}]_j = \sum_{t=0}^{T_{\mathrm{d}}} w_{\mathrm{u},t} \sum_{r=0}^{t-1} \left( \lambda_{\mathrm{u}i}^{t-r}[\boldsymbol{\Lambda}_{\mathrm{u}}]_j^r + \lambda_{\mathrm{u}i}^{t-r-1}[\boldsymbol{\Lambda}_{\mathrm{u}}]_j^{r+1} \right) = \begin{cases} 2\lambda_{\mathrm{u}i}\tilde{h}_{\mathrm{C}}'(\lambda_{\mathrm{u}i}) & \text{for } j = i, \\ \frac{\lambda_{\mathrm{u}i}+\lambda_{\mathrm{u}j}}{\lambda_{\mathrm{u}i}-\lambda_{\mathrm{u}j}}(\tilde{h}_{\mathrm{C}}(\lambda_{\mathrm{u}i}) - \tilde{h}_{\mathrm{C}}(\lambda_{\mathrm{u}j})) & \text{for } j \neq i. \end{cases} \tag{57}$$

**3. Bound of** $\|(\widehat{\boldsymbol{H}} - \boldsymbol{H})\boldsymbol{x}\|$. Now we are ready to bound $\|(\widehat{\boldsymbol{H}} - \boldsymbol{H})\boldsymbol{x}\|$ based on triangle inequality. First, given the small perturbations $\|\boldsymbol{E}_{\mathrm{d}}\| \leq \epsilon_{\mathrm{d}}$ and $\|\boldsymbol{E}_{\mathrm{u}}\| \leq \epsilon_{\mathrm{u}}$, we have for the last two terms in Eq. (46)

$$\|\boldsymbol{F}_{\mathrm{d}}\boldsymbol{x}\| \leq \mathcal{O}(\epsilon_{\mathrm{d}}^2)\|\boldsymbol{x}\|, \text{ and } \|\boldsymbol{F}_{\mathrm{u}}\boldsymbol{x}\| \leq \mathcal{O}(\epsilon_{\mathrm{u}}^2)\|\boldsymbol{x}\|. \tag{58}$$

Second, for the first term $\|\sum_{i=1}^{n} \tilde{x}_i \sum_{t=0}^{T_{\mathrm{d}}} w_{\mathrm{d},t} \boldsymbol{D}_{\mathrm{d}}^t \boldsymbol{u}_i\|$ in Eq. (46), we can bound its two terms in Eq. (52) and Eq. (53) as

$$\left\| \sum_{i=1}^{n} \tilde{x}_i \sum_{t=0}^{T_{\mathrm{d}}} w_{\mathrm{d},t} \boldsymbol{D}_{\mathrm{d},t}^t \boldsymbol{u}_i \right\| \leq \left\| 2\sum_{i=1}^{n} \tilde{x}_i q_{\mathrm{d}i} \lambda_{\mathrm{d}i} \tilde{h}_{\mathrm{G}}'(\lambda_{\mathrm{d}i}) \boldsymbol{u}_i \right\| + \left\| \sum_{i=1}^{n} \tilde{x}_i \boldsymbol{U}\,\mathrm{diag}(\boldsymbol{g}_{\mathrm{d}i})\boldsymbol{U}^{\top}\boldsymbol{E}_1\boldsymbol{u}_i \right\|. \tag{59}$$

For the first term on the RHS of Eq. (59), we can write

$$\left\| 2\sum_{i=1}^{n} \tilde{x}_i q_{\mathrm{d}i} \lambda_{\mathrm{d}i} \tilde{h}_{\mathrm{G}}'(\lambda_{\mathrm{d}i}) \boldsymbol{u}_i \right\|^2 \leq 4\sum_{i=1}^{n} |\tilde{x}_i|^2 |q_{\mathrm{d}i}|^2 |\lambda_{\mathrm{d}i}\tilde{h}_{\mathrm{G}}'(\lambda_{\mathrm{d}i})|^2 \leq 4\epsilon_d^2 c_{\mathrm{d}}^2 \|\boldsymbol{x}\|^2, \tag{60}$$

which results from, first, $|q_{\mathrm{d}i}| \leq \epsilon_{\mathrm{d}} = \|\boldsymbol{E}_{\mathrm{d}}\|$ since $q_{\mathrm{d}i}$ is an eigenvalue of $\boldsymbol{E}_{\mathrm{d}}$; second, the integral Lipschitz property of the SCF $|\lambda\tilde{h}_{\mathrm{G}}'(\lambda)| \leq c_{\mathrm{d}}$; and lastly, the fact that $\sum_{i=1}^{n} |\tilde{x}_i|^2 = \|\tilde{\boldsymbol{x}}\|^2 = \|\boldsymbol{x}\|^2$ and $\|\boldsymbol{u}_i\|^2 = 1$. We then have

$$\left\| 2\sum_{i=1}^{n} \tilde{x}_i q_{\mathrm{d}i} \lambda_{\mathrm{d}i} \tilde{h}_{\mathrm{G}}'(\lambda_{\mathrm{d}i}) \boldsymbol{u}_i \right\| \leq 2\epsilon_{\mathrm{d}} c_{\mathrm{d}} \|\boldsymbol{x}\|. \tag{61}$$

For the second term in RHS of Eq. (59), we have

$$\left\|\sum_{i=1}^{n} \tilde{x}_i \boldsymbol{U} \operatorname{diag}(\boldsymbol{g}_{\mathrm{d}i}) \boldsymbol{U}^{\top} \boldsymbol{E}_1 \boldsymbol{u}_i\right\| \leq \sum_{i=1}^{n} |\tilde{x}_i| \|\boldsymbol{U} \operatorname{diag}(\boldsymbol{g}_{\mathrm{d}i}) \boldsymbol{U}^{\top}\| \|\boldsymbol{E}_1\| \|\boldsymbol{u}_i\|, \tag{62}$$

which stems from the triangle inequality. We further have $\|\boldsymbol{U} \operatorname{diag}(\boldsymbol{g}_{\mathrm{d}i}) \boldsymbol{U}^{\top}\| = \|\operatorname{diag}(\boldsymbol{g}_{\mathrm{d}i})\| \leq 2C_{\mathrm{d}}$ resulting from $\|\boldsymbol{U}\| = 1$ and the $c_{\mathrm{d}}$-integral Lipschitz of $\tilde{h}_{\mathrm{G}}(\lambda)$ [cf. Definition 18]. Moreover, it follows that $\|\boldsymbol{E}_1\| \leq \epsilon_{\mathrm{d}} \delta_{\mathrm{d}}$ from Lemma 32, which results in

$$\left\|\sum_{i=1}^{n} \tilde{x}_i \boldsymbol{U} \operatorname{diag}(\boldsymbol{g}_{\mathrm{d}i}) \boldsymbol{U}^{\top} \boldsymbol{E}_1 \boldsymbol{u}_i\right\| \leq 2C_{\mathrm{d}} \epsilon_{\mathrm{d}} \delta_{\mathrm{d}} \sqrt{n} \|\boldsymbol{x}\| \tag{63}$$

where we use that $\sum_{i=1}^{n} |\tilde{x}_i| = \|\tilde{\boldsymbol{x}}\|_1 \leq \sqrt{n} \|\tilde{\boldsymbol{x}}\| = \sqrt{n} \|\boldsymbol{x}\|$. By combining Eq. (60) and Eq. (63), we have

$$\left\|\sum_{i=1}^{n} \tilde{x}_i \sum_{t=0}^{T_{\mathrm{d}}} w_{\mathrm{d},t} \boldsymbol{D}_{\mathrm{d},t}^{t} \boldsymbol{u}_i\right\| \leq 2\epsilon_{\mathrm{d}} c_{\mathrm{d}} \|\boldsymbol{x}\| + 2C_{\mathrm{d}} \epsilon_{\mathrm{d}} \delta_{\mathrm{d}} \sqrt{n} \|\boldsymbol{x}\|. \tag{64}$$

Analogously, we can show that

$$\left\|\sum_{i=1}^{n} \tilde{x}_i \sum_{t=0}^{T_{\mathrm{d}}} w_{\mathrm{u},t} \boldsymbol{D}_{\mathrm{u},t}^{t} \boldsymbol{u}_i\right\| \leq 2\epsilon_{\mathrm{u}} c_{\mathrm{u}} \|\boldsymbol{x}\| + 2C_{\mathrm{u}} \epsilon_{\mathrm{u}} \delta_{\mathrm{u}} \sqrt{n} \|\boldsymbol{x}\|. \tag{65}$$

Now by combining Eq. (58), Eq. (64) and Eq. (65), we can bound $\|(\widehat{\boldsymbol{H}} - \boldsymbol{H})\boldsymbol{x}\|$ as

$$\begin{aligned}
\|(\widehat{\boldsymbol{H}} - \boldsymbol{H})\boldsymbol{x}\| &\leq 2\epsilon_{\mathrm{d}} c_{\mathrm{d}} \|\boldsymbol{x}\| + 2C_{\mathrm{d}} \epsilon_{\mathrm{d}} \delta_{\mathrm{d}} \sqrt{n} \|\boldsymbol{x}\| + \mathcal{O}(\epsilon_{\mathrm{d}}^2) \|\boldsymbol{x}\| \\
&\quad + 2\epsilon_{\mathrm{u}} c_{\mathrm{u}} \|\boldsymbol{x}\| + 2C_{\mathrm{u}} \epsilon_{\mathrm{u}} \delta_{\mathrm{u}} \sqrt{n} \|\boldsymbol{x}\| + \mathcal{O}(\epsilon_{\mathrm{u}}^2) \|\boldsymbol{x}\|.
\end{aligned} \tag{66}$$

By defining $\Delta_{\mathrm{d}} = 2(1 + \delta_{\mathrm{d}} \sqrt{n})$ and $\Delta_{\mathrm{u}} = 2(1 + \delta_{\mathrm{u}} \sqrt{n})$, we can obtain that

$$\|\widehat{\boldsymbol{H}} - \boldsymbol{H}\| \leq c_{\mathrm{d}} \Delta_{\mathrm{d}} \epsilon_{\mathrm{d}} + c_{\mathrm{u}} \Delta_{\mathrm{u}} \epsilon_{\mathrm{u}} + \mathcal{O}(\epsilon_{\mathrm{d}}^2) + \mathcal{O}(\epsilon_{\mathrm{u}}^2). \tag{67}$$

Thus, we have $\|\boldsymbol{H}_k^l - \widehat{\boldsymbol{H}}_k^l\| \leq c_{k,\mathrm{d}} \Delta_{k,\mathrm{d}} \epsilon_{k,\mathrm{d}} + c_{k,\mathrm{u}} \Delta_{k,\mathrm{u}} \epsilon_{k,\mathrm{u}}$ with $\Delta_{k,\mathrm{d}} = 2(1 + \delta_{k,\mathrm{d}} \sqrt{n_k})$ and $\Delta_{k,\mathrm{u}} = 2(1 + \delta_{k,\mathrm{u}} \sqrt{n_k})$ where we ignore the second and higher order terms on $\epsilon_{k,\mathrm{d}}$ and $\epsilon_{k,\mathrm{u}}$. Likewise, we have $\|\boldsymbol{H}_{k,\mathrm{d}}^l - \widehat{\boldsymbol{H}}_{k,\mathrm{d}}^l\| \leq c_{k,\mathrm{d}} \Delta_{k,\mathrm{d}} \epsilon_{k,\mathrm{d}}$ for the lower SCF and $\|\boldsymbol{H}_{k,\mathrm{u}}^l - \widehat{\boldsymbol{H}}_{k,\mathrm{u}}^l\| \leq c_{k,\mathrm{u}} \Delta_{k,\mathrm{u}} \epsilon_{k,\mathrm{u}}$ for the upper SCF. $\square$

### F.2.2 Step II: Stability of SCCNNs

*Proof.* Given the initial input $\boldsymbol{x}_k^0$, the Euclidean distance between $\boldsymbol{x}_k^l$ and $\hat{\boldsymbol{x}}_k^l$ at layer $l$ can be bounded by using triangle inequality and the $c_\sigma$-Lipschitz property of $\sigma(\cdot)$ [cf. Assumption 22] as

$$\|\hat{\boldsymbol{x}}_k^l - \boldsymbol{x}_k^l\|_2 \leq c_\sigma(\phi_{k,\mathrm{d}}^l + \phi_k^l + \phi_{k,\mathrm{u}}^l), \tag{68}$$

with

$$\begin{aligned}
\phi_{k,\mathrm{d}}^l &:= \|\widehat{\boldsymbol{H}}_{k,\mathrm{d}}^l \widehat{\boldsymbol{R}}_{k,\mathrm{d}} \hat{\boldsymbol{x}}_{k-1}^{l-1} - \boldsymbol{H}_{k,\mathrm{d}}^l \boldsymbol{R}_{k,\mathrm{d}} \boldsymbol{x}_{k-1}^{l-1}\|, \\
\phi_k^l &:= \|\widehat{\boldsymbol{H}}_k^l \hat{\boldsymbol{x}}_k^{l-1} - \boldsymbol{H}_k^l \boldsymbol{x}_k^{l-1}\|, \\
\phi_{k,\mathrm{u}}^l &:= \|\widehat{\boldsymbol{H}}_{k,\mathrm{u}}^l \widehat{\boldsymbol{R}}_{k,\mathrm{u}} \hat{\boldsymbol{x}}_{k+1}^{l-1} - \boldsymbol{H}_{k,\mathrm{u}}^l \boldsymbol{R}_{k,\mathrm{u}} \boldsymbol{x}_{k+1}^{l-1}\|.
\end{aligned} \tag{69}$$

We now focus on upper bounding each of the terms.

**1. Term $\phi_k^l$.** By subtracting and adding $\widehat{\boldsymbol{H}}_k^l \boldsymbol{x}_k^{l-1}$ within the norm, and using the triangle inequality, we obtain

$$\begin{aligned}
\phi_k^l &\leq \|\widehat{\boldsymbol{H}}_k^l(\hat{\boldsymbol{x}}_k^{l-1} - \boldsymbol{x}_k^{l-1})\| + \|(\widehat{\boldsymbol{H}}_k^l - \boldsymbol{H}_k^l)\boldsymbol{x}_k^{l-1}\| \leq \|\hat{\boldsymbol{x}}_k^{l-1} - \boldsymbol{x}_k^{l-1}\| + \|\widehat{\boldsymbol{H}}_k^l - \boldsymbol{H}_k^l\| \|\boldsymbol{x}_k^{l-1}\| \\
&\leq \|\hat{\boldsymbol{x}}_k^{l-1} - \boldsymbol{x}_k^{l-1}\| + (c_{k,\mathrm{d}} \Delta_{k,\mathrm{d}} \epsilon_{k,\mathrm{d}} + c_{k,\mathrm{u}} \Delta_{k,\mathrm{u}} \epsilon_{k,\mathrm{u}}) \|\boldsymbol{x}_k^{l-1}\|
\end{aligned} \tag{70}$$

where we used the SCF stability in Eq. (67) and that all SCFs have a normalized bounded frequency response in Assumption 20. Note that $\widehat{\boldsymbol{H}}_k^l$ is also characterized by $\tilde{h}_{\mathrm{G}}(\lambda)$ with the same set of filter coefficients as $\boldsymbol{H}_k^l$.

**2. Term $\phi_{k,\mathrm{d}}^l$ and $\phi_{k,\mathrm{u}}^l$.** By subtracting and adding a term $\widehat{\boldsymbol{H}}_{k,\mathrm{d}}^l \widehat{\boldsymbol{R}}_{k,\mathrm{d}} \boldsymbol{x}_{k-1}^{l-1}$ within the norm, we have

$$
\begin{aligned}
\phi_{k,\mathrm{d}}^l &\leq \|\widehat{\boldsymbol{H}}_{k,\mathrm{d}}^l \widehat{\boldsymbol{R}}_{k,\mathrm{d}} (\hat{\boldsymbol{x}}_{k-1}^{l-1} - \boldsymbol{x}_{k-1}^{l-1})\| + \|(\widehat{\boldsymbol{H}}_{k,\mathrm{d}}^l \widehat{\boldsymbol{R}}_{k,\mathrm{d}} - \boldsymbol{H}_{k,\mathrm{d}}^l \boldsymbol{R}_{k,\mathrm{d}}) \boldsymbol{x}_{k-1}^{l-1}\| \\
&\leq \|\widehat{\boldsymbol{R}}_{k,\mathrm{d}}\| \|\hat{\boldsymbol{x}}_{k-1}^{l-1} - \boldsymbol{x}_{k-1}^{l-1}\| + \|\widehat{\boldsymbol{H}}_{k,\mathrm{d}}^l \widehat{\boldsymbol{R}}_{k,\mathrm{d}} - \boldsymbol{H}_{k,\mathrm{d}}^l \boldsymbol{R}_{k,\mathrm{d}}\| \|\boldsymbol{x}_{k-1}^{l-1}\|,
\end{aligned}
\tag{71}
$$

where we used again triangle inequality and $\|\widehat{\boldsymbol{H}}_{k,\mathrm{d}}^l\| \leq 1$ from Assumption 20. For the term $\|\widehat{\boldsymbol{R}}_{k,\mathrm{d}}^l\|$, we have $\|\widehat{\boldsymbol{R}}_{k,\mathrm{d}}^l\| \leq \|\boldsymbol{R}_{k,\mathrm{d}}^l\| + \|\boldsymbol{J}_{k,\mathrm{d}}\| \|\boldsymbol{R}_{k,\mathrm{d}}^l\| \leq r_{k,\mathrm{d}}(1 + \boldsymbol{\epsilon}_{k,\mathrm{d}})$ where we used $\|\boldsymbol{R}_{k,\mathrm{d}}^l\| \leq r_{k,\mathrm{d}}$ in Assumption 21 and $\|\boldsymbol{J}_{k,\mathrm{d}}^l\| \leq \boldsymbol{\epsilon}_{k,\mathrm{d}}$. For the second term of RHS in Eq. (71), by adding and subtracting $\widehat{\boldsymbol{H}}_{k,\mathrm{d}}^l \boldsymbol{R}_{k,\mathrm{d}}^l$ we have

$$
\begin{aligned}
\|\widehat{\boldsymbol{H}}_{k,\mathrm{d}}^l \widehat{\boldsymbol{R}}_{k,\mathrm{d}} - \boldsymbol{H}_{k,\mathrm{d}}^l \boldsymbol{R}_{k,\mathrm{d}}\| &= \|\widehat{\boldsymbol{H}}_{k,\mathrm{d}}^l \widehat{\boldsymbol{R}}_{k,\mathrm{d}} - \widehat{\boldsymbol{H}}_{k,\mathrm{d}}^l \boldsymbol{R}_{k,\mathrm{d}}^l + \widehat{\boldsymbol{H}}_{k,\mathrm{d}}^l \boldsymbol{R}_{k,\mathrm{d}}^l - \boldsymbol{H}_{k,\mathrm{d}}^l \boldsymbol{R}_{k,\mathrm{d}}\| \\
&\leq \|\widehat{\boldsymbol{H}}_{k,\mathrm{d}}^l\| \|\widehat{\boldsymbol{R}}_{k,\mathrm{d}} - \boldsymbol{R}_{k,\mathrm{d}}\| + \|\widehat{\boldsymbol{H}}_{k,\mathrm{d}}^l - \boldsymbol{H}_{k,\mathrm{d}}^l\| \|\boldsymbol{R}_{k,\mathrm{d}}\| \\
&\leq r_{k,\mathrm{d}} \boldsymbol{\epsilon}_{k,\mathrm{d}} + C_{k,\mathrm{d}}' \Delta_{k,\mathrm{d}} \boldsymbol{\epsilon}_{k,\mathrm{d}} r_{k,\mathrm{d}}
\end{aligned}
\tag{72}
$$

where we use the stability result of the lower SCF $\boldsymbol{H}_{k,\mathrm{d}}^l$ in Eq. (67). By substituting Eq. (72) into Eq. (71), we have

$$
\phi_{k,\mathrm{d}}^l \leq \hat{r}_{k,\mathrm{d}} \|\hat{\boldsymbol{x}}_{k-1}^{l-1} - \boldsymbol{x}_{k-1}^{l-1}\| + (r_{k,\mathrm{d}} \boldsymbol{\epsilon}_{k,\mathrm{d}} + C_{k,\mathrm{d}}' \Delta_{k,\mathrm{d}} \boldsymbol{\epsilon}_{k,\mathrm{d}} r_{k,\mathrm{d}}) \|\boldsymbol{x}_{k-1}^{l-1}\|.
\tag{73}
$$

By following the same procedure [cf. Eq. (71) and Eq. (72)], we obtain

$$
\phi_{k,\mathrm{u}}^l \leq \hat{r}_{k,\mathrm{u}} \|\hat{\boldsymbol{x}}_{k+1}^{l-1} - \boldsymbol{x}_{k+1}^{l-1}\| + (r_{k,\mathrm{u}} \boldsymbol{\epsilon}_{k,\mathrm{u}} + C_{k,\mathrm{u}}' \Delta_{k,\mathrm{u}} \boldsymbol{\epsilon}_{k,\mathrm{u}} r_{k,\mathrm{u}}) \|\boldsymbol{x}_{k+1}^{l-1}\|.
\tag{74}
$$

**3. Bound of $\|\hat{\boldsymbol{x}}_k^l - \boldsymbol{x}_k^l\|$.** Using the notations $t_k, t_{k,\mathrm{d}}$ and $t_{k,\mathrm{u}}$ in Theorem 24, we then have a set of recursions, for $k = 0, 1, \ldots, K$

$$
\begin{aligned}
\|\hat{\boldsymbol{x}}_k^l - \boldsymbol{x}_k^l\| \leq &c_\sigma (\hat{r}_{k,\mathrm{d}} \|\hat{\boldsymbol{x}}_{k-1}^{l-1} - \boldsymbol{x}_{k-1}^{l-1}\| + t_{k,\mathrm{d}} \|\boldsymbol{x}_{k-1}^{l-1}\| + \|\hat{\boldsymbol{x}}_k^{l-1} - \boldsymbol{x}_k^{l-1}\| + t_k \|\boldsymbol{x}_k^{l-1}\| \\
&+ \hat{r}_{k,\mathrm{u}} \|\hat{\boldsymbol{x}}_{k+1}^{l-1} - \boldsymbol{x}_{k+1}^{l-1}\| + t_{k,\mathrm{u}} \|\boldsymbol{x}_{k+1}^{l-1}\|).
\end{aligned}
\tag{75}
$$

Define vector $\boldsymbol{b}^l$ as $[\boldsymbol{b}^l]_k = \|\hat{\boldsymbol{x}}_k^l - \boldsymbol{x}_k^l\|$ with $\boldsymbol{b}^0 = \boldsymbol{0}$. Let $\boldsymbol{\beta}^l$ collect the energy of all outputs at layer $l$, with $[\boldsymbol{\beta}^l]_k := \|\boldsymbol{x}_k^{l-1}\|$. We can express the Euclidean distances of all $k$-simplicial signal outputs for $k = 0, 1, \ldots, K$, as

$$
\boldsymbol{b}^l \preceq c_\sigma \widehat{\boldsymbol{Z}} \boldsymbol{b}^{l-1} + c_\sigma \boldsymbol{T} \boldsymbol{\beta}^{l-1}
\tag{76}
$$

where $\preceq$ indicates elementwise smaller than or equal, and we have

$$
\boldsymbol{T} = \begin{bmatrix} t_0 & t_{0,\mathrm{u}} & & & \\ t_{1,\mathrm{d}} & t_1 & t_{1,\mathrm{u}} & & \\ & \ddots & \ddots & \ddots & \\ & & t_{K-1,\mathrm{d}} & t_{K-1} & t_{K-1,\mathrm{u}} \\ & & & t_{K,\mathrm{d}} & t_K \end{bmatrix} \quad \text{and} \quad \widehat{\boldsymbol{Z}} = \begin{bmatrix} 1 & \hat{r}_{0,\mathrm{u}} & & & \\ \hat{r}_{1,\mathrm{d}} & 1 & \hat{r}_{1,\mathrm{u}} & & \\ & \ddots & \ddots & \ddots & \\ & & \hat{r}_{K-1,\mathrm{d}} & 1 & \hat{r}_{K-1,\mathrm{u}} \\ & & & \hat{r}_{K,\mathrm{d}} & 1 \end{bmatrix}.
\tag{77}
$$

We are now interested in building a recursion for Eq. (76) for all layers $l$. We start with term $\boldsymbol{x}_k^l$. Based on its expression in Eq. (36), we bound it as

$$
\begin{aligned}
\|\boldsymbol{x}_k^l\| &\leq c_\sigma (\|\boldsymbol{H}_{k,\mathrm{d}}^l\| \|\boldsymbol{R}_{k,\mathrm{d}}\| \|\boldsymbol{x}_{k-1}^{l-1}\| + \|\boldsymbol{H}_k^l\| \|\boldsymbol{x}_x^{l-1}\| + \|\boldsymbol{H}_{k,\mathrm{u}}^l\| \|\boldsymbol{R}_{k,\mathrm{u}}\| \|\boldsymbol{x}_{k+1}^{l-1}\|) \\
&\leq c_\sigma (r_{k,\mathrm{d}} \|\boldsymbol{x}_{k-1}^{l-1}\| + \|\boldsymbol{x}_x^{l-1}\| + r_{k,\mathrm{u}} \|\boldsymbol{x}_{k+1}^{l-1}\|),
\end{aligned}
\tag{78}
$$

which holds for $k = 0, 1, \ldots, K$. Thus, it can be expressed in the vector form as $\boldsymbol{\beta}^l \preceq c_\sigma \boldsymbol{Z} \boldsymbol{\beta}^{l-1}$, with

$$
\boldsymbol{Z} = \begin{bmatrix} 1 & r_{0,\mathrm{u}} & & & \\ r_{1,\mathrm{d}} & 1 & r_{1,\mathrm{u}} & & \\ & \ddots & \ddots & \ddots & \\ & & r_{K-1,\mathrm{d}} & 1 & r_{K-1,\mathrm{u}} \\ & & & r_{K,\mathrm{d}} & 1 \end{bmatrix}.
\tag{79}
$$

Similarly, we have $\boldsymbol{\beta}^{l-1} \preceq c_\sigma \boldsymbol{Z} \boldsymbol{\beta}^{l-2}$, leading to $\boldsymbol{\beta}^l \preceq c_\sigma^l \boldsymbol{Z}^l \boldsymbol{\beta}^0$ with $\boldsymbol{\beta}^0 = \boldsymbol{\beta}$ [cf. Assumption 23]. We can then express the bound Eq. (76) as

$$\boldsymbol{b}^l \preceq c_\sigma \widehat{\boldsymbol{Z}} \boldsymbol{b}^{l-1} + c_\sigma^l \boldsymbol{T} \boldsymbol{Z}^{l-1} \boldsymbol{\beta}. \tag{80}$$

Thus, we have

$$\boldsymbol{b}^0 = \boldsymbol{0}, \ \boldsymbol{b}^1 \preceq c_\sigma \boldsymbol{T} \boldsymbol{\beta}, \ \boldsymbol{b}^2 \preceq c_\sigma^2 (\widehat{\boldsymbol{Z}} \boldsymbol{T} \boldsymbol{\beta} + \boldsymbol{T} \boldsymbol{Z} \boldsymbol{\beta}), \ \boldsymbol{b}^3 \preceq c_\sigma^3 (\widehat{\boldsymbol{Z}}^2 \boldsymbol{T} \boldsymbol{\beta} + \widehat{\boldsymbol{Z}} \boldsymbol{T} \boldsymbol{Z} \boldsymbol{\beta} + \boldsymbol{T} \boldsymbol{Z}^2 \boldsymbol{\beta}), \ \boldsymbol{b}^4 \preceq \dots, \tag{81}$$

which, inductively, leads to

$$\boldsymbol{b}^l \preceq c_\sigma^l \sum_{i=1}^l \widehat{\boldsymbol{Z}}^{i-1} \boldsymbol{T} \boldsymbol{Z}^{l-i} \boldsymbol{\beta}. \tag{82}$$

Bt setting $l = L$, we obtain the bound $\boldsymbol{b}^L \preceq \boldsymbol{d} = c_\sigma^L \sum_{l=1}^L \widehat{\boldsymbol{Z}}^{l-1} \boldsymbol{T} \boldsymbol{Z}^{L-l} \boldsymbol{\beta}$ in Theorem 24. $\qquad \square$

## G  Experiment details

### G.1  Synthetic experiments on Dirichlet energy evolution

We created a synthetic SC with 100 nodes, 241 edges and 135 triangles with the GUDHI toolbox Rouvreau (2015), and we set the initial inputs on three levels of simplices to be random sampled from $\mathcal{U}([-5, 5])$. We then built a SCCNN composed of simplicial shifting layers with weight $w_0$ and nonlinearities including id, tanh and relu. When the weight follows the condition in Proposition 5, from Fig. 10 (the dashed lines labled as "shift"), we see that the Dirichlet energies of all three outputs exponentially decrease as the number of layers increases. We then uncoupled the lower and upper parts of the Laplacians in the edge space in the shifting layers by setting $\gamma \neq 1$. As shown in Fig. 10 (the dotted lines), the Dirichlet energies of the edge outputs decrease at a slower rate than before. Lastly, we added the inter-simplicial couplings, which overcome the oversmoothing problems, as shown by the solid lines.

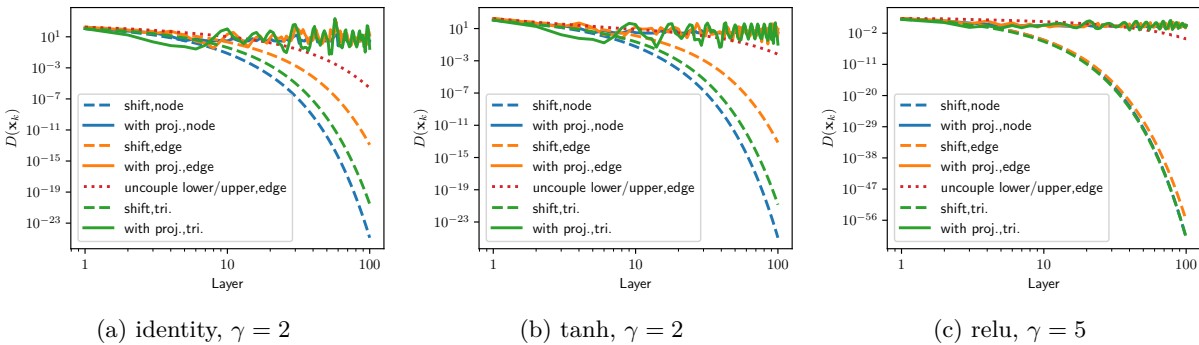

(a) identity, $\gamma = 2$          (b) tanh, $\gamma = 2$          (c) relu, $\gamma = 5$

Figure 10: Oversmoothing effects of simplicial shifting and the mitigation effects of uncoupling lower and upper adjacencies and accounting for inter-simplicial couplings.

### G.2  Additional details on Forex experiments

In the forex dataset, there are 25 currencies which can be exchanged pairwise at three timestamps. We first represented their exchange rates on the edges and took the logrithm, i.e., $[\boldsymbol{x}_1]_{[i,j]} = \log_{10} r^{i/j} = -[\boldsymbol{x}_1]_{[j,i]}$. Then, the total arbitrage can be computed as the total curl $\boldsymbol{B}_2^\top \boldsymbol{x}_1$.

We considered to recover the exchange rates under three types of settings: 1) random noise following normal distribution such that the signal-to-noise ration is $-3$dB, which is spread over the whole simplicial spectrum; 2) "curl noise" projected from triangle noise following normal distribution such that the signal-to-noise ration is $-3$dB, which is distributed only in the curl space; and 3) 50% of the total forex rates are recorded and the other half is not available, set as zero values.

Table 7: Forex results (nmse, arbitrage) and the corresponding hyperparameters.

| Methods | Random noise | "Curl noise" | Interpolation |
|---|---|---|---|
| Input | $0.119 \pm 0.004, 25.19 \pm 0.874$ | $0.552 \pm 0.027, 122.36 \pm 5.90$ | $0.717 \pm 0.030, 106.40 \pm 0.902$ |
| $\ell_2$-norm | $0.036 \pm 0.005, 2.29 \pm 0.079$ | $0.050 \pm 0.002, 11.12 \pm 0.537$ | $0.534 \pm 0.043 , 9.67 \pm 0.082$ |
| SNN | $0.11 \pm 0.005, 23.24 \pm 1.03$ $L = 5, F = 64, T = 4, \tanh$ | $0.446 \pm 0.017, 86.947 \pm 2.197$ $L = 6, F = 64, T = 3, \tanh$ | $0.702 \pm 0.033, 104.738 \pm 1.042$ $L = 2, F = 64, T = 1, \tanh$ |
| PSNN | $0.008 \pm 0.001, 0.984 \pm 0.17$ $L = 6, F = 64, \tanh$ | $0.000 \pm 0.000, 0.000 \pm 0.000$ $L = 5, F = 1, \text{id}$ | $0.009 \pm 0.001, 1.128 \pm 0.329$ $L = 6, F = 64, \tanh$ |
| Bunch | $0.981 \pm 0.0 , 22.912 \pm 1.228$ — | $0.981 \pm 0.0, 22.912 \pm 1.228$ — | $0.983 \pm 0.005 , 19.887 \pm 6.341$ — |
| MPSN | $0.039 \pm 0.004, 7.748 \pm 0.943$ $L = 2, F = 64, \text{id}, \text{sum}$ | $0.076 \pm 0.012, 14.922 \pm 2.493$ $L = 4, F = 64, \tanh, \text{mean}$ | $0.117 \pm 0.063, 23.147 \pm 11.674$ $L = 2, F = 64, \tanh, \text{sum}$ |
| SCCNN, id | $0.027 \pm 0.005, 0.000 \pm 0.000$ $L = 2, F = 16, T_{\mathrm{d}} = 0, T_{\mathrm{u}} = 3$ | $0.000 \pm 0.000, 0.000 \pm 0.000$ $L = 5, F = 1, T_{\mathrm{d}} = 1, T_{\mathrm{u}} = 1$ | $0.265 \pm 0.036 , 0.000 \pm 0.000$ $L = 2, F = 16, T_{\mathrm{d}} = 0, T_{\mathrm{u}} = 3$ |
| SCCNN, tanh | $0.002 \pm 0.000, 0.325 \pm 0.082$ $L = 6, F = 64, T_{\mathrm{d}} = 5, T_{\mathrm{u}} = 2$ | $0.000 \pm 0.000, 0.003 \pm 0.003$ $L = 1, F = 64, T_{\mathrm{d}} = 2, T_{\mathrm{u}} = 2$ | $0.003 \pm 0.002, 0.279 \pm 0.151$ $L = 6, F = 64, T_{\mathrm{d}} = 5, T_{\mathrm{u}} = 1$ |

First, as a baseline method, we chose $\ell_2$ norm of the curl $\boldsymbol{B}_2\boldsymbol{x}_1$ as a regularizer to reduce the total arbitrage, i.e., $\hat{\boldsymbol{x}}_1 = (\boldsymbol{I} + w\boldsymbol{L}_{1,\mathrm{u}})^{-1}\boldsymbol{x}_1$ with a regularization weight $w \in [0, 10]$. For the learning methods, we consider the following hyperparameter ranges: the number of layers to be $L \in \{1, 2, \ldots, 6\}$, the number of intermediate features to be $F \in \{1, 16, 32, 64\}$. For the convolutional methods including SNN Ebli et al. (2020), PSNN Roddenberry et al. (2021), Bunch Bunch et al. (2020) and SCCNN, we considered the intermediate layers with nonlinearities including id and tanh. The convolution orders of SNN and SCCNN are set to be $\{1, 2, \ldots, 5\}$. For the message-passing method, MPSN Bodnar et al. (2021b), we considered the setting from (Bodnar et al., 2021b, Eq. 35) where the sum and mean aggregations are used and each message update function is a two-layer MLP. With these noisy or masked rates as inputs and the clean arbitrage-free rates as outputs, we trained different learning methods at the first timestamp, and validated the hyperparameters at the second timestamp, and tested their performance at the thrid one. During the training of 1000 epochs, a normalized MSE loss function and adam optimizer with a fixed learning rate of 0.001 are used. We run the same experiments for 10 times. Table 7 reports the best results (nmse) and the total arbitrage, together with the hyperparameters.

### G.3 Additional details on Simplex prediction

#### G.3.1 Method in Detail

The method for simplex prediction is generalized from link prediction based on GNNs by Zhang & Chen (2018): For $k$-simplex prediction, we use an SCCNN in an SC of order up to $k$ to first learn the features of lower-order simplices up to order $k - 1$. Then, we concatenate these embedded lower-order simplicial features and input them to a two-layer MLP which predicts if a $k$-simplex is positive (closed, shall be included in the SC) or negative (open, not included in the SC).

For example, in 2-simplex prediction, consider an SC of order two, which is built based on nodes, edges and (existing positive) triangles. Given the initial inputs on nodes $\boldsymbol{x}_0$ and on edges $\boldsymbol{x}_1$ and zero inputs on triangles $\boldsymbol{x}_2 = \boldsymbol{0}$ since we assume no prior knowledge on triangles, for an open triangle $t = [i, j, k]$, an SCCNN is used to learn features on nodes and edges (denoted by $\boldsymbol{y}$). Then, we input the concatenation of the features on three nodes or three edges to an MLP, i.e., $\mathrm{MLP}_{\mathrm{node}}([\boldsymbol{y}_0]_i\|[\boldsymbol{y}_0]_j\|[\boldsymbol{y}_0]_k)$ or $\mathrm{MLP}_{\mathrm{edge}}([\boldsymbol{y}]_{[i,j]}\|[\boldsymbol{y}]_{[j,k]}\|[\boldsymbol{y}]_{[i,k]})$, to predict if triangle $t$ is positive or negative. A MLP taking both node and edge features is possible, but we keep it on one simplex level for complexity purposes. Similarly, we consider an SCCNN in an SC of order three for 3-simplex prediction, which is followed by an MLP operating on either nodes, edges or triangles.

### G.3.2 Data Preprocessing

We consider the data from the Semantic Scholar Open Research Corpus Ammar et al. (2018) to construct a coauthorship complex where nodes are authors and collaborations between $k$-author are represented by $(k-1)$-simplices. Following the preprocessing in Ebli et al. (2020), we obtain 352 nodes, 1472 edges, 3285 triangles, 5019 tetrahedrons (3-simplices) and a number of other higher-order simplices. The node signal $\boldsymbol{x}_0$, edge flow $\boldsymbol{x}_1$ and triangle flow $\boldsymbol{x}_2$ are the numbers of citations of single author papers and the collaborations of two and three authors, respectively.

For the 2-simplex prediction, we use the collaboration impact (the number of citations) to split the total set of triangles into the positive set $\mathcal{T}_P = \{t | [\boldsymbol{x}_2]_t > 7\}$ containing 1482 closed triangles and the negative set $\mathcal{T}_N = \{t | [\boldsymbol{x}_2]_t \leq 7\}$ containing 1803 open triangles such that we have balanced positive and negative samples. We further split the 80% of the positive triangle set for training, 10% for validation and 10% for testing; likewise for the negative triangle set. Note that in the construction of the SC, i.e., the incidence matrix $\boldsymbol{B}_2$, Hodge Laplacians $\boldsymbol{L}_{1,\mathrm{u}}$ and $\boldsymbol{L}_{2,\mathrm{d}}$, we ought to remove negative triangles in the training set and all triangles in the test set. That is, for 2-simplex prediction, we only make use of the training set of the positive triangles since the negative ones are not in the SC.

Similarly, we prepare the dataset for 3-simplex (tetrahedron) prediction, amounting to the tetradic collaboration prediction. We obtain balanced positive and negative tetrahedron sets based on the citation signal $\boldsymbol{x}_3$. In the construction of $\boldsymbol{B}_3$, $\boldsymbol{L}_{2,\mathrm{u}}$ and $\boldsymbol{L}_{3,\mathrm{d}}$, we again only use the tetrahedrons in the positive training set.

### G.3.3 Models

For comparison, we first use heuristic methods proposed in Benson et al. (2018) as baselines to determine if a triangle $t = [i, j, k]$ is closed, namely, 1) Harmonic mean: $s_t = 3/([\boldsymbol{x}_1]_{[i,j]}^{-1} + [\boldsymbol{x}_1]_{[j,k]}^{-1} + [\boldsymbol{x}_1]_{[i,k]}^{-1})$, 2) Geometric mean: $s_t = \lim_{p \to 0}[([\boldsymbol{x}_1]_{[i,j]}^p + [\boldsymbol{x}_1]_{[j,k]}^p + [\boldsymbol{x}_1]_{[i,k]}^p)]^{1/p}$, and 3) Arithmetic mean: $s_t = ([\boldsymbol{x}_1]_{[i,j]} + [\boldsymbol{x}_1]_{[j,k]} + [\boldsymbol{x}_1]_{[i,k]})/3$, which compute the triangle weight based on its three faces. Similarly, we generalized these mean methods to compute the weight of a 3-simplex $[i, j, k, m]$ based on the four triangle faces in 3-simplex prediction.

We then consider different learning methods. Specifically, 1) "Bunch" by Bunch et al. (2020) (we also generalized this model to 3-dimension for 3-simplex prediction); 2) Message passing simplicial network ("MPSN") by Bodnar et al. (2021b) which provides a baseline of message passing scheme in comparison to the convolution scheme; 3) Principled SNN ("PSNN") by Roddenberry et al. (2021); 4) SNN by Ebli et al. (2020); 5) SCNN by Yang et al. (2021); 6) GNN by Defferrard et al. (2016); 7) MLP: providing as a baseline for the effect of using inductive models.

For MLP, Bunch, MPSN and our SCCNN, we consider the outputs in the node and edge spaces, respectively, for 2-simplex prediction, which are denoted by a suffix "-Node" or "-Edge". For 3-simplex prediction, the output in the triangle space can be used as well, denoted by a suffix "-Tri.", where we also build SCNNs in both edge and triangle spaces.

### G.3.4 Experimental Setup and Hyperparameters

We consider the normalized Hodge Laplacians and incidence matrices, a particular version of the weighted ones Horak & Jost (2013); Grady & Polimeni (2010). Specifically, we use the symmetric version of the normalized random walk Hodge Laplacians in the edge space, proposed by Schaub et al. (2020), which were used in Bunch et al. (2020); Chen et al. (2022a) as well. We generalized the definitions for triangle predictions.

**Hyperparameters** 1) the number of layers: $L \in \{1, 2, 3, 4, 5\}$; 2) the number of intermediate and output features to be the same as $F \in \{16, 32\}$; 3) the convolution orders for SCCNNs are set to be the same, i.e., $T'_\mathrm{d} = T_\mathrm{d} = T_\mathrm{u} = T'_\mathrm{u} = T \in \{1, 2, 3, 4, 5\}$. We do so to avoid the exponential growth of the parameter search space. For GNNs (Defferrard et al., 2016) and SNNs (Ebli et al., 2020), we set the convolution orders to be $T \in \{1, 2, 3, 4, 5\}$ while for SCNNs (Yang et al., 2022a), we allow the lower and upper convolutions to have different orders with $T_\mathrm{d}, T_\mathrm{u} \in \{1, 2, 3, 4, 5\}$; 4) the nonlinearity in the feature learning phase: LeakyReLU with a negative slope 0.01; 5) MPSN is set as Bodnar et al. (2022); 6) the MLP in the prediction phase: two

layers with a sigmoid nonlinearity. For 2-simplex prediction, the number of the input features for the node features is $3F$, and for the edge features is $3F$. For 3-simplex prediction, the number of the input features for the node features is $4F$, for the edge features is $6F$ and for the triangle features is $4F$ since a 3-simplex has four nodes, six edges and four triangles. The number of the intermediate features is the same as the input features, and that of the output features is one; and, 7) the binary cross entropy loss and the adam optimizer with a learning rate of 0.001 are used; the number of the epochs is 1000 where an early stopping is used. We compute the AUC to compare the performance and run the same experiments for ten times with random data splitting.

### G.3.5 Results

In Table 8, we report the best results of each method with the corresponding hyperparameters. Different hyperparameters can lead to similar results, but we report the ones with the *least* complexity. All experiments for simplex predictions were run on a single NVIDIA A40 GPU with 48 GB of memory using CUDA 11.5.

Table 8: 2- *(Left)* and 3-Simplex *(Right)* prediction AUC (%) results.

| METHODS | AUC | PARAMETERS | METHODS | AUC | PARAMETERS |
|---|---|---|---|---|---|
| HARM. MEAN | $62.8\pm2.7$ | — | HARM. MEAN | $63.6\pm1.6$ | — |
| ARITH. MEAN | $60.8\pm3.2$ | — | ARITH. MEAN | $62.2\pm1.4$ | — |
| GEOM. MEAN | $61.7\pm3.1$ | — | GEOM. MEAN | $63.1\pm1.4$ | — |
| MLP-NODE | $68.5\pm1.6$ | $L=1, F=32$ | MLP-TRI. | $69.0\pm2.2$ | $L=3, F=32$ |
| GNN | $93.9\pm1.0$ | $L=5, F=32, T=2$ | GNN | $96.6\pm0.5$ | $L=5, F=32, T=5$ |
| SNN-EDGE | $92.0\pm1.8$ | $L=5, F=32, T=5$ | SNN-TRI. | $95.1\pm1.2$ | $L=5, F=32, T=5$ |
| PSNN-EDGE | $95.6\pm1.3$ | $L=5, F=32$ | PSNN-TRI. | $98.1\pm0.5$ | $L=5, F=32$ |
| SCNN-EDGE | $96.5\pm1.5$ | $L=5, F=32, T_{\mathrm{d}}=5, T_{\mathrm{u}}=2$ | SCNN-TRI. | $98.3\pm0.4$ | $L=5, F=32, T_{\mathrm{d}}=2, T_{\mathrm{u}}=1$ |
| BUNCH-NODE | $98.3\pm0.5$ | $K=1, L=4, F=32$ | BUNCH-EDGE | $98.5\pm0.5$ | $K=3, L=4, F=16$ |
| MPSN-NODE | $98.1\pm0.5$ | $K=1, L=3, F=32$ | MPSN-EDGE | $99.2\pm0.3$ | $K=3, L=3, F=32$ |
| SCCNN-NODE | $98.7\pm0.5$ | $K=1, L=2, F=32, T=2$ | SCCNN-NODE | $99.4\pm0.3$ | $K=3, L=3, F=32, T=3$ |

### G.3.6 Complexity

Table 9: *(Left)* Complexity of the best three methods for 2-simplex prediction. *(Right)* Running time of SCCNN with different layers and convolution orders.

| METHOD | #PARAMS. | RUNNING TIME (SECONDS PER EPOCH) | HYPERPARAMS. | $T=2$ | $T=5$ |
|---|---|---|---|---|---|
| SCCNN | 24288 | 0.073 | $L=2$ | 0.073 | 0.082 |
| BUNCH | 21728 | 0.140 | $L=3$ | 0.110 | 0.130 |
| MPSN | 84256 | 0.028 | $L=5$ | 0.192 | 0.237 |

Here we report the number of parameters and the running time of SCCNN for 2-simplex prediction on one NVIDIA Quadro K2200 with 4GB memory, compared with the two best alternatives. MPSN, compared to convolutional methods, has three times more parameters, analogous to the comparison between message-passing and graph convolutional NNs. We also report the running time as the layers and convolution orders increase.

### G.3.7 Ablation Study

We perform an ablation study to observe the roles of different components in SCCNNs.

**SC Order $K$** We investigate the influence of the SC order $K$. Table 10 reports the 2-simplex prediction results for $K=\{1,2\}$ and the 3-simplex prediction results for $K=\{1,2,3\}$. We observe that for $k$-simplex prediction, it does not necessarily guarantee a better prediction with a higher-order SC, which further

indicates that a positive simplex could be well encoded by both its faces and other lower-order subsets. For example, in 2-simplex prediction, SC of order one gives better results than SC of order two (similar for Bunch), showing that in this coauthorship complex, triadic collaborations are better encoded by features on nodes than pairwise collaborations. In 3-simplex prediction, SCs of different orders give similar results, showing that tetradic collaborations can be encoded by nodes, as well as by pairwise and triadic collaborations.

Table 10: Prediction results of SCCNNs with different SC order $K$.

| METHOD | 2-SIMPLEX | PARAMETERS |
|---|---|---|
| SCCNN-NODE | 98.7±0.5 | $K = 1, L = 2, F = 32, T = 2$ |
| SCCNN-NODE | 98.4±0.5 | $K = 2, L = 2, F = 32, T = 2$ |
| BUNCH-NODE | 98.3±0.4 | $K = 1, L = 4, F = 32$ |
| BUNCH-NODE | 98.0±0.4 | $K = 2, L = 4, F = 32$ |
| MPSN-NODE | 94.5±1.5 | $K = 1, L = 3, F = 32$ |
| MPSN-NODE | 98.1±0.5 | $K = 2, L = 3, F = 32$ |
| SCCNN-EDGE | 97.9±0.9 | $K = 1, L = 3, F = 32, T = 5$ |
| SCCNN-EDGE | 95.9±1.0 | $K = 2, L = 5, F = 32, T = 3$ |
| BUNCH-EDGE | 97.3±1.1 | $K = 1, L = 4, F = 32$ |
| BUNCH-EDGE | 94.6±1.2 | $K = 2, L = 4, F = 32$ |
| MPSN-EDGE | 94.1±2.4 | $K = 1, L = 3, F = 32$ |
| MPSN-EDGE | 97.0±1.2 | $K = 2, L = 2, F = 16$ |

| METHOD | 3-SIMPLEX | PARAMETERS |
|---|---|---|
| SCCNN-NODE | 99.3±0.3 | $K = 1, L = 2, F = 32, T = 1$ |
| SCCNN-NODE | 99.3±0.2 | $K = 2, L = 2, F = 32, T = 5$ |
| SCCNN-NODE | 99.4±0.3 | $K = 3, L = 3, F = 32, T = 3$ |
| MPSN-NODE | 96.0±1.2 | $K = 1, L = 3, F = 32$ |
| MPSN-NODE | 98.2±0.8 | $K = 2, L = 2, F = 32$ |
| SCCNN-EDGE | 98.9±0.5 | $K = 1, L = 3, F = 32, T = 5$ |
| SCCNN-EDGE | 99.2±0.4 | $K = 2, L = 5, F = 32, T = 5$ |
| SCCNN-EDGE | 99.0±1.0 | $K = 3, L = 5, F = 32, T = 5$ |
| MPSN-EDGE | 96.3±1.1 | $K = 1, L = 3, F = 32$ |
| MPSN-EDGE | 98.3±0.8 | $K = 2, L = 3, F = 32$ |
| SCCNN-TRI. | 97.9±0.7 | $K = 2, L = 4, F = 32, T = 4$ |
| SCCNN-TRI. | 97.4±0.9 | $K = 3, L = 4, F = 32, T = 4$ |
| MPSN-TRI. | 99.1±0.2 | $K = 2, L = 3, F = 32$ |

**Missing Components in SCCNN**  With a focus on 2-simplex prediction with SCCNN-Node of order one, to avoid overcrowded settings, we study how each component of an SCCNN influences the prediction. We consider the following settings without: 1) "Edge-to-Node", where the projection $\boldsymbol{x}_{0,\mathrm{u}}$ from edge to node is not included, equivalent to GNN; 2) "Node-to-Node", where for node output, we have $\boldsymbol{x}_0^l = \sigma(\boldsymbol{H}_{0,\mathrm{u}}^l \boldsymbol{R}_{1,\mathrm{u}} \boldsymbol{x}_1^{l-1})$; 3) "Node-to-Edge", where the projection $\boldsymbol{x}_{1,\mathrm{d}}$ from node to edge is not included, i.e., we have $\boldsymbol{x}_1^l = \sigma(\boldsymbol{H}_1^l \boldsymbol{x}_1^{l-1})$; and 4) "Edge-to-Edge", where for edge output, we have $\boldsymbol{x}_1^l = \sigma(\boldsymbol{H}_{1,\mathrm{d}}^l \boldsymbol{R}_{1,\mathrm{d}} \boldsymbol{x}_0^{l-1})$.

Table 11: 2-Simplex prediction (SCCNN-Node without certain components or with limited inputs).

| Missing Component | AUC | Parameters |
|---|---|---|
| — | 98.7±0.5 | $L = 2, F = 32, T = 2$ |
| Edge-to-Node | 93.9±0.8 | $L = 5, F = 32, T = 2$ |
| Node-to-Node | 98.7±0.4 | $L = 4, F = 32, T = 2$ |
| Edge-to-Edge | 98.5±1.0 | $L = 3, F = 32, T = 3$ |
| Node-to-Edge | 98.8±0.3 | $L = 4, F = 32, T = 3$ |

| Missing Input | AUC | Parameters |
|---|---|---|
| — | 98.7±0.5 | $L = 2, F = 32, T = 2$ |
| Node input | 98.2±0.5 | $L = 2, F = 32, T = 4$ |
| Edge input | 98.1±0.4 | $L = 2, F = 32, T = 3$ |
| Node, Edge inputs | 50.0±0.0 | — |

From the results in Table 11 (Left), we see that "No Edge-to-Node", i.e., GNN, gives much worse results as it leverages no information on edges with limited expressive power. For cases with other components missing, a similar performance can be achieved, however, at a cost of the model complexity, with either a higher convolution order or a larger number of layers $L$, while the latter in turn degrades the stability of the SCCNNs, as discussed in Section 5. SCCNNs with certain inter-simplicial couplings pruned/missing can be powerful as well (this is similarly shown by (Bodnar et al., 2021b, Thm. 6)), but if we did not consider certain component, it comes with a cost of complexity which might degrade the model stability if more number layers are required.

**Limited Input**  We study the influence of limited input data for model SCCNN-Node of order two. Specifically, we consider the input on either nodes or edges is missing. From Table 11, we see that the prediction performance does not deteriorate at a cost of the model complexity (higher convolution orders) when a certain part of the input missing except with full zeros as input. This ability of learning from limited data shows the robustness of SCCNNs.

### G.3.8 Stability Analysis

We then perform a stability analysis of SCCNNs. We artificially add perturbations to the normalization matrices when defining the Hogde Laplacians, which resemble the weights of simplices. We consider small perturbations $\boldsymbol{E}_0$ on node weights which is a diagonal matrix following that $\|\boldsymbol{E}_0\| \leq \epsilon_0/2$. We generate its diagonal entries from a uniform distribution $[-\epsilon_0/2, \epsilon_0/2)$ with $\epsilon_0 \in [0, 1]$, which represents one degree of deviation of the node weigths from the true ones. Similarly, perturbations on edge weights and triangle weights are applied to study the stability. In a SCCNN-Node for 2-simplex prediction of $K = 2$, we measure the distance between the simplicial outputs with and without perturbations on nodes, edges, and triangles, i.e., $\|\boldsymbol{x}_k^L - \hat{\boldsymbol{x}}_k^L\|/\|\boldsymbol{x}_k^L\|$, for $k = 0, 1, 2$.

**Stability dependence**  We first show the stability mutual dependence between different simplices in Fig. 11. We see that under perturbation on node weights, the triangle output is not influenced until the number of layers becomes two; likewise, the node output is not influenced by perturbations on triangle weights with a one-layer SCCNN. Also, a one-layer SCCNN under perturbations on edge weights will cause outputs on nodes, edges, triangles perturbed. Lastly, we observe that the same degree of perturbations added to different simplices causes different degrees of instability, owing to the number $n_k$ of $k$-simplices in the stability bound. Since $N_0 < N_1 < N_2$, the perturbations on node weights cause less instability than those on edge and triangle weights.

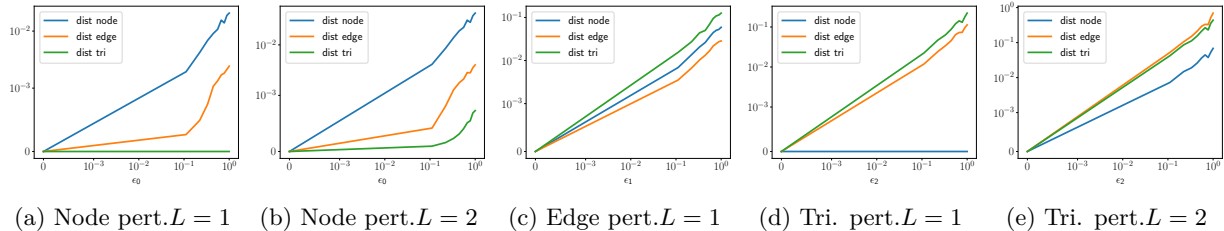

(a) Node pert.$L = 1$    (b) Node pert.$L = 2$    (c) Edge pert.$L = 1$    (d) Tri. pert.$L = 1$    (e) Tri. pert.$L = 2$

Figure 11: The stabilities of different simplicial outputs are dependent on each other.

**Number of Layers**  Fig. 12 shows that the stability of SCCNNs degrades as the number of layers increases as studied in Theorem 24. As the NN deepens, the stability deteriorates, which corresponds to our analysis of using shallow layers.

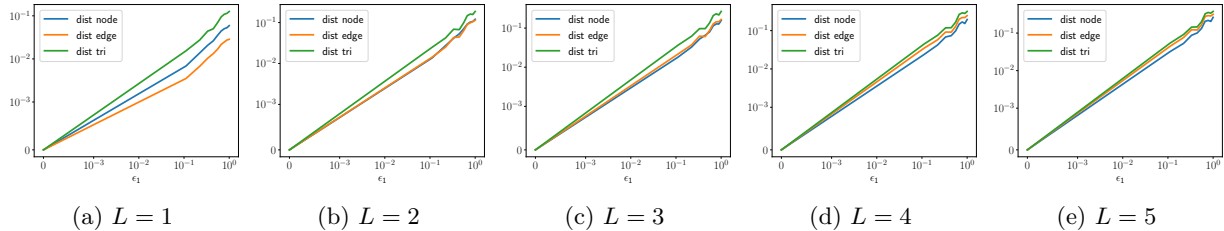

(a) $L = 1$            (b) $L = 2$            (c) $L = 3$            (d) $L = 4$            (e) $L = 5$

Figure 12: The relative difference of SCCNN outputs with and without perturbations in terms of different numbers of layers. We consider perturbations on edge weights.

### G.4 Additional details on Trajectory prediction

### G.4.1 Problem Formulation

A trajectory of length $m$ can be modeled as a sequence of nodes $[v_0, v_1, \ldots, v_{m-1}]$ in an SC. The task is to predict the next node $v_m$ from the neighbors of $v_{m-1}$, $\mathcal{N}_{v_{m-1}}$. The algorithm in Roddenberry et al. (2021) first represents the trajectory equivalently as a sequence of oriented edges $[[v_0, v_1], [v_1, v_2], \ldots, [v_{m-2}, v_{m-1}]]$. Then, an edge flow $\boldsymbol{x}_1$ is defined, whose value on an edge $e$ is $[\boldsymbol{x}_1]_e = 1$ if edge $e$ is traversed by the trajectory

in a forward direction, $[\boldsymbol{x}_1]_e = -1$ if edge $e$ is traversed in a backward direction by the trajectory, and $[\boldsymbol{x}_1]_e = 0$, otherwise.

With the trajectory flow $\boldsymbol{x}_1$ as the input, together with zero inputs on the nodes and triangles, an SCCNN of order two is used to generate a representation $\boldsymbol{x}_1^L$ of the trajectory, which is the output on edges. This is followed by a projection step $\boldsymbol{x}_{0,\mathrm{u}}^L = \boldsymbol{B}_1 \boldsymbol{W} \boldsymbol{x}_1^L$, where the output is first passed through a linear transformation via $\boldsymbol{W}$, then projected into the node space via $\boldsymbol{B}_1$. Lastly, a distribution over the candidate nodes $\mathcal{N}_{v_{m-1}}$ is computed via a softmax operation, $\boldsymbol{n}_j = \mathrm{softmax}([\boldsymbol{x}_{0,\mathrm{u}}^L]_j), j \in \mathcal{N}_{v_{m-1}}$. The best candidate is selected as $v_m = \mathrm{argmax}_j \boldsymbol{n}_j$. We refer to Roddenberry & Segarra (2019, Alg. S-2) for more details.

Given that an SCCNN of order two generates outputs also on nodes, we can directly apply the node feature output $\boldsymbol{x}_0^L$ to compute a distribution over the candidate nodes $\mathcal{N}_{v_{m-1}}$ without the projection step. We refer to this as SCCNN-Node, and the method of using the edge features with the projection step as SCCNN-Edge.

### G.4.2   Model

In this experiment, we consider the following methods: 1) PSNN by Roddenberry et al. (2021); 2) SNN by Ebli et al. (2020); 3) SCNN by Yang et al. (2022a) where we consider different lower and upper convolution orders $T_\mathrm{d}, T_\mathrm{u}$; and 4) Bunch by Bunch et al. (2020) where we consider both the node features and edge features, namely, Bunch-Node and Bunch-Edge.

**Synthetic Data**   Following the procedure in Schaub et al. (2020), we generate 1000 trajectories as follows. First, we create an SC with two "holes" by uniformly drawing 400 random points in the unit square, and then a Delaunay triangulation is applied to obtain a mesh, followed by the removal of nodes and edges in two regions. To generate a trajectory, we consider a starting point at random in the lower-left corner, and then connect it via a shortest path to a random point in the upper left, center, or lower-right region, which is connected to another random point in the upper-right corner via a shortest path.

We consider the random walk Hodge Laplacians Schaub et al. (2020). For Bunch method, we set the shifting matrices as the simplicial adjacency matrices defined in Bunch et al. (2020). We consider different NNs with three intermediate layers where each layer contains $F = 16$ intermediate features. The tanh nonlinearity is used such that the orientation equivariance holds. The final projection $\boldsymbol{n}$ generates a node feature of dimension one. In the 1000-epoch training, we use the cross-entropy loss function between the output $\boldsymbol{d}$ and the true candidate and we consider an adam optimizer with a learning rate of 0.001 and a batch size 100. To avoid overfitting, we apply a weight decay of $5 \cdot 10^{-6}$ and an early stopping.

As done in Roddenberry et al. (2021), besides the standard trajectory prediction task, we also perform a reverse task where the training set remains the same but the direction of the trajectories in the test set is reversed and a generalization task where the training set contains trajectories running along the upper left region and the test set contains trajectories around the other region. We evaluate the correct prediction ratio by averaging the performance over 10 different data generations.

**Real Data**   We also consider the Global Drifter Program dataset[3] localized around Madagascar. It consists of ocean drifters whose coordinates are logged every 12 hours. An SC can then be created as Schaub et al. (2020) by treating each mesh as a node, connecting adjacent meshes via an edge and filling the triangles, where the "hole" is yielded by the island. Following the process in Roddenberry et al. (2021), it results in 200 trajectories and we use 180 of them for training. In the training, a batch size of 10 is used and no weight decay is used. The rest experiment setup remains the same as the synthetic case.

### G.4.3   Results

We report the prediction accuracy of different tasks for both datasets in Table 12. We first investigate the effects of applying higher-order SCFs in the simplicial convolution and accounting for the lower and upper contributions. From the standard accuracy for both datasets, we observe that increasing the convolution orders improves the prediction accuracy, e.g., SCNNs become better as the orders $T_\mathrm{d}, T_\mathrm{u}$ increase and perform

---

[3] http://www.aoml.noaa.gov/envids/gld/,

always better than PSNN, and SCCNNs better than Bunch. Also, differentiating the lower and upper convolutions does help improve the performance as SCNN of orders $T_\mathrm{d} = T_\mathrm{u} = 3$ performs better than SNN of $T = 3$.

However, accounting for the node and triangle contributions in SCCNNs does not help the prediction compared to the SCNNs, likewise for Bunch compared to PSNN. This is due to the zero node and triangle inputs because there are no available node and triangle features. Similarly, the prediction directly via the node output features is not accurate compared to projection from edge features.

Moreover, we also observe that the performance of SCCNNs that are trained with the same data does not deteriorate in the reverse task because the orientation equivariance ensures SCCNNs to be unaffected by the orientations of the simplicial data. Lastly, we see that, like other NNs on SCs, SCCNNs have good transferability to the unseen data.

Table 12: Trajectory Prediction Accuracy. *(Left)*: Synthetic trajectory in the standard, reverse and generalization tasks. *(Right)*: Ocean drifter trajectories. For SCCNNs, we set the lower and upper convolution orders $T_\mathrm{d}, T_\mathrm{u}$ to be the same as $T$.

| Methods | Standard | Reverse | Generalization | Parameters | Standard | Parameters |
|---|---|---|---|---|---|---|
| PSNN | 63.1±3.1 | 58.4±3.9 | 55.3±2.5 | — | 49.0±8.0 | — |
| SCNN | 65.6±3.4 | 56.6±6.0 | 56.1±3.6 | $T_\mathrm{d} = T_\mathrm{u} = 2$ | 52.5±9.8 | $T_\mathrm{d} = T_\mathrm{u} = 2$ |
| SCNN | 66.5±5.8 | 57.7±5.4 | 60.6±4.0 | $T_\mathrm{d} = T_\mathrm{u} = 3$ | 52.5±7.2 | $T_\mathrm{d} = T_\mathrm{u} = 3$ |
| SCNN | 67.3±2.3 | 56.9±4.8 | 59.4±4.2 | $T_\mathrm{d} = T_\mathrm{u} = 4$ | 52.5±8.7 | $T_\mathrm{d} = T_\mathrm{u} = 4$ |
| SCNN | 67.7±1.7 | 55.3±5.3 | 61.2±3.2 | $T_\mathrm{d} = T_\mathrm{u} = 5$ | 53.0±7.8 | $T_\mathrm{d} = T_\mathrm{u} = 5$ |
| SNN | 65.5±2.4 | 53.6±6.1 | 59.5±3.7 | $T = 3$ | 52.5±6.0 | $T = 3$ |
| Bunch-Node | 35.4±3.4 | 38.1±4.6 | 29.0±3.0 | — | 35.0±5.9 | — |
| Bunch-Edge | 62.3±4.0 | 59.6±6.1 | 53.9±3.1 | — | 46.0±6.2 | — |
| SCCNN-Node | 46.8±7.3 | 44.5±8.2 | 31.9±5.0 | $T = 1$ | 40.5±4.7 | $T = 1$ |
| SCCNN-Edge | 64.6±3.9 | 57.2±6.3 | 54.0±3.0 | $T = 1$ | 52.5±7.2 | $T = 1$ |
| SCCNN-Node | 43.5±9.6 | 44.4±7.6 | 32.8±2.6 | $T = 2$ | 45.5±4.7 | $T = 2$ |
| SCCNN-Edge | 65.2±4.1 | 58.9±4.1 | 56.8±2.4 | $T = 2$ | 54.5±7.9 | $T = 2$ |

### G.4.4 Convolution Order and Integral Lipschitz Property

Here, to illustrate that the integral Lipschitz property of the SCFs helps the stability of NNs on SCs, we consider the effect of regularizer $r_\mathrm{IL}$ in Eq. (16) against perturbations in PSNNs and SCNNs with different $T_\mathrm{d}$ and $T_\mathrm{u}$ for the standard synthetic trajectory prediction. The regularization weight on $r_\mathrm{IL}$ is set as $5 \cdot 10^{-4}$ and the number of samples to approximate the frequencies is set such that the sampling interval is 0.01.

Fig. 13 shows the prediction accuracy and the relative distance between the edge outputs of the NNs trained with and without the integral Lipschitz regularizer in terms of different levels of perturbations. We see that the integral Lipschitz regularizer helps the stability of the NNs, especially for large SCF orders, where the edge output is less influenced by the perturbations compared to without the regularizer. Meanwhile, SCNN with higher-order SCFs, e.g., $T_\mathrm{d} = T_\mathrm{u} = 5$, achieves better prediction than PSNN (with one-step simplicial shifting), while maintaining a good stability with its output not influenced by perturbations drastically.

We also measure the lower and upper integral Lipschitz constants of the trained NNs across different layers and features, given by $\max_{\lambda_{k,\mathrm{G}}} |\lambda_{k,\mathrm{G}} \tilde{h}_{k,\mathrm{G}}(\lambda_{k,\mathrm{G}})|$ and $\max_{\lambda_{k,\mathrm{C}}} |\lambda_{k,\mathrm{C}} \tilde{h}_{k,\mathrm{C}}(\lambda_{k,\mathrm{C}})|$, shown in Fig. 14. We see that the SCNN trained with $r_\mathrm{IL}$ indeed has smaller integral Lipschitz constants than the one trained without the regularizer, thus, a better stability, especially for NNs with higher-order SCFs.

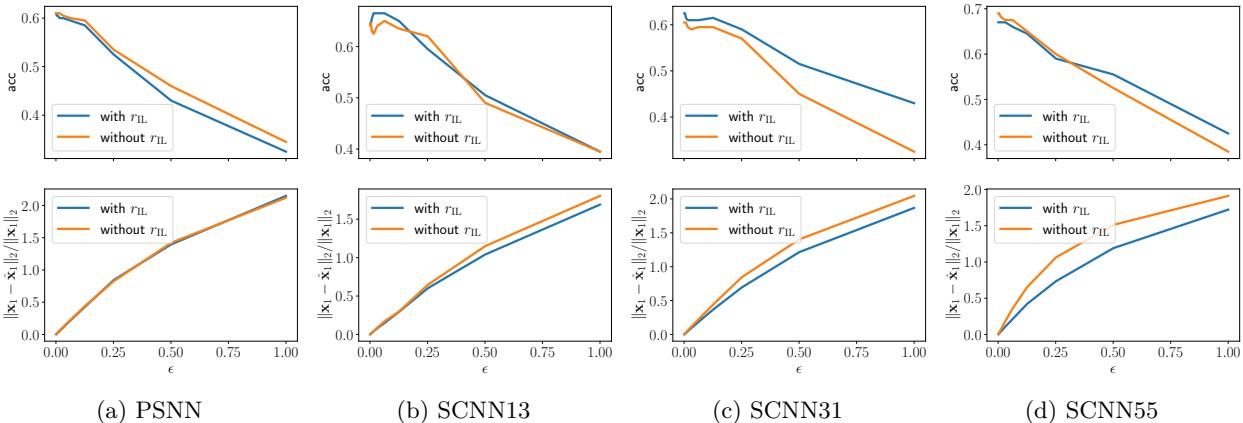

Figure 13: Effect of the integral Lipschitz regularizer $r_{\mathrm{IL}}$ in the task of synthetic trajectory prediction against different levels $\epsilon$ of random perturbations on $\boldsymbol{L}_{1,\mathrm{d}}$ and $\boldsymbol{L}_{1,\mathrm{u}}$. We show the accuracy (Top row) and the relative distance between the edge output (Bottom row) for different NNs on SCs with and without $r_{\mathrm{IL}}$. SCNN13 is the SCNN with $T_{\mathrm{d}} = 1$ and $T_{\mathrm{u}} = 3$.

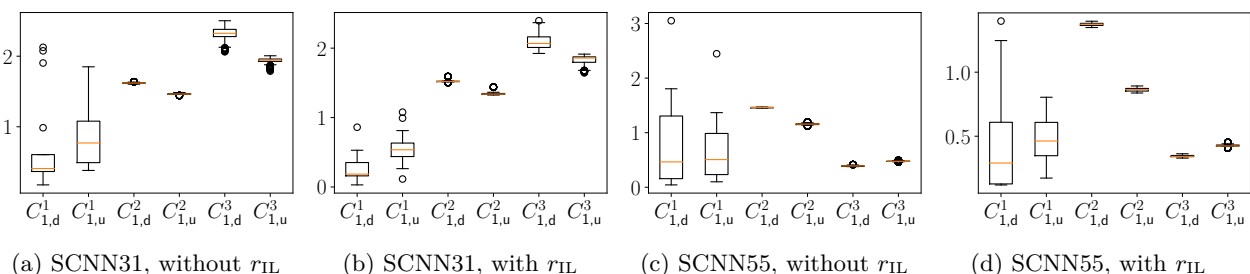

Figure 14: The integral Lipschitz constants of SCFs at each layer of the trained SCNNs with and without the integral Lipschitz regularizer $r_{\mathrm{IL}}$. We use symbols $c^l_{k,\mathrm{d}}$ and $c^l_{k,\mathrm{u}}$ to denote the lower and upper integral Lipschitz constants at layer $l$. Regularizer $r_{\mathrm{IL}}$ promotes the integral Lipschitz property, thus, the stability, especially for NNs with large SCF orders.

