## Contents

# A   Illustration for Background

Ths paper relies on the Hodge decomposition and the spectral simplicial theory. To ease the exposition, we illustrate them for the edge flow space. We refer to Barbarossa & Sardellitti (2020); Yang et al. (2021; 2022b) for more details.

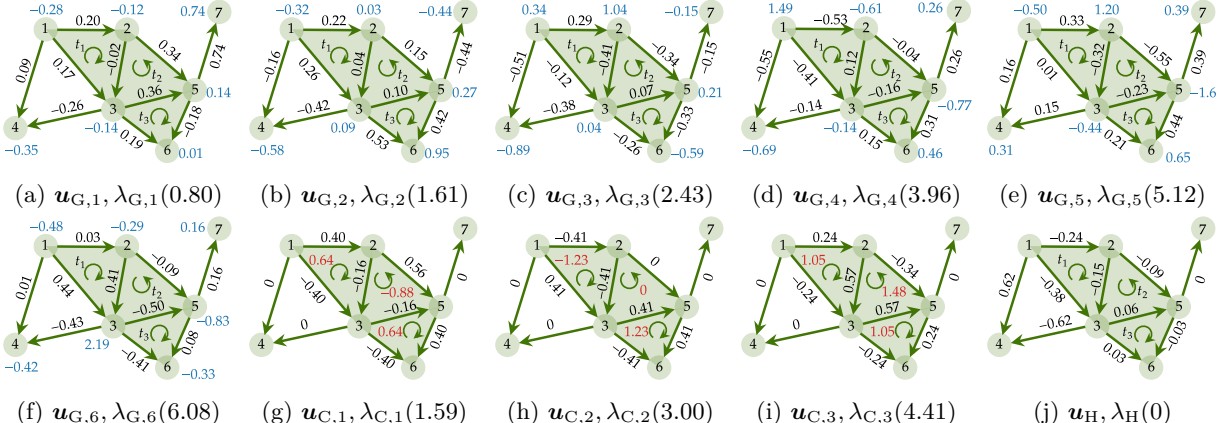

(a) $\boldsymbol{u}_{\mathrm{G},1}, \lambda_{\mathrm{G},1}(0.80)$   (b) $\boldsymbol{u}_{\mathrm{G},2}, \lambda_{\mathrm{G},2}(1.61)$   (c) $\boldsymbol{u}_{\mathrm{G},3}, \lambda_{\mathrm{G},3}(2.43)$   (d) $\boldsymbol{u}_{\mathrm{G},4}, \lambda_{\mathrm{G},4}(3.96)$   (e) $\boldsymbol{u}_{\mathrm{G},5}, \lambda_{\mathrm{G},5}(5.12)$

(f) $\boldsymbol{u}_{\mathrm{G},6}, \lambda_{\mathrm{G},6}(6.08)$   (g) $\boldsymbol{u}_{\mathrm{C},1}, \lambda_{\mathrm{C},1}(1.59)$   (h) $\boldsymbol{u}_{\mathrm{C},2}, \lambda_{\mathrm{C},2}(3.00)$   (i) $\boldsymbol{u}_{\mathrm{C},3}, \lambda_{\mathrm{C},3}(4.41)$   (j) $\boldsymbol{u}_{\mathrm{H}}, \lambda_{\mathrm{H}}(0)$

Figure 9: (a)-(f) Six gradient frequencies and the corresponding Fourier basis. We also annotate their divergences, and we see that these eigenvectors with a small eigenvalue have a small magnitude of total divergence, i.e., the edge flow variation in terms of the nodes. Gradient frequencies reflect the nodal variations. (g)-(i) Three curl frequencies and the corresponding Fourier basis. We annotate their curls and we see that these eigenvectors with a small eigenvalue have a small magnitude of total curl, i.e., the edge flow variation in terms of the triangles. Curl frequencies reflect the rotational variations. (j) Harmonic basis with a zero frequency, which has a zero nodal and zero rotational variation.

## A.1   Spectral simplicial theory

Here we show how the eigenvalues of $\boldsymbol{L}_k$ carry the notion of simplicial frequency Yang et al. (2022b). Specifically, we show for $k = 1$ an eigenvalue measures the total divergence or curl of the eigenvector.

- *Gradient Frequency:* the nonzero eigenvalues associated with the eigenvectors $\boldsymbol{U}_{1,\mathrm{G}}$ of $\boldsymbol{L}_{1,\mathrm{d}}$, which span the gradient space $\mathrm{im}(\boldsymbol{B}_1^\top)$, admit $\boldsymbol{L}_{1,\mathrm{d}}\boldsymbol{u}_{1,\mathrm{G}} = \lambda_{1,\mathrm{G}}\boldsymbol{u}_{1,\mathrm{G}}$ for any eigenpair $\boldsymbol{u}_{1,\mathrm{G}}$ and $\lambda_{1,\mathrm{G}}$. Thus, we have $\lambda_{1,\mathrm{G}} = \boldsymbol{u}_{1,\mathrm{G}}^\top \boldsymbol{L}_{1,\mathrm{d}} \boldsymbol{u}_{1,\mathrm{G}} = \boldsymbol{u}_{1,\mathrm{G}}^\top \boldsymbol{B}_1^\top \boldsymbol{B}_1 \boldsymbol{u}_{1,\mathrm{G}} = \|\boldsymbol{B}_1 \boldsymbol{u}_{1,\mathrm{G}}\|_2^2$, which is an Euclidean norm of the divergence, i.e., the total nodal variation of $\boldsymbol{u}_{1,\mathrm{G}}$. If an eigenvector has a larger eigenvalue, it has a larger total divergence. For the SFT of an edge flow, if the gradient embedding $\tilde{\boldsymbol{x}}_{1,\mathrm{G}}$ has a large weight on such an eigenvector, it contains components with a large divergence, and we say it has a large gradient frequency. Thus, we call such eigenvalues associated with $\boldsymbol{U}_{1,\mathrm{G}}$ gradient frequencies.

- *Curl Frequency:* the nonzero eigenvalues associated with the eigenvectors $\boldsymbol{U}_{1,\mathrm{C}}$ of $\boldsymbol{L}_{1,\mathrm{u}}$, which span the curl space $\mathrm{im}(\boldsymbol{B}_2)$, admit $\boldsymbol{L}_{1,\mathrm{u}}\boldsymbol{u}_{1,\mathrm{C}} = \lambda_{1,\mathrm{C}}\boldsymbol{u}_{1,\mathrm{C}}$ for any eigenpair $\boldsymbol{u}_{1,\mathrm{C}}$ and $\lambda_{1,\mathrm{C}}$. Thus, we have $\lambda_{1,\mathrm{C}} = \boldsymbol{u}_{1,\mathrm{C}}^\top \boldsymbol{L}_{1,\mathrm{u}} \boldsymbol{u}_{1,\mathrm{C}} = \boldsymbol{u}_{1,\mathrm{C}}^\top \boldsymbol{B}_2 \boldsymbol{B}_2^\top \boldsymbol{u}_{1,\mathrm{C}} = \|\boldsymbol{B}_2^\top \boldsymbol{u}_{1,\mathrm{C}}\|_2^2$, which is an Euclidean norm of the curl, i.e., the total rotational variation of $\boldsymbol{u}_{1,\mathrm{C}}$. If an eigenvector has a larger eigenvalue, it has a larger total curl. For the SFT of an edge flow, if the curl embedding $\tilde{\boldsymbol{x}}_{1,\mathrm{C}}$ has a large weight on such an eigenvector, it contains components with a large curl, and we say it has a large curl frequency. Thus, we call such eigenvalues associated with $\boldsymbol{U}_{1,\mathrm{C}}$ curl frequencies.

- *Harmonic Frequency:* the zero eigenvalues associated with the eigenvectors $\boldsymbol{U}_{1,\mathrm{H}}$, which span the harmonic space $\ker(\boldsymbol{L}_1)$, admit $\boldsymbol{L}_1 \boldsymbol{u}_{1,\mathrm{H}} = \boldsymbol{0}$ for any eigenpair $\boldsymbol{u}_{1,\mathrm{H}}$ and $\lambda_{1,\mathrm{H}} = 0$. From the definition of $\boldsymbol{L}_1$, we have $\boldsymbol{B}_1 \boldsymbol{u}_{1,\mathrm{H}} = \boldsymbol{B}_2^\top \boldsymbol{u}_{1,\mathrm{H}} = \boldsymbol{0}$. That is, the eigenvector $\boldsymbol{u}_{1,\mathrm{

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

We investigate the effect of the integral Lipschitz property of the SCFs in an NN on SC. To do so, given an NN on SCs with an SCF $\boldsymbol{H}_k$ for $k$-simplicial signals, we add the following integral Lipschitz regularizer to the loss function during training so to promote the integral Lipschitz property

$$r_{\mathrm{IL}} = \|\lambda_{k,\mathrm{G}} \tilde{h}'_{k,\mathrm{G}}(\lambda_{k,\mathrm{G}})\| + \|\lambda_{k,\mathrm{C}} \tilde{h}'_{k,\mathrm{C}}(\lambda_{k,\mathrm{C}})\| = \left\| \sum_{t=0}^{T_{\mathrm{d}}} t w_{k,\mathrm{d},t} \lambda_{k,\mathrm{G}}^t \right\| + \left\| \sum_{t=0}^{T_{\mathrm{u}}} t w_{k,\mathrm{u},t} \lambda_{k,\mathrm{C}}^t \right\| \tag{82}$$

for $\lambda_{k,\mathrm{G}} \in \{\lambda_{k,\mathrm{G},i}\}_{i=1}^{n_{k,\mathrm{G}}}$ and $\lambda_{k,\mathrm{C}} \in \{\lambda_{k,\mathrm{C},i}\}_{i=1}^{n_{k,\mathrm{C}}}$, which are the gradient and curl frequencies. To avoid computing the eigendecomposition of the Hodge Laplacian, we can approximate the true frequencies by sampling certain number of points in the frequency band $(0, \lambda_{k,\mathrm{G},\mathrm{m}}]$ and $(0, \lambda_{k,\mathrm{C},\mathrm{m}}]$ where the maximal gradient and curl frequencies can be computed by efficient algorithms, e.g., power iteration (Watkins, 2007; Sleijpen & Van der Vorst, 2000).

Here, to illustrate that the integral Lipschitz property of the SCFs helps the stability of NNs on SCs, we consider the effect of regularizer $r_{\mathrm{IL}}$ against perturbations in PSNNs and SCNNs with different $T_{\mathrm{d}}$ and $T_{\mathrm{u}}$ for the standard synthetic trajectory prediction. The regularization weight on $r_{\mathrm{IL}}$ is set as $5 \cdot 10^{-4}$ and the number of samples to approximate the frequencies is set such that the sampling interval is 0.01.

Fig. 13 shows the prediction accuracy and the relative distance between the edge outputs of the NNs trained with and without the integral Lipschitz regularizer in terms of different levels of perturbations. We see that the integral Lipschitz regularizer helps the stability of the NNs, especially for large SCF orders, where the

edge output is less influenced by the perturbations compared to without the regularizer. Meanwhile, SCNN with higher-order SCFs, e.g., $T_\mathrm{d} = T_\mathrm{u} = 5$, achieves better prediction than PSNN (with one-step simplicial shifting), while maintaining a good stability with its output not influenced by perturbations drastically.

We also measure the lower and upper integral Lipschitz constants of the trained NNs across different layers and features, given by $\max_{\lambda_{k,\mathrm{G}}} |\lambda_{k,\mathrm{G}} \tilde{h}_{k,\mathrm{G}}(\lambda_{k,\mathrm{G}})|$ and $\max_{\lambda_{k,\mathrm{C}}} |\lambda_{k,\mathrm{C}} \tilde{h}_{k,\mathrm{C}}(\lambda_{k,\mathrm{C}})|$, shown in Fig. 14. We see that the SCNN trained with $r_\mathrm{IL}$ indeed has smaller integral Lipschitz constants than the one trained without the regularizer, thus, a better stability, especially for NNs with higher-order SCFs.

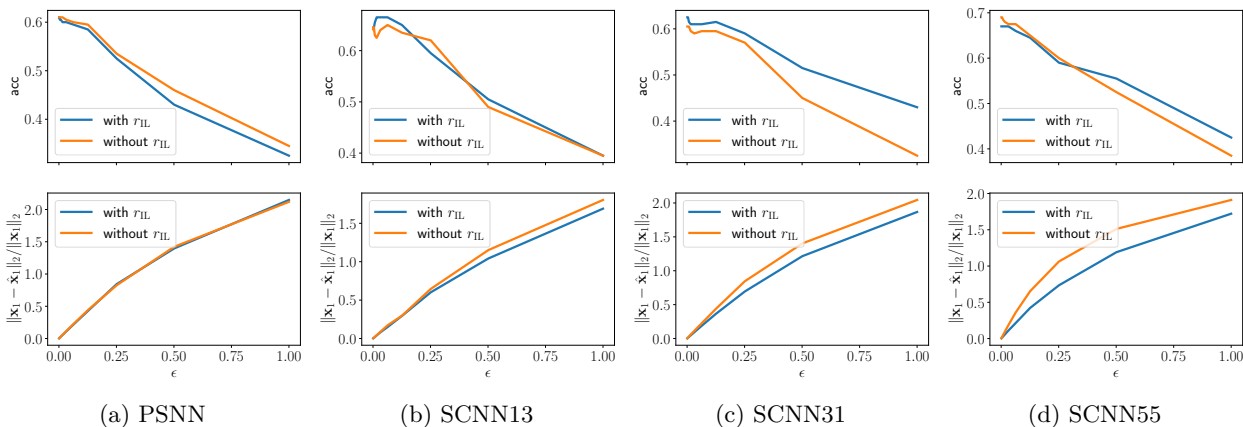

(a) PSNN        (b) SCNN13        (c) SCNN31        (d) SCNN55

Figure 13: Effect of the integral Lipschitz regularizer $r_\mathrm{IL}$ in the task of synthetic trajectory prediction against different levels $\epsilon$ of random perturbations on $\boldsymbol{L}_{1,\mathrm{d}}$ and $\boldsymbol{L}_{1,\mathrm{u}}$. We show the accuracy (Top row) and the relative distance between the edge output (Bottom row) for different NNs on SCs with and without $r_\mathrm{IL}$. SCNN13 is the SCNN with $T_\mathrm{d} = 1$ and $T_\mathrm{u} = 3$.

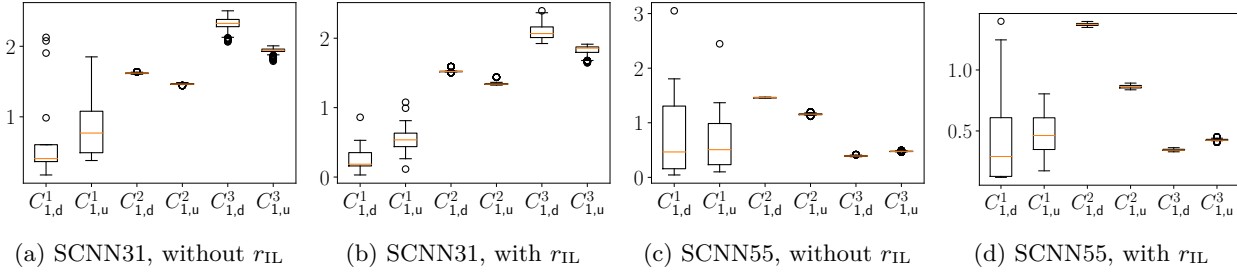

(a) SCNN31, without $r_\mathrm{IL}$    (b) SCNN31, with $r_\mathrm{IL}$    (c) SCNN55, without $r_\mathrm{IL}$    (d) SCNN55, with $r_\mathrm{IL}$

Figure 14: The integral Lipschitz constants of SCFs at each layer of the trained SCNNs with and without the integral Lipschitz regularizer $r_\mathrm{IL}$. We use symbols $c_{k,\mathrm{d}}^l$ and $c_{k,\mathrm{u}}^l$ to denote the lower and upper integral Lipschitz constants at layer $l$. Regularizer $r_\mathrm{IL}$ promotes the integral Lipschitz property, thus, the stability, especially for NNs with large SCF orders.