# OpenReview forum: "Hodge-Aware Convolutional Learning on Simplicial Complexes"
_TMLR — Accepted by TMLR_

### Review · Reviewer_yPbz · 2024-09-26

**Summary Of Contributions:**

In their paper "Hodge-aware convolutional learning on simplicial complexes", the authors suggest a new architecture (SCCNN) generalizing graph convolutional neural networks (GNN) from graphs to more general simplicial complexes. They argue that SCCNN have good theoretical properties and show that in experiments, they outperform some existing convolutional architectures on simplicial complexes.

**Audience:**

Yes

**Broader Impact Concerns:**

No concerns

**Claims And Evidence:**

Yes

**Requested Changes:**

MAJOR COMMENTS

* Description of experiments in Section 6 should be MUCH MORE DETAILED. I cannot follow the tasks and the experimental setups. For example in 6.1: what exactly is the dataset? What is K, are there any inputs beyond x_1? What is the data size? What is the training set and the test set? What is being predicted exactly? What exactly is the loss function of the network? Please really spell all of it out in detail.

* The same in 6.3: what are you predicting? "Predicting collaborations" -- what does it mean? If the simplex is present or not? How can the network make this prediction? If you remove one simplex from the traning data, then how can the network "fill it in"? Are you masking out simplexes? So what does the network get as input for that simplex then? Please explain the train/test splits and the loss function etc.

* Tables should have in bold not only max values, but everything that is not  signif different from the max value. E.g. in Table 2 it seems MPSN peforms as good as SCCNN given the uncertainties. So make it bold too. Same in other tables, esp. Table 4. By the way, what are the uncertainties? Std over multiple runs? How many runs? What was randomized? Please specify.

* Table 3: Would be good to have a column for performance at L=2, T=2. If I understand the logic correctly, the performance should be lower for many ablations.

* Are GNNs a special case of SCCNNs? Can you make an explicit comment about it?

* Please include data/code availability statement even though it cannot contain a link yet due to anonymity.


MINOR COMMENTS

* Section 2.2, "Simplicial signals": I did not undersatnd the technical defition of "alternating map". If f: S^k -> R^n, then f acts on S^k which is the set of all k-simplices defined above without orientation. I.e. orientation is not part of S^k, right? Then what does "alternating" mean?

* Section 2.2: the first time you say "edge flow", this term is not defined. Maybe state that "simplicial signal x_1 is also called edge flow".

* Equation 4: this equation contains lots of terms that are partially defined in Equation 5 below and partially in the subsequent text (H_k,d etc.). This makes it difficult to parse. I suggest to define all terms in a \gather environment together with Eq 5.

* Section 3.1, "Complexity" or earlier in "Example 3". I think it would be helpful to be explicit about the number of parameters, e.g. in a specific situation T_d = T_u = 2, and k=1. If I understand correctly, there are only VERY FEW parameters, like maybe 2*2*2 = 8 if counted correctly? That's not what one expects from a neural network "layer". Of course this is because you only consider d=1 dimensional features. But to be honest, I am even wondering if it is possible to learn anything useful at all, if original "features" are scalars...

* Section 5: I did not understand the sentences between Definitions 17 and 18.

* Section 5: I would like to see some comments on why the specific form of the perturbation in Definition 17 makes sense.

* Theorem 24 formulation is VERY complex and hard to parse. Can you provide some intuition about what it implies? I cannot even easily see if the bound is linear/quadratic/cubic in epsilon and r terms...

* Table 1: What is "interpolation" column?

* Table 1: Random Noise, total arbitrage column: SCCNN id should be bold, not SCCNN tanh. Same in the interpolation. Also in Curl Noise something should be bold.

* Typos: "generalzing", "arbitray", "messgae"

**Strengths And Weaknesses:**

Strengths: The paper is very mathematical and this is not my field; still, I found the exposition clear and illustrations very helpful and well done. The suggested architecture makes sense. I cannot fully judge on the mathematical theory as it went too far beyond my knowledge and I could not follow everything.

Weaknesses: The experimental section requires more details -- I could not fully understand the experimental setup.

Overall, I think the paper can be accepted to TMLR after a straightforward revision.

---

> ### Author Response · Authors · 2025-04-21
> **On major comments**
>
> Dear reviewer, we appreciate your careful comments on the draft and the positive recommendation. We notice the major comments are about experiment setups which were provided in Appendix G. in a separate file. We apologize for the unclarity this may have caused. We now put the supplementary materials in a single file with the main text. Please see our detailed replies below:
>
> __About simplex predictions__:
> The full experimental details are in Appendix G.3.
> - This experiment aims to predict if a 2-simplex exists (is closed) given its three nodes and three edges which pairwise connect the three nodes. This is a direct extension of the commonly studied link prediction which is to predict if a 1-simplex (edge) exists given the two nodes. In the context of collaboration complex, this indeed amounts to predicting collaborations between three while the classical link prediction is about collaborations between two.
> - About how to predict simplices:
>   - First of all, given all the triangles in the SC of order two, we use the number of citations on triangles (i.e., triadic collaborations) as a threshold to divide them into the positive and negative sets such that they contain about the same numbers of triangles.
>   - Then, for both positive and negative triangles, we split them into train/val/test sets (80/10/10 split ratio). Note that during training, we shall ONLY use the positive triangles which are in the training set. This is achieved by removing the ones in the val/test sets. Specifically, we remove the corresponding columns in the $B_2$ matrix, and then obtain the actual Hodge Laplacian used for training. This effectively means in the SC used for training, we only have positive triangles in the training set.
>   - We use the numbers of citations of one-, two- and three-collaborations as inputs for nodes, edges and triangles, and note that we again remove the inputs on triangles which are not in the training set. We use SCCNNs to learn representations of nodes and edges, then we use as input the concatenations of the learned edge (or node) representations of the to-be-classified triangles (from both positive and negative classes) to a MLP classifier, outputting a score used for classification. We train the models using the cross-entropy loss (because our task is essentially a binary classification of if a triangle is positive or negative).
>   - During testing, we use the learned representations to compute the score for each test triangle, then compare with the true labels to compute the AUC.
> - Overall, we designed this algorithm by generalizing the existing link prediction algorithm in a graph by Zhang & Chen 2018. We refer to [this python tutorial](https://docs.dgl.ai/en/0.8.x/tutorials/blitz/4_link_predict.html) for this graph link prediction, and to Appendix G for more details on simplex prediction and other additional experiments.
>
>
> __About forex experiments__:
> The full experimental details are in Appendix G.2
> - The dataset is from Jia et al. 2019 which records 210 exchange rates between 25 currencies at three timestamps. The SC order is 2 where we include all the triangles if three nodes are pairwise connected.
> - We perform the training on the first timestamp, validate on the second and test on the last.
> - For the denoising task, the input is the noisy data (synthetic random noise and curl noise); for the interpolation task, the input is 50% of the data. During training, the output is the true data and the loss function is the MSE between the output of a neural network and the true data. Note that here we do not have inputs on nodes or triangles, thus, the SCCNN reduces to SCNN. During testing, we measure the NMSEs between the true data and the denoised/interpolated ones, as well as the total arbitrage.
> - These details are in Appendix G. Overall, this experiment serves as a concrete example on how other models without hodge awareness can indeed learn the models to fit the data, like mpnn; however, they purely learn to reduce the loss without learning the physics (being curl-free), whereas SCCNN/SCNN can not only learn to fit the data but also learn the physics due to the awareness of the hodge theory involved in the convolution model. This validates our convolutional design choice.
>
> __About some tables__:
> We agree with the reviewer and have updated the tables accordingly. We reported the _mean $\pm$ std_ over 10 runs for all experiments and the randomization comes from the randomized train/test split, as well as the random initialization of the learnable weights.
> - For Table 3, we included a column for performance at L=2, T=2. The results are indeed lower for many ablations, but not significantly lower.
>
> __about GNNs__: Indeed, convolutional based GNNs are special cases of SCCNNs. In a simplicial complex of order 1, SCCNNs return to GNNs when not considering the edge-to-node feature projections.
>
> __About the data/code availability statement__:
> - We included a data/code availability statement.

---

> > ### Author Response · Authors · 2025-04-21
> > **on minor comments: 1**
> >
> > __about "alternating map"__:
> > In the case no orientation is chosen on simplices, we do not have "alternating" meaning, nor an alternating map. However, going back to the original question, if we should have an orientation for simplices. In general, for graph-related tasks and methods, the orientation of nodes is rather trivial and we often ignore it. Also, the graph Laplacian is defined as $L=D-A$, and this definition does not require orientation. However, this definition does not simply generalize to higher-order simplices. We instead followed the idea of treating discrete Laplacians as a second order differential operator, and to define first order difference operator, we need orientation, which is the oriented incidence matrix. This allows generalization from nodes to higher-order cases.
> >
> > __about the parameter complexity__:
> > We have the complexity discussed in detail in Appendix B.2. In the case of one-dim feature, one layer of SCCNN for learning edge features has indeed 8 parameters if we set $T_d = T_u =2$. This is in comparison to the standard one-layer GCN $X = \sigma(AXw)$, which actually has one parameter $w$ in the case of 1-dim feature. In practice, when we have the scalar input features, we usualy set the number of intermediate features large.
> >
> >
> > __about the perturbation model__:
> > In fact, this perturbation model is a direct generalization of the graph relative perturbation model (_Gamma et al. 2019b_) to simplicial complexes. We did not extensively reason the specific choice, but provided the related literature simply because we do not want to repeat the similar arguments. Here, to answer the reviewer's question: we can first think about the absolute perturbation model of a Laplacian--- $\hat{L} = L + E$ with the perturbation $E$ respecting $\Vert E\rVert \leq \epsilon$. This was commonly used until the improved (relative) model proposed by _Gamma et al. 2019b_. However, this will result in a stability bound that is potentially misleading given that the perturbation's norm is not tied to the norm of the Laplacian.
> > So, a natural way is to consider a relative version with the perturbation norm scaled by the norm of the Laplacian, i.e., $\lVert E \rVert = \Vert \hat{L}-  L \rVert   \leq \epsilon \lVert L \rVert$. However, both norms $\Vert L \rVert$  and $\Vert E \rVert$  are global properties of the error and the SC topology. This may imply that parts of the SC with small weights have large relative modifications because some other parts of the SC have larger weights.
> >
> >
> > The relative perturbation model of the form $\hat{L} = L + E L + L E$ defines the relativity in terms of the local properties of the Laplacian. Specifically, $L_{ij}$ is perturbed by ${[E L+L E]}_{ij}$, which is proportional to entries in the $i$th and $j$th rows of $L$, scaled by entries in $E$. (For the graph case, this relative part is proportional to the sum of the degrees of nodes $i$ and $j$ scaled by entries in $E$.) As $\lVert E \rVert$ grows, the perturbation makes $\hat{L}$ more dissimilar to $L$. But parts of the SC that are characterized by weaker connectivity (smaller weights) change by amounts that are proportionally smaller to the changes that are observed in parts of the SC characterized by stronger connectivity.
> >
> > After all, when moving from absolute to relative, the question is about whether to define the relativity in terms of the global norm, or to define it in terms of the matrix multiplication. We chose the latter based on this reasoning of the previous works.

---

> ### Author Response · Authors · 2025-04-21
> **on minor comments: 2**
>
> __about theorem 24's intuition__:
> Let us consider a concrete case with K = 2, L = 1. The bound now reads:
> $$
>     \\begin{bmatrix}
>     d_0 \\\ d_1 \\\ d_2
>     \\end{bmatrix}
>     = c_\sigma^L T
>     \\begin{bmatrix}
>     \beta_0 \\\ \beta_1 \\\ \beta_2
>     \\end{bmatrix}
>     = c_\sigma^L
>     \\begin{bmatrix}
>     t_0 \beta_0 + t_{0,u} \beta_1 \\\ t_{1,d} \beta_0 + t_1 \beta_1 + t_{1,u} \beta_2 \\\ t_{2,d} \beta_1 + t_2 \beta_2
>     \\end{bmatrix}
> \sim
>     \\begin{bmatrix}
>     O(\epsilon_{0}\beta_0 +(\varepsilon_{0,u} +\epsilon_{0,u})r_{0,u} \beta_1) \\\ O((\varepsilon_{1,d} + \epsilon_{1,d}) r_{1,d} \beta_0 + (\epsilon_{1,d} + \epsilon_{1,u})\beta_1 + (\epsilon_{1,u} + \varepsilon_{1,u}) r_{1,u} \beta_2)   \\\ O((\varepsilon_{2,d} +\epsilon_{2,d})r_{2,d}\beta_1 +  \epsilon_{2} \beta_2)
>     \\end{bmatrix}
> $$
> Recursively, we can analyze the case for $L=2,3,\dots$, etc. While they can be repetitive, we highlight the following points:
> - there is the dependence on factors of the adjacent-order simplices and this dependence becomes stronger when layer deepens. That is, for $L=1$, the node output bound is $O(\epsilon_{0}\beta_0 +(\varepsilon_{0,u} +\epsilon_{0,u})r_{0,u} \beta_1)$. And for $L=2$, it also depends on factors of 2-simplices. These factors are becoming multiplicative when $L=2$. We have factors such as $O(r_{0,u} r_{1,d}(\epsilon_{0} + \varepsilon_{0,u})\beta_0)$. This is what we aimed to show in Fig. 3(c) and Fig. 7 (Section 6.3).
> - On the other hand, in the case where there is no inter-simplicial coupling considered, we can obtain a bound for SCNN, which has order one dependence on epsilon: $d_k = c_\sigma^L t_k \beta_k = O((\epsilon_{k,d} + \epsilon_{k,u})\beta_k)$
>
> So, overall, from this theorem, we recommend to use higher-order convolutions with shallow layers rather than deep layers with lower-order convolutions so to avoid this stability dependence on factors of the adjacent-order simplices. This has been experimentally corroborated in Fig. 8 (Section 6.4).
>
> Moreover, this result implies how to further increase the stability while maintaining the performance by enforcing the convolutional filters to have better integral Lipschitz constants. For this, we introduced a regularizer, which has been experimentally studied as well.
>
>
> __About the rest__:
> The "interpolation" column in table 1 is the results given 50% of the rates as inputs.
>
> We appreciate the suggestions on some unclear terms, better formatting of equations and typos from the reviewer and have made the changes in the revised version accordingly. Please refer to the summary of changes.
>
> We hope our above replies clarify the questions and concerns raised by the reviewer. We are happy to provide any further clarifications if needed.

---

> > ### Comment · Reviewer_yPbz · 2025-04-22
> >
> > Thank you for the detailed replies. I am going to recommend acceptance.

---

### Review · Reviewer_sSBr · 2025-02-25

**Summary Of Contributions:**

The present paper studies learning from data defined on simplicial complexes, such as nodes, edges, triangles, etc. The key idea is to leverage the Hodge theorem that decomposes simplicial data into three orthogonal characteristic subspaces, such as the identifiable gradient, curl and harmonic components of edge flows.

The main contributions of the paper are then threefold: (1) proposing a unified network architecture called Simplicial Complex Convolutional Neural Network (SCCNN) that uncouples lower and upper simplicial adjacencies, accounts for inter-simplicial couplings, and performs higher-order convolutions to mitigate oversmoothing; (2) characterising the spectral behaviour and expressive power of SCCNN using Hodge decomposition, enabling effective and rational learning on simplicial complexes; and (3) providing a theoretical stability bound for SCCNN outputs against perturbations in simplicial connections, guiding the design of robust convolutional architectures. These contributions are validated through various simplicial tasks, demonstrating the effectiveness of Hodge-aware learning and stability analysis.

**Audience:**

Yes

**Claims And Evidence:**

No

**Requested Changes:**

n/a

**Strengths And Weaknesses:**

I have made a serious attempt to read this article, which seems interesting and well-polished, but I'm afraid that even after looking up several concepts, I still find it difficult to follow and I seriously suspect that this is due to a lack of expertise on my side, rather than the writing of the article, since the article is out of my expertise - for example I am not familiar with most of the cited literature. Hence I am afraid that I am not able to offer a useful review of the paper. I am sorry I cannot be of more help.

---

> ### Author Response · Authors · 2025-04-21
>
> Dear Reviewer, we appreciate your acknowledging that the subject of this work may fall outside of your expertise. We sincerely thank you for your time and effort in reading this manuscript, as well as for your concise summary of contributions of this work.

---

### Review · Reviewer_bo12 · 2025-04-14

**Summary Of Contributions:**

In this article, the authors propose a new generalization of simplicial complex convolutional neural networks (SCCNNs), which encompasses previous approaches from the literature, and, most importantly, satisfy key theoretical properties related to the discrete Hodge decomposition theorem of flows on simplicial complexes. As these flows can always be written uniquely as a linear combination of a harmonic, div-free and curl-free part, the authors propose to extend the usual simplicial convolutional filters (SCFs), which are intuitively polynomial functions of the up and down Laplacians, with similar polynomial functions of the projections from higher and lower dimensional flows. This uncoupling of the up and down Laplacians, incorporation of the lower and higher dimensional signals, and higher order convolutions (quantified by the polynomial degrees) allows to make use of the Hodge decomposition, as all Hodge subspaces are preserved under the operation, while keeping high discriminativity and expression levels. Moreover, the authors relate their operations to the Dirichlet energy of the signal (which do not necessarily decrease to zero, as in a traditional SCF, thanks to the projections), and provide a theoretical upper stability bound upon perturbations of both the Laplacians and simplicial flows, which depends on the integral Lipschitz constants of the SCFs. Finally, they provide several experiments showcasing the different properties of their SCCNNs.

**Audience:**

Yes

**Claims And Evidence:**

Yes

**Requested Changes:**

There are a few things and questions that I would like the authors to clarify or take into account:

- I did not understand the paragraph on information spillage at the end of page 8 (just before Proposition 13) and just before Section 6. I do not see why the nonlinearity plays a role in it, and I do not understand the connection with Figure 3(b). I think this should be discussed in greater details.

- In Section 3.2, if the goal is to avoid over-smoothing, wouldn't a lower bound on $D(x_k)$ be more interesting than an upper bound in Proposition 6?

- I am not sure how to interpret Assumption 21. What would be the values of $r_{k,d}$ and $r_{k,u}$ for a simplicial complex with $n$ simplices? Is there a way to control these values in practice?

- It would be nice to connect Theorem 24 more explicitly with the parameters  $w_{k,d/u,t}$, and $w'_{k,d/u,t}$. How should I tune these parameters in practice if I want a tight control on, e.g., the integral Lipschitz constants? Overall, it would be useful to make the discussion just before Section 6 more formal by quantifying it with these weights.

- In the experiment in Section 6.1, it is claimed that SNNs perform poorly because of a shared eigenvalue in the curl and gradient spaces. Is it just a guess, or has it really been observed numerically? I have the impression that those spaces sharing the exact same eigenvalue would never happen generically.

- The text should be proofread, there are several minor typos.

**Strengths And Weaknesses:**

Overall, I think it is a strong contribution, and I feel quite positive about this work. The writing is clear (although it could be better, see 'Requested Changes'), the proposed generalization of SCFs is natural, its theoretical guarantees are nice and important, and numerical experiments are conducted to investigate the practical behavior of the network. To me, this work would be a good fit to TMLR.

---

> ### Author Response · Authors · 2025-04-21
>
> Dear Reviewer,
>
> We appreciate your effort in reviewing our work and thank you for the positive recommendations, as well as the valuable comments. We here address your questions:
>
> __About information spillage__:
> In Figure 3(b), the top subfigure shows the Fourier coefficients in each Hodge subspace, i.e., the spectrum, of the input $\mathbf{x}$; whereas the bottom subfigure show the Fourier coefficients (the spectrum) of the output $\sigma(\mathbf{y})$. When the input has only gradient component, applying a linear SCF to it leads to an output with only gradient component too. This is however not the case when a nonlinearity is further applied. The goal here is to showcase the effect of nonlinearity that ``spills`` information over the other Hodge subspaces, i.e., the curl and harmonic ones in this case.
>
> The discussion in stability analysis just before section 6: For simplicial signals with large components at large gradient freuqncies, to ensure selectivity, we expect a more expressive SCF to be more sensitive to the change of gradient frequencies, as illustrated in the brown curve of Figure 3(a)(bottom). The selectivity-stability tradeoff indicates the stability would decrease in this case. We aim to highlight that the nonlinearity could help --- in the sense that part of the information at the large gradient frequencies would be spilled over the lower curl frequencies where it is more stable.
>
> __About oversmoothing__: Yes, we agree that a lower bound is also needed here. Notice that the objective eq.(3) can have a zero minimum. This corresponds to a lower bound of Dirichlet energy, $D(x_k) \geq \lVert x_{k-1} \rVert^2 + \lVert x_{k+1} \rVert^2$, where the inter-simplicial coupling acts as energy sources. In fact, our discussion followed by prop 6 says that the optimal solution is in the images of $\mathbf{B}\_{k+1}$ and $\mathbf{B}\_{k}$, implicitly giving the lower bound. We now have added the lower bound as well in proposition 6.
>
> __About assumption 21__: These two values upper bounds the energy of projections from the simplices of the adjacent orders. For example, $r_{1,d/u}$ measures the projections from nodes and triangles to edges. In practice, they are determined by the topological structure in the complex, so we cannot control them. Yet, they are always finite.
>
> __About how to control the weights in practice if one wants a tight control on, e.g., the integral Lipschitz constants__:
> In practice, to have a tight control on the integral Lipschitz constants, we proposed to introduce a regularization term during training, detailed in Appendix G.4.4.
> In Figure 13, we showed that imposing a regularization on the integral Lipschitz property when training the parameters, we are able to improve the stability while guaranteeing the performance in the trajectory prediction experiments. In Figure 14, we also showed that with the regularization, the integral Lipschitz constants of the trained SCFs are better controlled.
> As suggested by the reviewer, we now moved this right before section 6 and connect with theorem 24.
>
> __About if gradient and curl spaces can share the same eigenvalues__:
> Yes, they can, and we observe this for the experiment in section 6.1. We highlighted this below table 1. In general, the two nontrivial eigenspaces could share common eigenvalues. Here, we also examine the collaboration complex in Section 6.2. Please refer to [this anonymous figure](https://anonymous.4open.science/r/proj_p1-46A4/eigenvalues_collaboration_complex.png) for the visualizations where the blue and orange points are the nonzero eigenvalues of the down and up Laplacians, respectively, and the green horizontal lines are the common eigenvalues shared by the two spaces.
>
> We hope our answers address your questions. These questions help improve the paper and we appreciate them. We have made changes in the paper accordingly. Please refer to the summary of changes.

---

> > ### Comment · Reviewer_bo12 · 2025-04-28
> >
> > Thank you for the detailed answers.

---

### Author Response · Authors · 2025-04-21
**summary of changes in the paper**

Dear three reviewers,

Here's we summarize the changes we have made in the revised paper.

First, we appended the supplementary materials to the main paper so that they are in a single-pdf file. This helps readers refer to the appendices for, e.g., experimental details and additional results.

Second, we added/modified the following parts in the paper:
- We detailed the caption in Figure 3 and added more discussion on information spillage, and improved the discussion before section 6. There, we also added a small discussion on a regularizer to control the integral Lipschitz properties when training the parameters.
- We added the lower bound in proposition 6.
- We made some values in tables 2 and 4 in bold. We also added another column of results in Table 3 where the hyperparameters remain the same as SCCNN.
- We added reproducibility statement in the end of the paper.
- We fixed the typos.

We appreciate the time and effort from all three reviewers on this paper.

---

### Decision · Action_Editor_6p41 · 2025-08-11

**Recommendation:** Accept as is

**Comment:**

The reviewers were unanimous in recommending acceptance of the manuscript. They recognized its methodological, theoretical and experimental contributions to the problem of learning from simplicial complexes with neural networks. The authors addressed the reviewer's requests for changes for the revised version of the manuscript.

**Audience:**

Yes. The reviewers were unanimous that this paper has an audience at TMLR.

**Claims And Evidence:**

Yes. No concerns regarding evidence were raised by reviewers.